# Green energy and steel imports reduce Europe's net-zero infrastructure needs

Fabian Neumann [1] ✉, Johannes Hampp [2] & Tom Brown [1]

Importing renewable energy to Europe may offer many potential benefits, including reduced energy costs, lower pressure on infrastructure development, and less land use within Europe. However, open questions remain: on the achievable cost reductions, how much should be imported, whether the energy vector should be electricity, hydrogen, or derivatives like ammonia or steel, and their impact on Europe's infrastructure needs. This study integrates a global energy supply chain model with a European energy system model to explore net-zero emission scenarios with varying import volumes, costs, and vectors. We find system cost reductions of 1-10%, within import cost variations of ± 20%, with diminishing returns for larger import volumes and a preference for methanol, steel and hydrogen imports. Keeping some domestic power-to-X production is beneficial for integrating variable renewables, leveraging local carbon sources and power-to-X waste heat. Our findings highlight the need for coordinating import strategies with infrastructure policy and reveal maneuvering space for incorporating non-cost decision factors.

Importing renewable energy to Europe may offer several advantages for achieving a swift energy transition. It might lower costs, help circumvent the slow domestic deployment of renewable energy infrastructure and reduce pressure on land usage in Europe. Many parts of the world have cheap and abundant renewable energy supply potentials that they could offer to existing or emerging global energy markets[1-8]. Partnering with these regions could help Europe reach its carbon neutrality goals while stimulating economic development in exporting regions.

However, even if energy imports are economically attractive for Europe, a strong reliance may not be desirable because of energy security concerns. Awareness of energy security has risen since Russia constrained fossil gas supplies to Europe in 2022[9], at a time when the 27 member states of the European Union (EU27) imported around two-thirds of their fossil energy needs[10]. Europe must take care to avoid repeating the mistakes of previous decades when it became dependent on a small number of exporters with market power and reliant on rigid pipeline infrastructure.

Europe's strategy for clean energy imports will also strongly affect the requirements for domestic energy infrastructure. Previous research found many ways to develop a self-sufficient energy system[11-13]. To support such scenarios without energy imports into Europe, reinforcing the European power grid or building a hydrogen network was often identified as beneficial[14,15]. However, depending on the volumes and vectors of imports (electricity, hydrogen, or hydrogen derivatives) and levels of industry migration, Europe might not need to expand its hydrogen pipeline infrastructure. Most hydrogen is used to make derivative products (e.g., ammonia for fertilisers, sponge iron for steel, or Fischer-Tropsch fuels for aviation and shipping)[14]. If Europe imported these products at scale, much of the hydrogen demand would fall away. In consequence, this would reduce the need for hydrogen transport. However, if hydrogen itself were imported and to be transported to today's industry clusters, this would require a pipeline topology tailored to connecting these to the hydrogen arriving from North Africa or maritime entry points across Europe.

Policy has reflected these different visions for imports in various ways. In particular, hydrogen imports have recently attracted considerable interest, with plans of the European Commission[16] to import 10 Mt (333 TWh, lower heating value) hydrogen and derivatives by 2030. New financing instruments, like the European Hydrogen Bank[17] or H2Global[18] are set up to support the scale-up of green hydrogen imports. The desire to import hydrogen and derivative products is also

[1]Department of Digital Transformation in Energy Systems, Institute of Energy Technology, Technische Universität Berlin, Berlin, Germany. [2]Potsdam Institute for Climate Impact Research (PIK), Member of the Leibniz Association, Potsdam, Germany. ✉e-mail: f.neumann@tu-berlin.de

present in various national strategies[19]. In particular, Germany's new import strategy plans to cover up to 70% of its demand for hydrogen and its derivatives through imports by 2030 and highlights bilateral partnerships as well as the expansion of import infrastructure as a means to accomplish this[20,21]. Conversely, hydrogen roadmaps of Denmark[22], Ireland[23], Spain[24], and the United Kingdom (UK)[25], recognise these countries' potential to become major exporters of renewable energy, whereas France's strategy focuses on local hydrogen production to meet domestic needs[26]. Beyond direct energy imports, the Draghi report[27] also raises broader concerns about European industrial competitiveness and discusses the benefits of relocating energy-intensive industries to renewable-rich regions inside Europe. In addition, European grid development plans[28] reveal renewed enthusiasm for electricity imports via ultra-long high-voltage direct current (HVDC) cables, evolving from early DESERTEC[29] ideas to contemporary proposals like the Morocco-UK Xlinks project[30].

While many previous academic studies have evaluated the cost of 'green' renewable energy and energy-intensive material imports in the form of electricity[5,31–35] hydrogen[2,6,36–42], ammonia[7,43–45], methane[2,46,47], steel[48–50], carbon-based fuels[4,51,52], or a broader variety of power-to-X fuels[1,3,8,53–56], these do not address the interactions of imports with European energy infrastructure requirements. On the other hand, among studies dealing with the detailed planning of net-zero energy systems in Europe, some do not consider energy imports[11,13,15], while others only consider hydrogen imports or a limited set of alternative endogenously optimised import vectors[14,57–61]. Only a few consider at least elementary cost uncertainties[42,62], and none investigate a larger range of potential import volumes across subsets of available import vectors.

In this study, we explore the full range between the two poles of complete self-sufficiency and wide-ranging renewable energy imports into Europe in scenarios with high shares of wind and solar electricity and net-zero carbon emissions. We investigate how the infrastructure requirements of a self-sufficient European energy system that exclusively leverages domestic resources from the continent may differ from a system that relies on energy imports from outside of Europe. For our analysis, we integrate an open optimisation model of global energy supply chains, TRACE[54], with a spatially and temporally resolved sector-coupled open-source energy system optimisation model for Europe, PyPSA-Eur[63], to investigate the impact of imports on European energy infrastructure needs. We evaluate potential import locations and costs for different supply vectors, by how much system costs can be reduced through imports, and how their inclusion affects deployed transport networks, storage and backup capacities. For this purpose, we perform sensitivity analyses interpolating between very high levels of imports and no imports at all, exploring low and high costs for imports to account for associated uncertainties, and system responses to the exclusion of subsets of import vectors, in order to identify the cost-effective manoeuvreing space.

As possible import options, we consider electricity by transmission line, hydrogen as gas by pipeline and liquid by ship, methane as liquid by ship, liquid ammonia, crude steel and its precursor, hot briquetted iron (HBI), methanol and Fischer-Tropsch fuels by ship. Each energy vector has unique characteristics with regard to its production, storage, transport and consumption. Electricity offers the most flexible usage but is challenging to store and requires variability management if sourced from wind or solar energy. Hydrogen is easier to store and transport in large quantities, but at the expense of conversion losses and less versatile applications. Large quantities could be used for backup power and heat, steel production, industry feedstocks and the domestic synthesis of shipping and aviation fuels. On the other hand, imported synthetic carbonaceous fuels like methane, methanol and Fischer-Tropsch fuels could largely substitute the need for domestic synthesis. There is more experience with storing and transporting these fuels, and part of the existing infrastructure could potentially be

reused or repurposed. However, they require a sustainable carbon source and, particularly for methane, effective carbon management and leakage prevention[64]. Ammonia is similarly easier to handle than hydrogen, but does not require a carbon source. However, it faces safety and acceptance concerns due to its toxicity and potentially adverse effects on the global nitrogen cycle[65,66]. Its demand in Europe is mostly driven by fertiliser usage. Crude steel and HBI represent the import of energy-intensive materials and offer low long-distance transport costs.

The PyPSA-Eur[63] model co-optimises the investment and operation of generation, storage, conversion and transmission infrastructures in a single linear optimisation problem. The model is further given the opportunity to relocate ammonia and primary crude steel production within Europe, capturing potential renewables pull effects within Europe and abroad[67–69]. We resolve 115 regions comprising the European Union without Cyprus and Malta, as well as the United Kingdom, Norway, Switzerland, Albania, Bosnia and Herzegovina, Montenegro, North Macedonia, Serbia, and Kosovo. In combination with a 4-hourly-equivalent time resolution for the weather year of 2013, grid bottlenecks, renewable variability, and seasonal storage requirements are sufficiently captured. The model includes regional demands from the electricity, industry, buildings, agriculture and transport sectors, international shipping and aviation, and non-energy feedstock demands in the chemicals industry. Transmission infrastructure for electricity, gas and hydrogen, and candidate entry points like existing and prospective liquefied natural gas (LNG) terminals, as well as cross-continental pipelines, are also represented. However, no pathways are modelled in this overnight scenario, and the model has perfect operational foresight. We utilise techno-economic assumptions for 2040 and enforce net-zero $CO_2$ emissions and limit the annual carbon sequestration to 200 $Mt_{CO_2}$ $a^{-1}$, similar to the 250 $Mt_{CO_2}$ $a^{-1}$ highlighted in the European Union's carbon management strategy[70]. This suffices to offset unabated industrial process emissions and limits the use of fossil fuels beyond that, whose emissions are compensated either through capturing emissions at source or by carbon dioxide removal. More details are included in the Methods section.

## Results

### Assessment of energy and material import unit costs

Green fuel and steel import costs seen by the model are based on an extension of recent research by Hampp et al.[54], who assessed the levelised cost of energy exports for different green energy and material supply chains from various world regions to Europe. Our selection of exporting regions comprises all 53 coloured or dotted regions in Fig. 1a. In the TRACE optimisation model[54], regional wind and solar potentials are assessed based on prevailing weather conditions and land availability while prioritising projected domestic demand. In combination with the techno-economic modelling of the various fuel-specific supply chains stages (Supplementary Fig. 1), the lowest levelised supply cost for each carrier, exporter and importer combination are determined for a reference volume of 500 TWh $a^{-1}$ (or 100 Mt $a^{-1}$ of steel/HBI), thus incorporating the trade-off between import cost and import location (Fig. 1b). Unlike domestic electrofuel synthesis in Europe, which could use captured $CO_2$ from point sources, direct air capture is assumed to be the only carbon source of imported fuels. Concepts involving the shipment of captured $CO_2$ from Europe to exporting regions for carbonaceous fuel synthesis or permanent sequestration of $CO_2$ captured by direct air capture abroad are not considered[71,72].

The import costs for each combination of carrier, exporter and importer are then included as supply options in the PyPSA-Eur model. Hydrogen and methane can be imported where there are LNG terminals in operation or under construction, or where pipeline entry points exist (except for entry points from Russia). Due to higher volatility, electricity imports are endogenously optimised, meaning

that the capacities and operation of wind and solar generation, as well as storage in the respective exporting regions and the HVDC transmission lines, are co-planned with the rest of the European system.

Ammonia, carbonaceous fuels, and ferrous materials are not spatially resolved in the model, assuming they can be transported within Europe at negligible cost. Thus, their specific import location is not

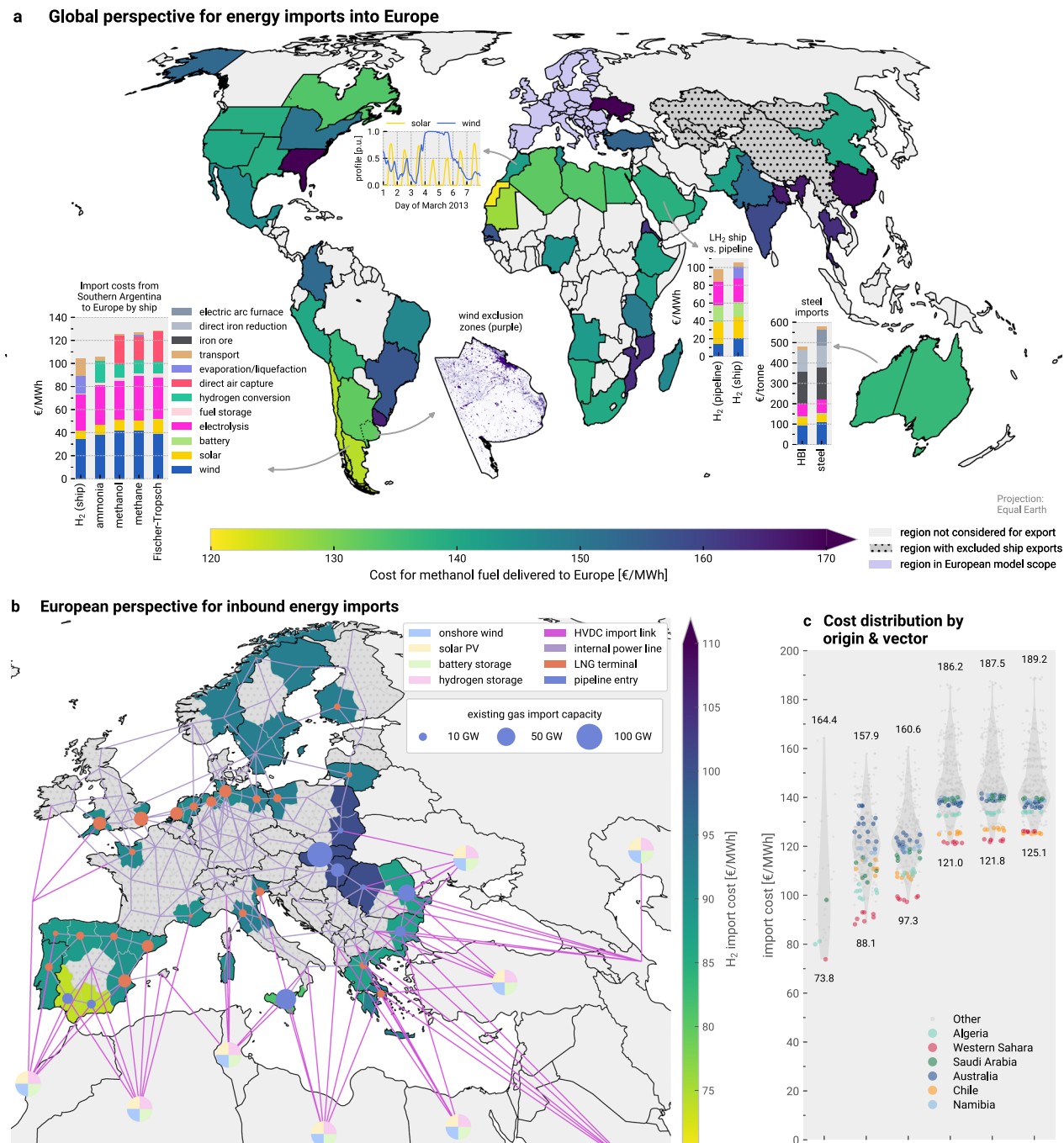

**Fig. 1 | Overview of considered import options into Europe. a** shows the regional differences in the cost to deliver green methanol to Europe (choropleth layer), the cost composition of different import vectors (bar charts), an illustration of the wind and solar availability in Morocco, and an illustration of the land eligibility analysis for wind turbine placement in the region of Buenos Aires in Argentina. **b** depicts considered potential entry points for energy imports into Europe like the location of existing and planned liquefied natural gas (LNG) terminals and gas pipeline entry points, the lowest costs of hydrogen imports in different European regions (choropleth layer), and the considered connections for long-distance high-voltage direct current (HVDC) import links and hydrogen pipelines from the Middle East and North Africa (MENA) region, Turkey, Ukraine and Central Asia. **c** displays the

distribution and range of import costs for different energy carriers and entry points with indications for selected origins from the TRACE model (violin charts), i.e., differences in identically coloured markers are due to regional differences in the transport costs to alternative entrypoints. These are more variable for liquid hydrogen as transport distance is a more substantial cost factor for this import vector. Costs are given for techno-economic assumptions for 2040. Supplementary Fig. 3 shows the world map for the lowest hydrogen import costs by pipeline or ship into Europe. Source data are provided as a Source Data file. Maps made with Natural Earth. Subnational regions created from geoBoundaries[133]. LH$_2$ = liquefied hydrogen; PV = photovoltaics.

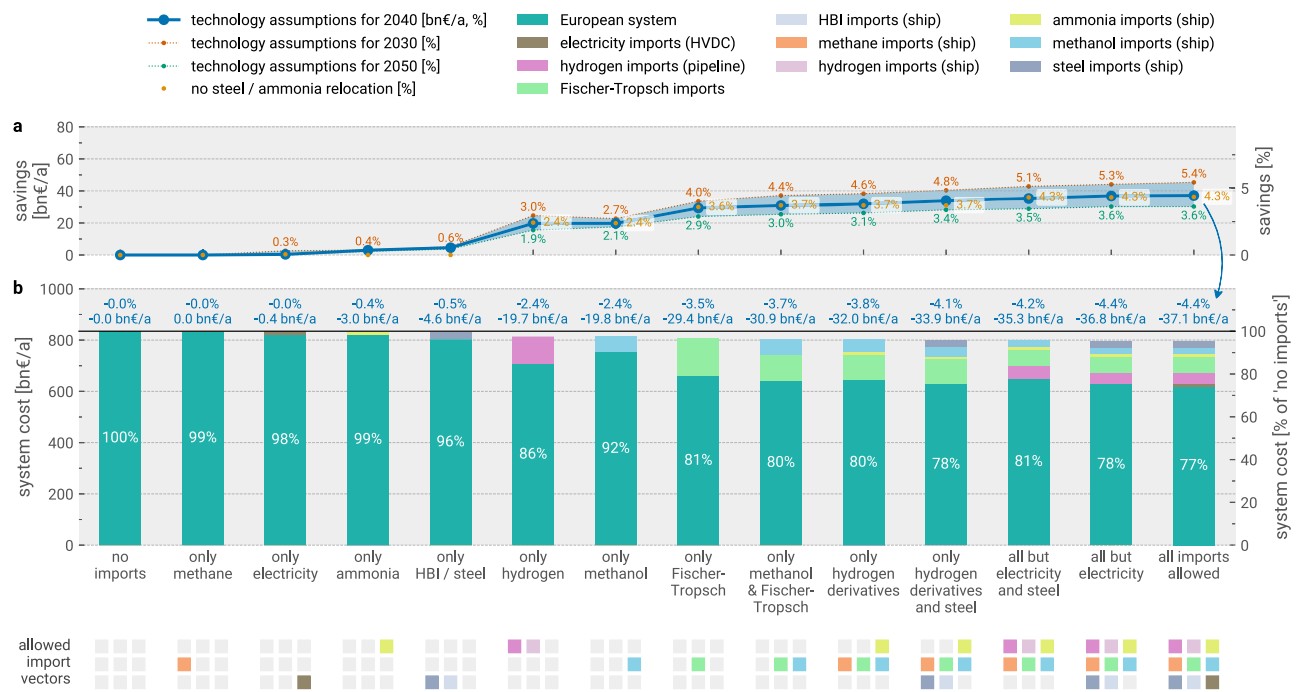

**Fig. 2 | Potential for cost reductions with reduced sets of import options available.** Subsets of available import options are sorted by ascending cost reduction potential. **a** shows the profile of total system cost savings. Shaded ranges show cost savings (%) for technology assumptions for 2030 and 2050, in addition to the default assumptions for 2040. Cost savings (%) are also shown for scenarios without crude steel and ammonia relocation, in addition to the default case where relocation is allowed. **b** shows the composition and extent of imports in relation to total energy system costs. Percentage numbers in the bar plot indicate the share of total system costs spent on domestic energy infrastructure. Alternative scenarios of this figure with higher and lower import cost assumptions are shown in Supplementary Figs. 13 and 14. Source data are provided as a Source Data file. HBI = hot briquetted iron; HVDC = high-voltage direct current; bn = billion.

determined. An import limit of 500 TWh per region for the sum of all exports is imposed to prevent over-reliance on single exporters.

For imports of hydrogen by pipeline, North African regions offer the lowest cost (ca. 74-88 € MWh$^{-1}$, Supplementary Fig. 3). Importing hydrogen by ship is substantially more expensive due to liquefaction and evaporation losses, with a cost difference of 18% between each vector's lowest cost supplier (Supplementary Fig. 7). For hydrogen derivatives, Argentina and Chile offer additional potential for low-cost imports, for instance, 125–132 € MWh$^{-1}$ for Fischer-Tropsch fuels or 548–566 € t$^{-1}$ for steel. These values are similar to those achieved in the Maghreb region. Further notable regions include Australia and Canada. Methanol is slightly cheaper than the Fischer-Tropsch route because it is assumed to be more flexible with a 20% minimum part load compared to 50% for Fischer-Tropsch synthesis[73]. The lower process flexibility shifts the energy mix towards solar electricity and causes higher levels of curtailment and battery storage, increasing costs (Supplenmentary Fig. 9). The transport costs of $CH_4(l)$ are lower than for $H_2(l)$ since the liquefaction consumes less energy and individual ships can carry more energy with $CH_4(l)$. Pipeline imports of $CH_4(g)$ were also considered, but costs were higher than for $CH_4(l)$ shipping under the assumption that new pipelines would have to be built or renewed.

### Cost savings for fuel and material import combinations

In Fig. 2, we first explore the cost reduction potential of various energy and material import options. Without energy imports, total energy system costs add up to 836 bn€$_{2020}$ a$^{-1}$. By enabling imports from outside Europe and considering all import vectors, we find a potential reduction of total energy system costs by up to 37 bn€$_{2020}$ a$^{-1}$, using technology assumptions for 2040. This corresponds to a relative reduction of 4.4%, which remains nearly unchanged regardless of the domestic crude steel and ammonia production relocation potential. With more long-term (2050) or near-term (2030) technology

assumptions, the cost savings range from 3.6% to 5.4%, with higher cost savings achieved with near-term technology assumptions (Supplementary Fig. 12).

For cost-optimal imports, around 77% of these costs are used to develop domestic energy infrastructure. The remaining 23% are spent on importing a volume of 50 Mt of green steel and around 1498 TWh of green energy, which is around 13% of the system's total energy supply (Fig. 3). Our results show a cost-effective import mix consisting primarily of liquid carbon-based fuels, hydrogen, and steel imports with small volumes of ammonia and electricity imports.

Next, we investigate the impact of restricting the available import options to subsets of import vectors. We find that if only hydrogen can be imported, cost savings are reduced to 20 bn€$_{2020}$ a$^{-1}$ (2.4%), with pipeline-based hydrogen imports being preferred to imports as liquid by ship. By importing a larger volume of hydrogen as an intermediary carrier (1338 TWh instead of 576 TWh, Supplementary Fig. 20), low-cost renewable energy from abroad can still be leveraged for the synthesis of derivative products in Europe. However, the benefit is reduced as domestic $CO_2$ feedstocks from industrial sources are depleted.

Conversely, when direct hydrogen imports are excluded from the available import options, cost savings are close to the maximum with 34 bn€$_{2020}$ a$^{-1}$ (4.1%). This indicates that the benefit of using domestically captured biogenic or fossil $CO_2$ is similar to tapping into low-cost renewable resources abroad. Focusing imports exclusively on liquid carbonaceous fuels derived from hydrogen, i.e., methanol or Fischer-Tropsch fuels, still achieves high cost savings of 31 bn€$_{2020}$ a$^{-1}$ (3.7%), which is due to the smaller demand or variety of applications for ammonia, methane, and steel compared to liquid carbonaceous fuels. Thus, excluding them has a small effect on cost savings. This aligns with the finding that restricting options to only methane, ammonia, or ferrous material imports yields negligible to small cost savings below 5 bn€$_{2020}$ a$^{-1}$ (0.6%). Negligible cost savings were also

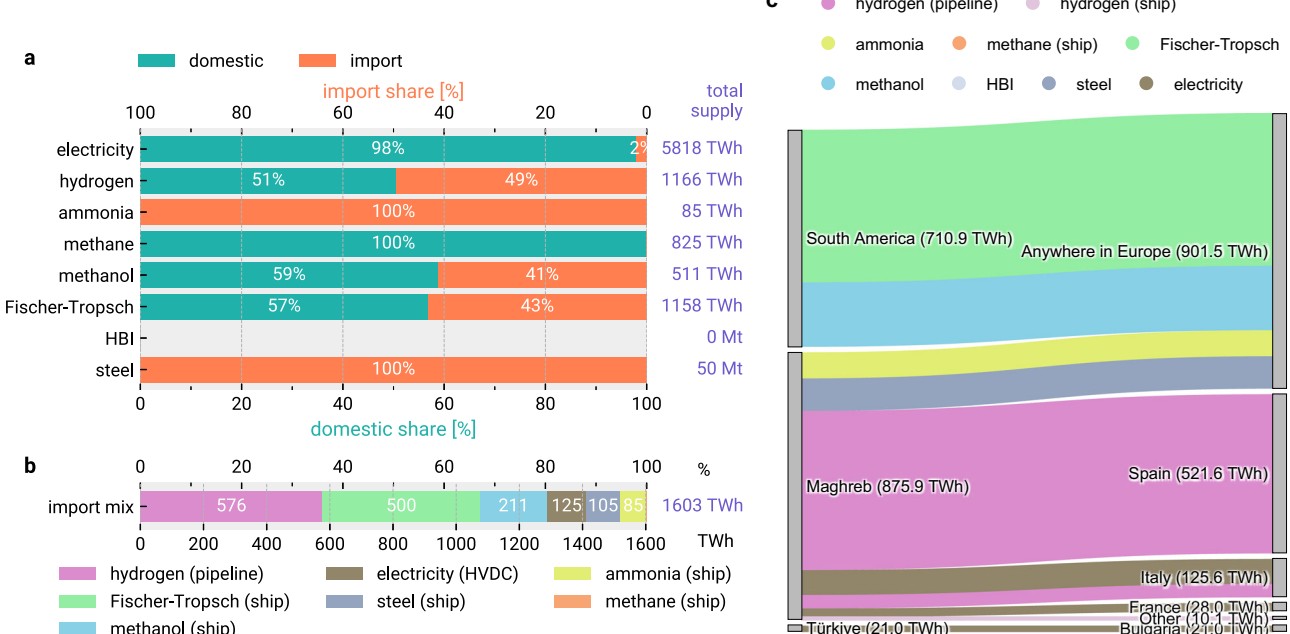

**Fig. 3 | Shares of imports and domestic production by carrier, optimised import mix and trade flows. a** shows the optimised import shares by carrier for the import scenario with flexible carrier choice and volume, technology assumptions for 2040, and allowed relocation of crude steel and ammonia production. **b** shows the total supply for each carrier for the same scenario. **c** shows trade flows for the same scenario as a Sankey diagram. The flows are aggregated to broader regions to emphasise that alternative origin-destination pairs could often yield similar results. Import shares for further import scenarios are included in Supplementary Figs. 20 and 21. Steel is included in energy terms, applying a factor of 2.1 kWh kg$^{-1}$ as released by the oxidation of iron. Source data are provided as a Source Data file. HBI = hot briquetted iron; HVDC = high-voltage direct current.

found for the direct import of electricity, as it poses more challenges for integration into the European system.

Overall, while varying import costs within ±20% affects the magnitude of attainable cost savings, the relative impact of restricting specific import options remains broadly consistent (Supplementary Figs. 13 and 14). Likewise, using more long-term (2050) or near-term (2030) technology assumptions do not affect the dynamics substantially (Fig. 2).

**Import dynamics for different energy carriers**

Figure 3a, b outline which carriers are imported in which quantities in relation to their total supply under default assumptions when the vector and volume can be flexibly chosen ('all imports allowed' in Fig. 2). In energy terms, cost-optimal imports comprise around 45% carbonaceous fuels, 35% hydrogen, and less than 10% electricity. Noticeably, all primary crude steel and ammonia for fertilisers is imported, whereby steel imports are preferred over HBI imports. Around half of the total hydrogen supply is imported, matching the ratio of the 2030 REPowerEU targets[16]. Hydrogen is imported so that it can be processed into derivative products domestically rather than used for direct applications for hydrogen. Smaller import shares are observed for electricity, which is largely supplied from domestic resources, because of higher costs and losses in electricity transmission than other import vectors, and for methane, which is supplied from domestic fossil and biogenic sources (Supplementary Fig. 24).

In terms of trade flows (Fig. 3c), we observe carbonaceous fuel imports by ship from South America – leveraging low transport costs of dense liquid fuels – as well as ammonia, steel, and hydrogen imports from the Maghreb region. Hydrogen is mainly received by pipeline in Spain. Moreover, due to its proximity to Italy, some electricity imports are received by HVDC connections from Tunisia. While the model suggests trade routes from particular regions, we aggregate these to broader regions to emphasise that alternative origin-destination pairs within the regions could often yield similar results (Supplementary Figs. 7–10).

To explain the import shares in Fig. 3a in more detail, we compare import costs with average domestic production cost split by cost and revenue components in Fig. 4. First, for the scenario without imports, imported fuels appear substantially cheaper than domestic production, which is mostly driven by levelised cost differences of wind and solar electricity supply. The high demand for hydrogen derivatives (Supplementary Fig. 4) means that the most attractive domestic potentials for renewable electricity and captured carbon dioxide have been exhausted. Consequently, power from wind and solar needs to be produced in regions with worse capacity factors and higher levelised costs.

Part of this gap is closed when hydrogen imports are allowed. By sourcing cheaper hydrogen from outside Europe, the domestic costs of derivative fuel synthesis are reduced. However, the large remaining volume of $CO_2$ handled in the European system for use and sequestration (Supplementary Fig. 25) means that direct air capture is still the price-setting technology for $CO_2$, as economic applications for biogenic and industrial carbon capture (i.e., those with high full load hours) are depleted.

With all import vectors allowed, we see minimal cost differences between domestic production and imports as the supply curves reach equilibrium (Supplementary Fig. 22). This is because imports of hydrogen and derivative products lower the strain on the domestic supply curves for hydrogen and carbon dioxide. Thereby, domestic production would only ramp up where it competes with imports and associated infrastructure costs. This was the case for hydrogen, methanol, and Fischer-Tropsch fuels in the British Isles and parts of Southern Europe and Nordic countries (Fig. 4). Consequently, not all hydrogen is imported, but some domestic production is retained.

**Sensitivity of potential cost savings to import costs**

Thus far, the presented findings originate from a central estimate for the import cost. However, the cost-optimal import mix strongly depends on the assumed import costs. This uncertainty is addressed in

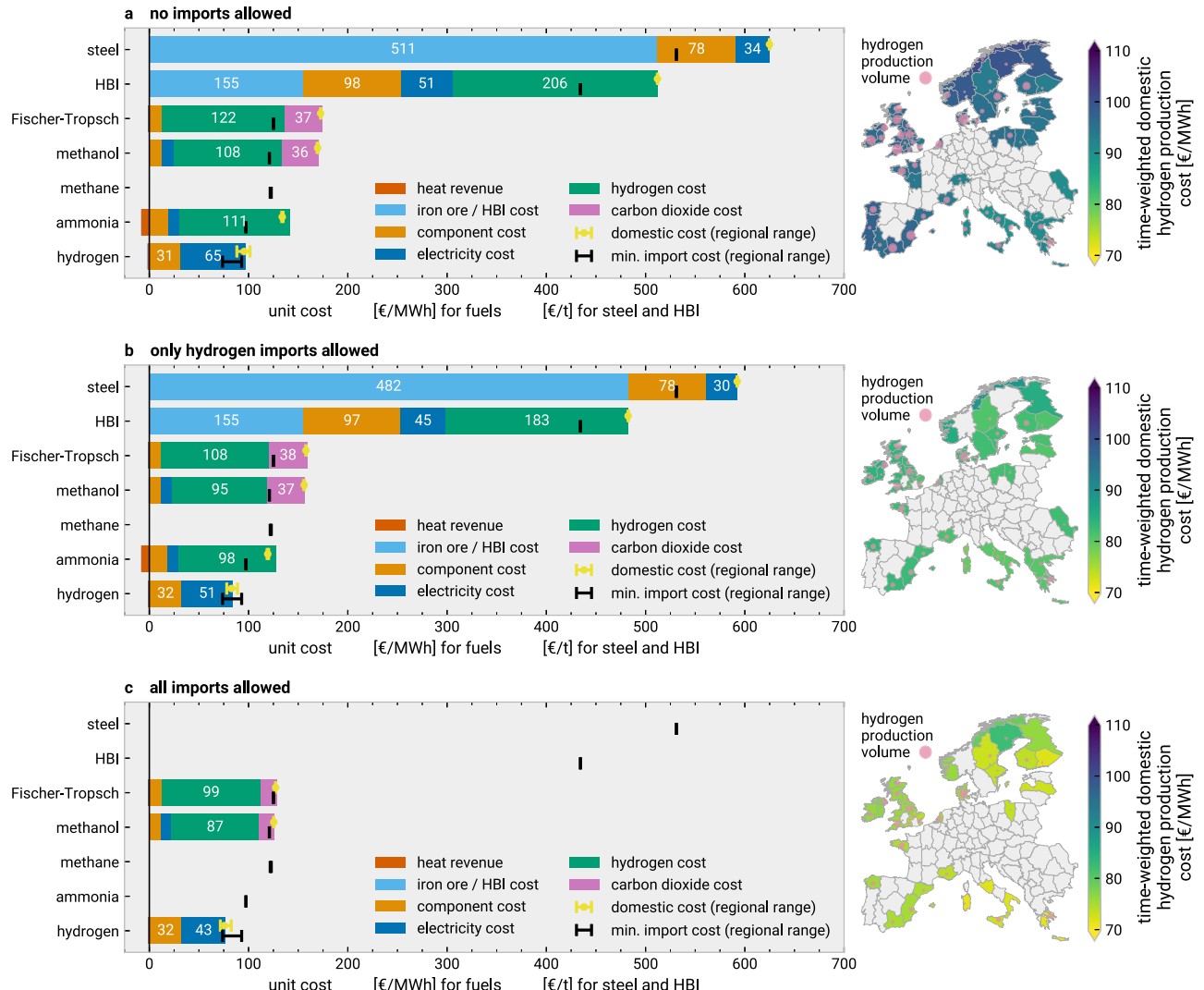

**Fig. 4 | Comparison of domestic synthetic production costs and import costs for varying import scenarios.** The three panels (**a**–**c**) refer to different import scenarios. In each panel, the bar charts show the production-weighted average costs of domestic production of steel, hydrogen and its derivatives, split into its cost and revenue components. These have been computed using the marginal prices of the respective inputs and outputs for the production volume of each region and snapshot. Capital expenditures are distributed to hours in proportion to the production volume. Missing bars indicate that no domestic production occurred in the scenario, e.g., for the case of methane, where all demand is met by biogenic and fossil methane and no synthetic production occurred (cf. energy balances in Supplementary Figs. 24–25). All hydrogen is produced from

electrolysis; i.e., the model did not choose to produce hydrogen via steam methane reforming with or without carbon capture. For each bar, the yellow error bars show the range of time-averaged domestic production costs across all regions. The black error bars show the range of import costs across all regions. The maps on the right of each panel relate the hydrogen production volume to the weighted cost of domestic hydrogen production (colorbar). The shown scenarios use technology assumptions for 2040, allow crude steel and ammonia relocation, and do not constraint import volume in (**b**, **c**). Confer Supplementary Fig. 22 for information on the domestic cost supply curves. Source data are provided as a Source Data file. Maps made with Natural Earth. HBI = hot briquetted iron.

Fig. 5. Figure 5a highlights the extensive range in potential cost reductions if higher or lower import costs could be attained and underlines the resulting variance in cost-effective import mixes. Within ± 30% of the default import costs applied to all carriers, total cost savings vary between 2 bn€$_{2020}$ a$^{-1}$ (0.3%) and 112 bn€$_{2020}$ a$^{-1}$ (13.5%). Within this range, import volumes vary between 500 and 2646 TWh. Across most scenarios, there is a stable role for ammonia and liquid carbonaceous fuel imports. Within a narrower ± 20% range, hydrogen imports also appear in larger quantities, while steel imports become less attractive with cost increases of 10% or more. Electricity imports grow with declining costs.

However, not all carriers are equally affected by technology cost variations. Fuel synthesis technologies do not influence electricity imports, and only carbon-based fuels are subject to the cost of $CO_2$ supply. We find that when the relative cost variation is not applied to

electricity imports (Fig. 5b), they remain less attractive than other vectors, even when those alternative vectors face a 20% cost rise.

One central assumption regarding costs for carbon-based fuels is that imported fuels rely on direct air capture (DAC) as a carbon source. Arguments for this assumption relate to the potential remoteness of the ideal locations for renewable fuel production or the absence of industrial point sources in the exporting region. In contrast, domestic electrofuels can mostly use less expensive captured biogenic or fossil carbon dioxide from industrial processes. Therefore, the higher cost for DAC partially cancels out the savings from utilising better renewable resources abroad. This is one of the reasons why there is substantial power-to-X production in Europe, even with corresponding import options. However, the availability of cheaper (biogenic) $CO_2$ in exporting regions would lower the costs of carbonaceous fuel imports (Table 1).

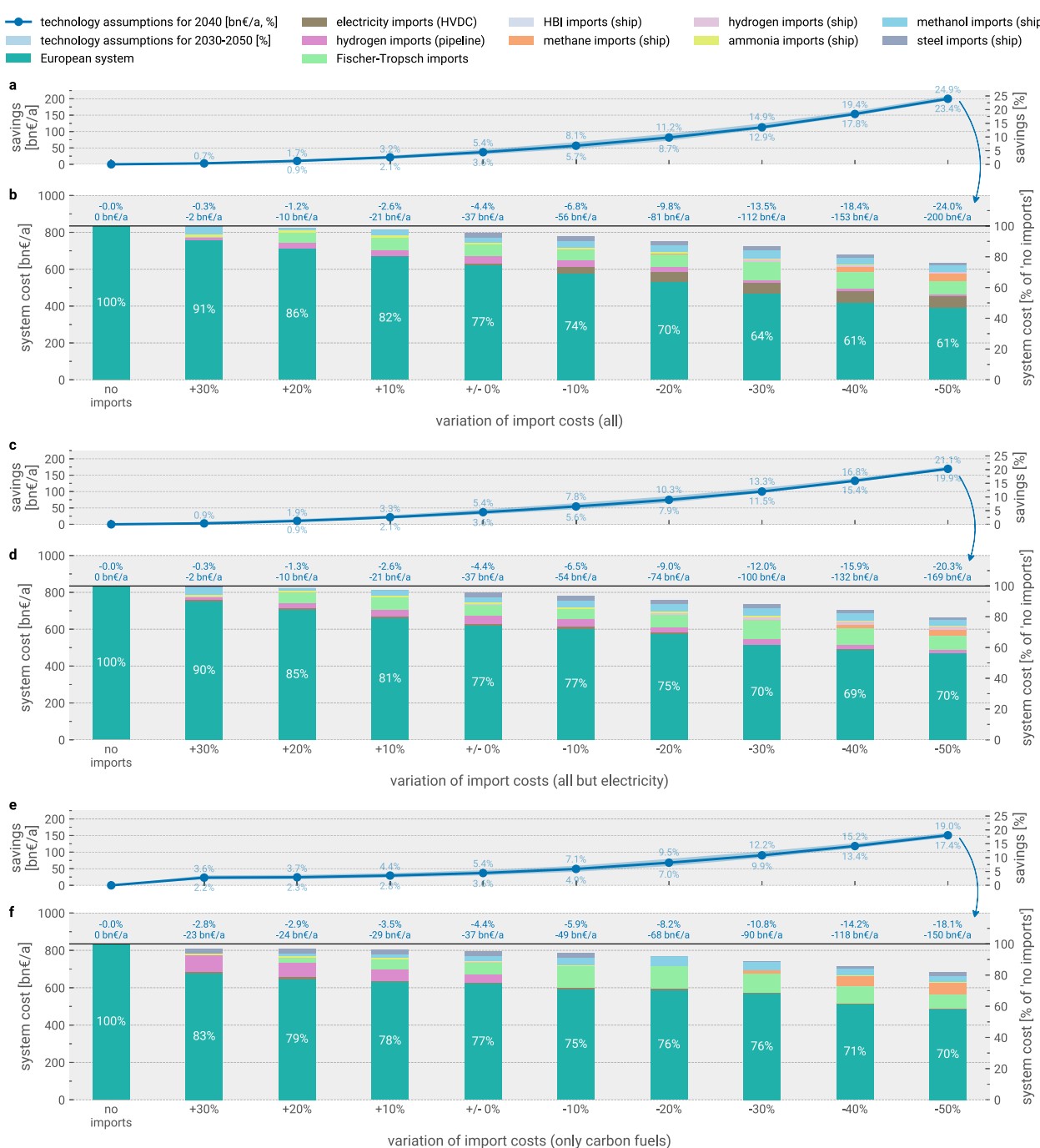

**Fig. 5 | Effect of import cost variations on system cost savings and import shares with all vectors allowed and unconstrained import volume.** In (**a**, **b**), indicated relative import cost changes are applied uniformly to all vectors. In (**c**, **d**), cost changes are applied uniformly to all vectors but electricity imports. In (**e**, **f**), cost changes are only applied to carbonaceous fuels (methane, methanol and Fischer-Tropsch). **a**, **c**, **e** show potential system cost savings compared to the scenario without imports. **b**, **d**, **f** show the share and composition of different import vectors in relation to total energy system costs. The information is shown both in absolute terms and relative terms compared to the scenario without imports. Ranges in (**a**, **c**, **e**) show cost savings (%) for technology assumption years 2030 and 2050, in addition to 2040. All shown scenarios allow relocation of crude steel and ammonia production within Europe. HBI = hot briquetted iron; HVDC = high-voltage direct current.

When the relative cost variation is only applied to carbon-based fuels (Fig. 5c), reflecting cost uncertainty in carbon provision, hydrogen imports are quickly displaced by Fischer-Tropsch and methanol imports with falling costs. Only when import costs rise by 20% do domestically produced liquid hydrocarbons – derived mainly from imported hydrogen – become more cost-effective than direct imports.

In all three cases of import cost variations, methane imports become relevant only with substantial cost reductions of 40%, replacing biogas and residual fossil gas consumption.

Overall, Fig. 5 also demonstrates that the impact of import cost variations on savings remains stable with more near-term (2030) and long-term (2050) technology assumptions.

**Table 1 | Examples for potential import cost increases or decreases**

| Cost factor | Absolute change | Unit | Relative change | Unit |
|---|---|---|---|---|
| Higher WACC of 12% abroad (e.g., high project risk) | + 48.8 | € MWh$^{-1}$ | + 38.0 | % |
| Higher WACC of 10% abroad (e.g., high project risk) | + 28.6 | € MWh$^{-1}$ | + 22.3 | % |
| Higher WACC of 8% abroad (e.g., high project risk) | + 9.2 | € MWh$^{-1}$ | + 7.2 | % |
| Higher direct air capture investment cost abroad (+200%) | + 44.3 | € MWh$^{-1}$ | + 34.5 | % |
| Higher direct air capture investment cost abroad (+100%) | + 22.3 | € MWh$^{-1}$ | + 17.4 | % |
| Higher direct air capture investment cost abroad (+50%) | + 11.2 | € MWh$^{-1}$ | + 8.7 | % |
| Higher electrolysis investment cost abroad (+50%) | + 17.3 | € MWh$^{-1}$ | + 13.5 | % |
| Lower WACC of 3% abroad (e.g., government guarantees) | − 33.4 | € MWh$^{-1}$ | − 26.0 | % |
| Lower WACC of 5% abroad (e.g., government guarantees) | − 17.5 | € MWh$^{-1}$ | − 13.6 | % |
| Lower WACC of 6% abroad (e.g., government guarantees) | − 8.9 | € MWh$^{-1}$ | − 6.9 | % |
| Lower electrolysis investment cost abroad (-50%) | − 18.4 | € MWh$^{-1}$ | − 14.3 | % |
| Sell excess curtailed electricity at 50 € MWh$^{-1}$ abroad | − 8.3 | € MWh$^{-1}$ | − 6.5 | % |
| Sell excess curtailed electricity at 30 € MWh$^{-1}$ abroad | − 4.6 | € MWh$^{-1}$ | − 3.6 | % |
| Sell excess curtailed electricity at 10 € MWh$^{-1}$ abroad | − 1.5 | € MWh$^{-1}$ | − 1.2 | % |
| Buy available biogenic or cycled $CO_2$ for 50 € t$^{-1}$ abroad | − 20.1 | € MWh$^{-1}$ | − 15.6 | % |
| Buy available biogenic or cycled $CO_2$ for 75 € t$^{-1}$ abroad | − 13.7 | € MWh$^{-1}$ | − 10.7 | % |
| Buy available biogenic or cycled $CO_2$ for 100 € t$^{-1}$ abroad | − 7.2 | € MWh$^{-1}$ | − 5.6 | % |
| Availability of geological hydrogen storage at 2.1 €/kWh (reduction by 95.5%) | − 5.1 | € MWh$^{-1}$ | − 4.0 | % |
| Sell power-to-X waste heat at 10 € MWh$^{-1}$ abroad | − 7.8 | € MWh$^{-1}$ | − 6.1 | % |
| Sell power-to-X waste heat at 5 € MWh$^{-1}$ abroad | − 6.9 | € MWh$^{-1}$ | − 5.4 | % |
| Highly flexible operation of Fischer-Tropsch synthesis (20% minimum part-load) | − 3.6 | € MWh$^{-1}$ | − 2.8 | % |

The table presents cost sensitivities in absolute and relative terms based on the supply chain for producing Fischer-Tropsch fuels in Southern Argentina for export to Europe (Portugal), using techno-economic assumptions for 2040. The reference fuel import cost for this case is 128.5 € MWh$^{-1}$. WACC = weighted average cost of capital.

## Attainable cost savings for varying import volumes

What is consistent for many scenarios with higher or lower import costs is the flat solution space around the respective cost-optimal import volumes. Increasing or decreasing the total amount of imports from the optimum barely affects system costs within ± 1000 TWh. This is illustrated in Fig. 6 and extended Supplementary Figs. 15–18, which show the system cost as a function of enforced import volumes and different import costs for hydrogen and its derivatives. A wide range of scenarios with import volumes below 4100 TWh (2300 TWh for 20% higher import costs, 5800 TWh for -20% lower import costs) have lower total energy system costs than the no-imports scenario. These ranges of import values are two to three times as large as the corresponding cost-optimal import volumes, which are indicated by the red markers in Fig. 6 and correspond to the bars previously shown in Fig. 5b. Naturally, the cost-optimal volume of imports increases as their costs decrease, but with noticeably varying slopes for system cost savings per unit of additional imported energy.

As we explore the effect of increasing import volumes on system costs, we find that already 56% (48–80% within ± 20% import costs) of the 4.4% (1.3–9.0%) total cost benefit can be achieved with the first 500 TWh of imports. This corresponds to 31% (25–49%) of the cost-optimal import volumes, highlighting the diminishing returns of large amounts of energy imports in Europe. The initial 1000 TWh realise 90% (80–100%) of the highest cost savings, for which primary crude steel and liquid carbonaceous fuel imports are prioritised, followed by ammonia and hydrogen and, subsequently, larger volumes of electricity beyond cost-optimal import levels. Once more than 5500 TWh (5000–8200 TWh) are imported, less than half the total system cost would be spent on domestic energy infrastructure.

As imports increase, there is a corresponding decrease in the need for domestic power-to-X (PtX) production and renewable capacities. A large share of the hydrogen, methanol, and primary steel production is outsourced from Europe, reducing the need for domestic wind and solar capacities. This trend is further characterised by the

displacement of biogas usage in favour of hydrogen imports around the 4000 TWh mark (3000–5000 TWh within ± 20% import costs) as demand for domestic $CO_2$ utilisation drops and methane use for power and heat provision is displaced by hydrogen. The increase in hydrogen imports results in the build-out of more hydrogen fuel cell combined heat and power (CHP) units for power and heat supply in district heating networks. Regarding electricity imports from the Middle East and North Africa (MENA) region, Fig. 6 reveals a mix of wind and solar power with some batteries to establish favourable feed-in profiles for the European system integration and higher utilisation rates for the long-distance HVDC links. For instance, for imports of 4000 TWh in Fig. 6, the capacity-weighted average utilisation rate was 85%. This is because a considerable share of the electricity import costs can be attributed to power transmission.

As import costs are varied, the composition of the domestic system and import mix for different import volumes is primarily similar (Supplementary Figs. 15–18). The main difference is a less prominent role for steel imports with higher import costs. What is furthermore noteworthy is that reducing import costs from − 30% to − 50% only marginally reduces domestic infrastructure costs, indicating largely saturated import potentials. Regarding available import options, the windows for cost savings are more limited if only subsets are available (Supplementary Fig. 19). However, up to an import volume of 1500 TWh for the central cost estimate, excluding electricity imports or constraining imports to methanol and Fischer-Tropsch fuels only, would not substantially diminish the cost-saving potential.

## Interactions of import strategy & domestic infrastructure

Across the range of import scenarios analysed, we find that the decision which import vectors are used strongly affects domestic energy infrastructure needs (Fig. 7).

In the fully self-sufficient European energy supply scenario, we see large PtX production within Europe to cover the demand for hydrogen and hydrogen derivatives in steelmaking, fertilisers, high-value

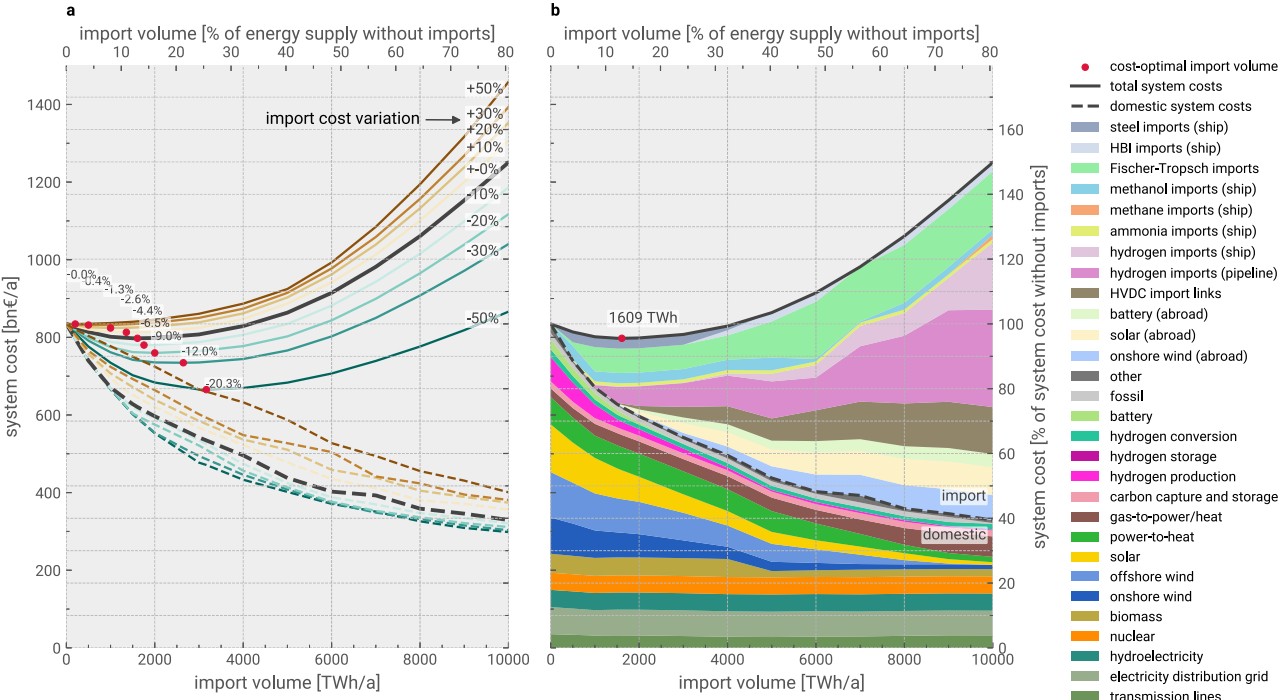

**Fig. 6 | Sensitivity of import volume on total system cost and composition.**
**a** Solid lines show the total system cost as a function of enforced import volumes for higher (brown scale) or lower (blue scale) import costs. The dashed lines indicate the corresponding shares of the domestic system cost. The red markers denote the maximum cost reductions and cost-optimal import volume for given import cost levels (extreme points of the curves). The cost alterations are uniformly applied to all import options but direct electricity imports. Steel is included in energy terms, applying 2.1 kWh kg$^{-1}$ as released by the oxidation of iron. **b** shows the composition of the total system cost as a function of enforced import volumes for the central import cost estimate. The dashed line splits the system costs into costs for imports and the domestic system. All shown scenarios use technology assumptions for 2040 and allow relocation of crude steel and ammonia production within Europe. Cost compositions for the alternative import cost scenarios are presented in Supplementary Figs. 15–18. Source data are provided as a Source Data file. HBI = hot briquetted iron; HVDC = high-voltage direct current.

chemicals, green shipping, and aviation fuels. Production sites are concentrated mainly in and around the North and Baltic Seas, using wind-based electrolysis, and some additional hubs in Southern Europe using solar-based electrolysis. Electricity grid reinforcements, representing around 50% of the current transmission capacity, are focused in Northwestern Europe, with numerous long-distance HVDC connections, but are broadly distributed overall.

With a total of 57 TWkm, the hydrogen pipeline build-out is smaller, mostly serving regional connections. For several reasons, it is also considerably smaller than the 204–306 TWkm observed previously in Neumann et al.[14] or the European Hydrogen Backbone reports[74] which envisioned a similar order of magnitude. Besides assumed full electrification of heavy-duty road transport and assuming low CO$_2$ transport costs from point sources to low-cost hydrogen sites[75], one reason is the considered relocation of crude steel and ammonia production to where hydrogen is cheap and abundant, reducing the need to transport hydrogen (Supplementary Fig. 6). Not considering relocation of primary crude steel and ammonia production would result in a slightly larger hydrogen network of 71 TWkm (Supplementary Fig. 26), while increasing system costs by 2.5 bn€$_{2020}$ a$^{-1}$ (0.3%) in the no-imports scenario. With permitted relocation of ammonia and crude steel production, primary steel production shifts to the British Isles and Spain, while ammonia production moves to the Nordic-Baltic region. Both sectors become more strongly localised, with individual regions capturing a market share surpassing 30%.

However, the main reason why hydrogen consumption is mainly concentrated in regions with low-cost production is that over 80% of the hydrogen is used to produce electrofuels for aviation, shipping, and chemical feedstocks, compared to about 10% for crude steel and

ammonia production. These liquid fuels can be transported at a lower cost to airports, ports, and industrial sites across Europe than hydrogen. Consequently, there is low impetus for transporting hydrogen directly, resulting in a hydrogen network that is much smaller than envisioned in the European Hydrogen Backbone[74].

Considering imports of renewable electricity, green hydrogen, and electrofuels substantially alters the magnitude of energy infrastructure in Europe. Imports displace much of the European power-to-X production capacities and, particularly, domestic solar energy generation in Southern Europe. Much of the remaining derivative fuel synthesis in Southern Spain uses imported hydrogen, assuming the delivery of captured CO$_2$ from other parts of Europe at low cost[75]. In contrast, the British Isles retain some domestic electrolyser capacities to produce synthetic fuels locally. Electricity imports of 131 TWh, compared to total imports of 1609 TWh, mainly enter from Tunisia at multiple nodes in Mallorca, Corsica, Sardinia, Sicily, and mainland Italy. This distribution facilitates grid integration without strong reinforcement needs in the Italian peninsula.

While the broad regions of domestic power grid reinforcements are not significantly affected by the import of electricity and other fuels, the volume of power grid expansion is reduced by 20%. The reduction in network infrastructure is even more pronounced with the hydrogen network; the hydrogen network size is reduced by 70% with many of the North and East European connections omitted. Compared to the self-sufficiency scenario, the cost-benefit of the hydrogen network shrinks from 3 bn€$_{2020}$ a$^{-1}$ (0.4%) to less than 1 bn€$_{2020}$ a$^{-1}$ (0.1%). This is caused by substantial amounts of hydrogen derivative imports or direct processing of imported hydrogen at the entry points, which diminishes the demand for hydrogen in Europe and, hence, the need to

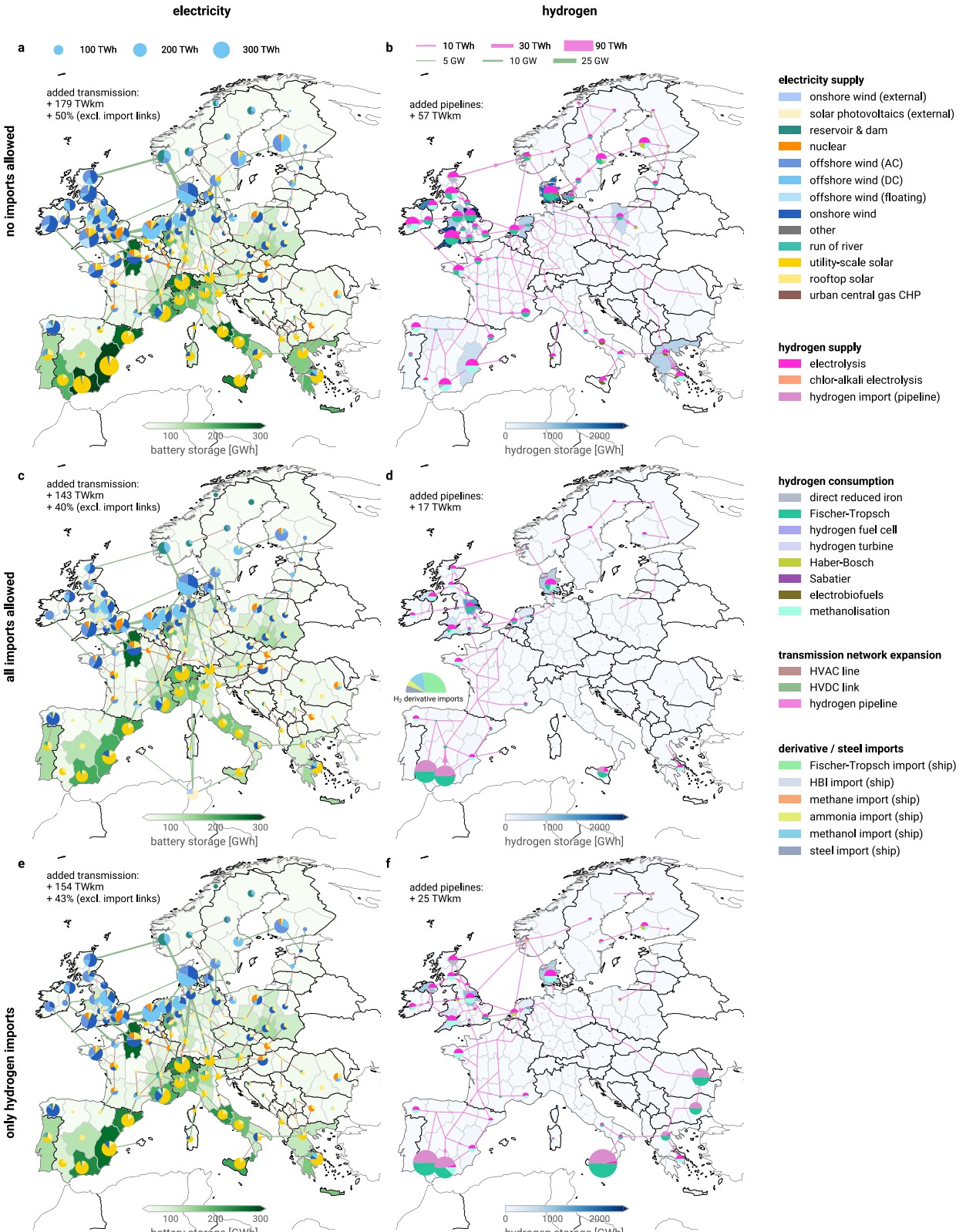

**Fig. 7 | Layout of European energy infrastructure for different import scenarios.** Panels (**a**, **c**, **e**) show the regional electricity supply mix (pies), added HVDC and HVAC transmission capacity (lines), and the siting of battery storage (choropleth). Panels (**b**, **d**, **f**) show the hydrogen supply (top half of pies) and consumption (bottom half of pies), net flow and direction of hydrogen in newly built pipelines (lines), and the siting of hydrogen storage subject to geological potentials (choropleth). Total volumes of transmission expansion are given in TWkm, which is the sum product of the capacity and length of individual connections. The half circle in the Bay of Biscay indicates the imports of hydrogen derivatives that are not spatially resolved: ammonia, steel, HBI, methanol, Fischer-Tropsch fuels. Hydrogen imports are shown at the entry points. All shown scenarios use technology assumptions for 2040, allow steel and ammonia relocation and have no import volume constraint (when available). Maps for more scenarios are included in Supplementary Figs. 26–29. Source data are provided as a Source Data file. Maps made with Natural Earth. HBI = hot briquetted iron; HVDC = high-voltage direct current; HVAC = high-voltage alternating current; AC = alternating current; DC = direct current; CHP = combined heat and power.

transport it. With further 10% cheaper carbonaceous fuel imports, the hydrogen network would then shrink to 9 TWkm (Supplementary Fig. 27).

Changes in the magnitude of domestic PtX production also affect Europe's backup capacity needs. Less PtX production means lower wind and solar capacities, which reduces the amount of available generation in dark wind lulls. Next to energy storage and demand-side management of electric vehicles and heat pumps, the operational flexibility of electrolysers and derivative fuel production yields significant benefits for integrating variable wind and solar feed-in and reduces reserve capacity requirements. In a theoretical scenario without imports, where all PtX processes must run inflexibly at full capacity, system costs rise by 8.8%. Between the main scenarios with and without imports, we observe that as imported fuels displace some flexible domestic power-to-X, domestic thermal backup capacities increase from 129 $GW_{el.}$ (no imports allowed) to 276 $GW_{el.}$ (all imports allowed). Instead of curtailing the domestic production of electrofuels, backup power plants need to be dispatched. Most of these power plants are CHPs fuelled by fossil gas, providing backup heat alongside backup power when electricity prices are high during winter (Supplementary Fig. 32). The resulting emissions are then compensated elsewhere in the system through biogenic carbon dioxide removal (Supplementary Fig. 25). Spatially, these are distributed across Central Europe, while batteries provide backup power in Southern Europe (Supplementary Fig. 33). The model leverages Europe's extensive power grid to widely distribute centralised backup power, even though, in reality, individual nations may prefer maintaining domestic reserve capacities.

A further observation is the high potential value of PtX waste heat and its role in siting fuel synthesis plants (Supplementary Figs. 26 and 28). Alongside the flexible operation of electrolysis to integrate variable wind and solar feed-in and the broad availability of industrial and biogenic carbon sources in Europe, waste heat usage in district heating networks is a potential revenue stream that could make electricity and hydrogen imports with subsequent domestic conversion more attractive relative to the direct import of derivative products. Our default assumption that only 25% of the waste heat can be utilised stems from potential challenges in co-locating PtX plants with district heating networks within the 115 model regions. If all waste heat could be leveraged, notable system cost savings of 20 bn€$_{2020}$ a$^{-1}$ (2.4%) could be achieved in the no-imports scenario compared to a scenario where waste heat is fully vented. To realise these benefits, Fischer-Tropsch and Haber-Bosch plants tend to be geographically distributed where space heating demand is high (e.g., Paris or Hamburg) (Supplementary Fig. 26), which increases hydrogen network build-out to 98 TWkm (+ 72%) compared to the reference scenario with 25% waste heat utilisation. This is not the case for methanolisation plants, which have lower waste heat potential.

### Causes of import cost variations and their effect

In Table 1, we present a breakdown of some potential causes for import cost variations compared to domestic supply chains relating to technology costs, financing costs, excess power and heat revenues, fuel synthesis flexibility, and the availability of geological hydrogen storage and alternative sources of $CO_2$.

For example, we show that a higher weighted average cost of capital (WACC) than the uniformly applied 7%, e.g., due to higher project financing risks, and lower WACC, e.g., due to the government-backing of projects, strongly affect import costs[76]. An increase or decrease by just one percentage point already alters the unit costs by around ± 7%. Likewise, technology cost variations abroad for electrolysers and DAC units have a strong influence. Biogenic $CO_2$ – or fossil $CO_2$ from industrial processes that is largely cycled between use and synthesis and, hence, not emitted to the atmosphere – can reduce the levelised fuel cost by 16% if it can be provided for 50 € t$^{-1}$.

By default, we assume islanded fuel synthesis sites, which causes curtailment rates of 8%. If surplus electricity production could be sold and absorbed by the local power grid in exporting regions, additional cost reductions could be achieved. Furthermore, process integration with waste heat usage and flexible operation can also reduce fuel cost by 3−6%. Import costs are also reduced by 4% where geological hydrogen storage is available by reducing the need for flexible power-to-X operation.

In contrast, the cost impact is low if the cheapest exporting region withdraws from the market. Within a cost premium of 10% in relation to the lowest cost exporting region, Chile, ten other regions could step in if these regions were unavailable for exports (Supplementary Fig. 9).

## Discussion

Our analysis offers insights into how renewable energy imports might reduce overall systems costs and interact with European energy infrastructure. Our results show that imports of green energy reduce the costs of a carbon-neutral European energy system by 37 bn€$_{2020}$ a$^{-1}$ (4.4%), noting, however, that the uncertainty range is considerable. While we find that some imports are beneficial within a ± 20% variation of import costs, system cost savings range between 1% and 10%. However, what is consistent within this range are the diminishing returns of energy imports for larger quantities, with peak cost savings below imports of 3000 TWh/a (equivalent to 90 Mt of hydrogen). We also find that there is value in pursuing some power-to-X production in Europe as a source of flexibility for wind and solar integration and as a potential source of waste heat for district heating networks. Another siting factor favouring European power-to-X is the wide availability of sustainable biogenic and industrial carbon sources, which helps reduce reliance on more costly direct air capture.

With reference to the meta-study by Genge et al.[55], our import costs to Europe for lowest-cost to median-level-cost exporters mostly conform to the review's interquartile ranges for different carriers and technology assumptions for 2030 and 2050. Some higher costs, e.g., for ammonia, can be attributed to our updated electrolyser cost assumptions (1100/950/800€ $kW_{el}^{-1}$ in 2030/2040/2050), which reflect recent market developments[77]. Also for crude steel imports, our central cost estimate of 531 € t$^{-1}$ for the lowest-cost exporter is positioned between studies with lower[50] and higher[67] cost estimates. Among other studies investigating the relationship between energy imports and the European energy system, several analyses report lower import shares in the range of 10−20% of total hydrogen supply[58,61,62]. For instance, Kountouris et al.[61] see limited hydrogen imports of 182 TWh a$^{-1}$ from the Maghreb region and Ukraine despite favourable import costs of 33 € $MWh_{H_2}^{-1}$. Conversely, Wetzel et al.[59] find higher import shares of 53% for methane and 43% for hydrogen. The latter closely aligns with our 49% import share for hydrogen. Wetzel et al.[59] find that imports reduce system costs by 2.8%, which is also comparable to our 2.4% system cost reduction when only direct hydrogen imports are considered. The most pronounced import dependency we found in Schmitz et al.[42], with import shares beyond 90% for Germany. Results in Kountouris et al.[61] further substantiate our finding that derivative imports and demand relocation could diminish hydrogen network benefits.

Several limitations of our study should be noted. First, the optimisation results represent a long-term equilibrium that disregards potential transition-related infrastructure lock-ins or mid-term ramp-up constraints of export capacities or domestic infrastructure development. The development speed of key technologies is also uncertain and could affect cost-optimal infrastructure and import strategies. A further limitation is that our cost-based analysis of imports, which best reflects long-term bilateral purchase agreements, neglects price impacts of intensifying global competition for green fuel imports and exports[56]. Besides unclear market developments, local challenges in exporting regions, such as public acceptance for export-oriented

energy projects[78] and potential water scarcity[41,79] to produce large amounts of hydrogen in renewable-rich but arid regions are not addressed. We also do not assess potential impacts on the regional economy and local employment effects within Europe, as some ammonia production and steel manufacturing relocates in the model. Furthermore, the model's lack of spatial resolution for $CO_2$ means that carbonaceous hydrogen derivatives are sited where $H_2$ is cheapest, implicitly assuming that the $CO_2$ from biogenic or industrial sources can be transported there. However, such required $CO_2$ pipelines could be built at relatively low additional system cost[75]. In the context of carbon management, more lenient assumptions on sustainable biofuel potentials, allowed levels of geological carbon sequestration, or plastic landfill could alter the results, shifting the system away from synthetic electrofuels towards more fossil fuel use with carbon capture or carbon dioxide removal[75,80].

Overall, we find that the import vectors used strongly affect domestic infrastructure needs. For example, only a smaller hydrogen network would be required if hydrogen derivatives were largely imported and the domestic ammonia and crude steel industry is allowed to relocate. We also identify higher electricity backup requirements in the absence of large power-to-X flexibilities. These findings underscore the importance of coordination between energy import strategies and infrastructure policy decisions. Our results present a quantitative basis for further discussions about the trade-offs between system cost, carbon neutrality, public acceptance, energy security, infrastructure buildout, and imports.

The small differences in cost observed between some scenarios are particularly relevant because factors other than pure costs, which are not reflected in our infrastructure optimisation model, might then drive import strategies. To some, the relatively limited cost benefit of imports and offshoring of industrial production may speak against imports. Concerns about energy security could motivate more domestic supply and diversified imports. For instance, shipborne imports of hydrogen derivatives could be preferred to reduce pipeline lock-in and to mitigate the risks of sudden supply disruptions and the exercise of market power. From a practical perspective, it may also be more appealing to focus on carriers that are already globally traded commodities and to prefer infrastructure offering quick deployment.

Policymakers in Europe might prefer alternative systems featuring, for instance, lower domestic infrastructure requirements, reuse of existing infrastructure, lower technology risk, and reduced land usage for broader public support than the most cost-effective solution. Moreover, policies favoring local energy supply chains and importing intermediary products like sponge iron could be favoured to preserve European jobs while outsourcing only the most energy-intensive processes. However, in shifting potential land use and infrastructure conflicts to abroad, where population densities are often lower, potential exporting countries would need to weigh the prospect of economic development against internal social and environmental concerns, particularly in countries with a history of colonial exploitation[81]. Ultimately, Europe's energy strategy would likely seek to balance cost savings from green energy and material imports with broader concerns like geopolitics, economic development, public opinion, and the willingness of potential exporting countries in order to ensure a swift, secure, and sustainable energy future. Our research shows that there is manoeuvreing space around Europe's energy import strategies to accommodate such non-cost concerns.

## Methods

### Overview of European energy system model PyPSA-Eur

For our analysis, we use the European sector-coupled high-resolution energy system model PyPSA-Eur[82] (derivative of v0.13.0) based on the open-source modelling framework PyPSA[83] (Python for Power System Analysis), covering the energy demands of all sectors including electricity, heat, transport, industry, agriculture, as well as non-energy

feedstock demands, international shipping, and aviation. An overview of considered supply, consumption, and balancing technologies per carrier is shown in Supplementary Fig. 2.

The model simultaneously optimises spatially explicit investments and the operation of generation, storage, conversion, and transmission assets to minimise total system costs in a single linear optimisation problem, which assumes perfect operational foresight and is solved with Gurobi (v11.0.1)[84]. To manage computational complexity, no pathways with multiple investment periods are calculated, but overnight scenarios targeting net-zero $CO_2$ emissions. The capacity expansion is based on technology cost and efficiency assumptions for 2040 (see 'Data availability'), acknowledging that much of the required infrastructure must be constructed well before reaching net-zero emissions. Figures 2 and 5 and Supplementary Fig. 29 feature additional scenarios using technology assumptions for 2030 and 2050.

Existing hydro-electric power plants[85] are included, as well as nuclear power plants built after 1990 or currently under construction according to Global Energy Monitor's Global Nuclear Plant Tracker (52 GW total of 106 GW in current operation)[86]. While hydroelectricity is assumed to be non-extendable due to geographic constraints, additional nuclear capacities can be expanded where cost-effective. We assume the existing nuclear fleet is operated inflexibly and apply country-specific historical availability factors from 2021 to 2023[87].

Temporally, the model is solved with an uninterrupted 4 h equivalent resolution for a single year (2190 time steps), using a segmentation clustering approach implemented in the tsam toolbox on all time-varying data[88]. While weather variations between years are not considered for computational reasons, the chosen weather year 2013 is representative in terms of wind and solar availability and heat demand[89]. Some demands are associated with a time-varying profiles (e.g., residential/services electricity, electric vehicles, and heating demand) based on travel patterns or ambient weather conditions, while the other exogenous demands are assumed to be time-constant (e.g., kerosene, naphtha, methanol, ammonia, and industry electricity).

Spatially, the model resolves 115 European regions[90], covering the European Union, the United Kingdom, Norway, Switzerland, and the Balkan countries without Malta and Cyprus. For computational reasons, only electricity, heat, and hydrogen are modelled at high spatial resolution, while oil, methanol, methane, ammonia, and carbon dioxide are treated as easily transportable without spatial constraints. Of the total final energy and non-energy demand (Supplementary Fig. 5), only some demands are spatially fixed (Supplementary Fig. 4). These include electricity for residential, industry, services, and agriculture; heat; electric vehicles; solid biomass for industry; naphtha/methanol feedstocks; and hydrogen for crude steel and ammonia production unless these industries can relocate.

Most other hydrogen demands are spatially variable. Only a small demand of 5 TWh $a^{-1}$ in the chemicals industry (excluding liquid feedstocks) remains, which is offset by spatially fixed hydrogen production of around 10 TWh $a^{-1}$ from chlor-alkali electrolysis for chlorine production. High-temperature industrial heat is supplied by methane, shipping and aviation use carbonaceous fuels, and land transport is fully electrified. In district heating and the power sector, backup hydrogen capacities are endogenously sized and sited just as the production capacities of hydrogen derivatives (Fischer-Tropsch, methane, methanol), which account for more than 80% of the hydrogen consumption. Since the model optimises the siting and operation of these fuel synthesis plants and electrolysers, many demands are spatially variable (e.g., electricity demand for electrolysers or hydrogen demand for methanolisation). Existing hydrogen production capacities from fossil gas reforming are not considered, as they are expected to reach the end of life over the model horizon.

A mathematical description of PyPSA-Eur can be found in Supplementary Note 1, adapted from Neumann et al.[14]

## Gas and electricity network modelling

Networks are considered for electricity, methane, and hydrogen transport. Existing gas pipelines taken from SciGRID_gas[91], can be repurposed to hydrogen in addition to new hydrogen pipelines[14]. Data on the gas transmission network is further supplemented by the locations of fossil gas extraction sites and gas storage facilities based on SciGRID_gas[91], as well as investment costs and capacities of LNG terminals in operation or under construction from Global Energy Monitor's Europe Gas Tracker[92]. Geological potentials for hydrogen storage are taken from Caglayan et al.[93], restricting where this low-cost storage option is available. In modelling gas and hydrogen flows, we incorporate electricity demands for compression of 1% and 2% per 1000km of the transported energy, respectively[94]. Existing high-voltage grid data is taken from OpenStreetMap[95]. For HVDC transmission lines, we assume 2% static losses at the substations and additional losses of 3% per 1000 km. The losses of high-voltage AC transmission lines are estimated using the piecewise linear approximation from Neumann et al.[96], in addition to applying linearised power flow equations[97]. Up to a maximum capacity increase of 30%, we consider dynamic line rating (DLR), leveraging the cooling effect of wind and low ambient temperatures to exploit existing transmission assets fully[98]. To approximate $N-1$ resilience, transmission lines may only be used up to 70% of their rated dynamic capacity[99]. To prevent excessive expansion of single connections, power transmission reinforcements between two regions are limited to 15 GW, while an upper limit of 50.7 GW is placed on hydrogen pipelines, which corresponds to three 48-inch pipelines[94].

## Wind and solar potentials

Renewable potentials and time series for wind and solar electricity generation are calculated with atlite[100], considering land eligibility constraints like nature reserves, excluded land use types, topography, bathymetry, and distance criteria to settlements. Given low onshore wind expansion in many European countries in recent years[101], a deployment density of $1.5\,MW\,km^{-2}$ is assumed for eligible land for onshore wind expansion[102]. For reference, this assumption leads to an onshore wind potential for Germany of 244 GW. The temporal renewable generation potential for the available area is then assessed based on reanalysis weather data, ERA5[103], and satellite observations for solar irradiation, SARAH-3[104], in combination with standard solar panel and wind turbine models provided by atlite.

## Biomass potentials

Biomass potentials are restricted to residues from agriculture and forestry, as well as waste and manure, based on the regional medium potentials specified for 2050 in the JRC-ENSPRESO database[105]. Continued use of energy crops or biomass imports are not considered. The finite sustainable biomass resource can be employed for low-temperature heat provision in industrial applications, biomass boilers, and CHPs, and (electro-)biofuel production for use in aviation, shipping, and the chemicals industry. In addition, we allow biogas upgrading, including capturing the $CO_2$ contained in biogas, which unlocks all considered uses of regular methane (Supplementary Fig. 2). The total assumed bioenergy potentials are 1372 TWh, which splits into 358 TWh/a for biogas and 1014 TWh/a for solid biomass. The total carbon content corresponds to $605\,Mt_{CO_2}\,a^{-1}$, which is not fully available as a feedstock for fuel synthesis or sequestration for negative emissions due to imperfect capture rates of up to 90%. Biogenic $CO_2$ can be captured from biogas upgrading, biomass CHPs and biomass-based low-temperature heat provision in industrial use, if the added cost of carbon capture is economically viable.

## Carbon management

The carbon management features of the model trace the carbon cycles through various conversion stages: industrial emissions, biomass and gas combustion, carbon capture in numerous applications, direct air capture, intermediate storage, electrofuels, recycling, landfill or long-term sequestration. The overall annual sequestration of $CO_2$ is limited to $200\,Mt_{CO_2}\,a^{-1}$, similar to the $250\,Mt_{CO_2}\,a^{-1}$ highlighted in the European Commission's carbon management strategy[70]. This number allows for sequestering the industry's unabated fossil emissions (e.g., in the cement industry) while minimising reliance on carbon removal technologies. A carbon dioxide network topology is not co-optimised since $CO_2$ is not spatially resolved. This means that the location of biogenic or industrial point sources of $CO_2$ is not a siting factor that this model version considers for PtX processes, implicitly assuming that the $CO_2$ would be transported there at low cost[75,106].

## Transport sector fuel assumptions

While the shipping sector is assumed to use methanol as fuel, given its high technology-readiness level compared to hydrogen or ammonia[107], land-based transport, including heavy-duty vehicles, is fully electrified in the presented scenarios[108]. Aviation can use green kerosene derived from Fischer-Tropsch fuels or methanol, owing to the lower technology readiness levels of fuel cell or battery-electric aircraft[107]. Alternative uses for methanol and Fischer-Tropsch fuels extend beyond transport, including power-to-methanol[73], diesel for agriculture machinery and as feedstock for high-value chemicals.

## Technical constraints of synthetic fuel production

We consider potential flexibility restrictions in the synthesis processes to obtain more realistic operational patterns of green electrofuel synthesis plants. We apply a minimum part load of 20% for methanolisation and 50% for methanation and Fischer-Tropsch synthesis[109-112]. The assumed lower operational flexibility is a potential disadvantage of Fischer-Tropsch over methanol synthesis, where theses fuels compete. These 'green' options then compete with 'blue' and 'grey' options, such as steam methane reforming of fossil gas with or without carbon capture for hydrogen (Supplementary Fig. 2). Some carriers also feature a biogenic production route (e.g., methane and oil).

## Heating sector modelling and PtX waste heat

Heating supply technologies like heat pumps, electric boilers, gas boilers, and combined heat and power (CHP) plants are endogenously optimised separately for decentral use and central district heating. District heating shares of demand are exogenously set to a maximum of 60% of the total urban heat demand with sufficiently high population density. Besides the options for long-duration thermal energy storage, district heating networks can further be supplemented with waste heat from various power-to-X processes: electrolysis, methanation, ammonia synthesis, and Fischer-Tropsch fuel synthesis. Because the thermal discharge from the methanol synthesis is primarily used to distillate the methanol-water output mix[73], its waste heat potential is not considered for district heat. Here, we assume a utilisable share of waste heat of 25%, considering that within the 115 regions, only a fraction of fuel synthesis plants might be connected to district heating systems. In further sensitivity analyses, we explore the effect of no or full waste heat utilisation.

## Backup heat and power options

The model includes a variety of options for providing backup power and heating in periods of low renewable generation and high demand (Supplementary Fig. 2). Backup power options include hydrogen, gas and methanol turbines. Backup heat options include gas boilers and resistive heaters. For combined backup heat and power, we consider biomass, hydrogen, and gas CHPs. Furthermore, flexible demands like electric vehicles, heat pumps and fuel synthesis units, as well as batteries and thermal storage in district heating, can be utilised to reduce the need for backup capacities.

## Industry relocation modelling for crude steel and ammonia production

Unless indicated otherwise, all scenarios also allow the model to relocate the crude steel and ammonia industry within Europe endogenously. This allows the best sites within Europe to compete with outsourced production abroad. While this captures some of the most energy-intensive industry sectors, other sectors, like concrete and alumina production, are not considered for relocation.

Without relocation of crude steel and ammonia production allowed, the production volumes of primary crude steel, by direct iron reduction (DRI) and electric arc furnace (EAF), and ammonia for fertilisers, by Haber-Bosch synthesis, are spatially fixed. This results in exogenous hydrogen demand per region. Total production volumes are based on current levels[113,114]. For the spatial distribution, we use data on the existing integrated steelworks listed in Global Energy Monitor's Global Steel Plant Tracker[115] and manually collected data on the location and size of ammonia plants in Europe.

With the relocation of crude steel and ammonia production allowed, the model endogenously chooses the regional production volumes of primary crude steel, HBI, and ammonia, subject to the availability of cheap hydrogen. Thereby, the regional capacities and operation of Haber-Bosch, DRI, and EAF plants are co-optimised with the rest of the system, similar to the siting of Fischer-Tropsch or methanolisation plants. For DRI and EAF, investment costs and specific requirements for fuels and iron ore are taken from the Steel Sector Transition Strategy Model (ST-STSM) of the Mission Possible Partnership[116,117]. and assume steel can be stored and transported without constraints within Europe.

For both cases, we assume a rise in the steel recycling rate from 40% today to 70% in our carbon-neutral scenarios[118]. We assume that the electric arc furnaces for secondary steel remain, in proportion, at current locations and do not relocate.

A limitation of the relocation modelling of crude steel and ammonia production is that it only considers the cost of energy in the siting of these industries. Other factors, such as impacts on regional economies and local jobs, integration with other production processes, or availability of other existing infrastructure, are not considered, largely due to a lack of data. The resulting relocation patterns should therefore be interpreted with caution, as they might underestimate total relocation costs and frictions. We allow domestic relocation, nevertheless, in most scenarios, as it would be inconsistent to allow crude steel and ammonia imports from abroad while preventing relocation within Europe.

## Import supply chain modelling with TRACE

The European energy system model is extended with data from the TRACE model (derivative of v1.1) used in Hampp et al.[54] to assess the unit costs of different vectors for importing green energy and material to entry points in Europe from various world regions. For consistency with the European model, the techno-economic assumptions were aligned, using the same values for 2040 (plus 2030 / 2050 in Fig. 2 and 5 and Supplementary Fig. 29 and a uniform weighted average cost of capital (WACC) of 7%[119]. As possible import vectors, we consider electricity by transmission lines, hydrogen as a gas by pipelines and as a liquid by ship, methane as a liquid by ship, liquid ammonia, crude steel and HBI, methanol and Fischer-Tropsch fuels by ship. Liquid organic hydrogen carriers (LOHC) are not considered as export vectors due to their lower technology readiness level (TRL) compared to other vectors[1].

Our selection of 53 potential exporting regions broadly comprises countries with favourable wind and solar resources and large enough potential for substantial exports above 500 TWh $a^{-1}$ in addition to domestic consumption. We exclude some countries due to political instability (e.g., Sudan, Somalia, Yemen), using a Fragile States Index[120] value of 100 as a threshold, or due to severe imposed sanctions (e.g.,

Russia, Iran, Iraq), following the EU Sanctions Map[121]. Landlocked countries without access to seaports or realistic pipeline connections are excluded. For landlocked regions within pipeline reach, we only exclude shipborne vectors. Some large countries are split into multiple subregions for a more differentiated view (e.g., USA, Argentina, Brazil, and China). The resulting regions are marked in Fig. 1A.

To determine the levelised cost of energy for exports, the methodology first assesses the regional potentials for solar, onshore, and offshore wind energy. These potentials and time series are calculated using atlite[100], applying similar land eligibility constraints as in PyPSA-Eur (but using other datasets with global coverage) and applying the same wind turbine and solar panel models to ERA5[103] weather data for 2013 in eligible regions. Since TRACE evaluates whole regions without further transmission network resolution, the renewable potentials and profiles within a region are split into different resource classes to reduce smoothing effects. We consider 30 classes each for onshore wind and solar, and 10 for offshore wind, where applicable. Based on these calculations, levelised cost of electricity (LCOE) curves can be determined for each region. A selection of LCOE curves is shown in Supplementary Fig. 22.

In the next step, potentials are reduced by the projected future local energy demand, starting with the lowest LCOE resource classes. With this approach, domestic consumption is prioritised and supplied by the regions' best renewable resources, even though we do not model the energy transition in exporting regions in detail. To create the demand projections, we use the GEGIS[122] tool, which utilises machine learning on historical time series, weather data, and macroeconomic factors to create artificial electricity demand time series based on population and gross domestic product (GDP) growth scenarios following the SSP2 scenario of the Shared Socioeconomic Pathways[123]. From these time series, we take the annual total and increase it by a factor of two to account for further electrification of other sectors, which the GEGIS tool does not consider.

The remaining wind and solar electricity supply can then be used to produce the specific energy or material vector according to the flow chart of conversion pathways shown in Supplementary Fig. 1. Considered technologies include water electrolysis for $H_2$, direct air capture (DAC) for $CO_2$, synthesis of methane, methanol, ammonia or Fischer-Tropsch fuels from $H_2$ with $CO_2$ or $N_2$, and $H_2$ direct iron reduction (DRI) for sponge iron with subsequent processing to green steel in electric arc furnaces (EAF) from iron ore priced at 97.7 € $t^{-1}$[116]. Other $CO_2$ sources than DAC are not considered in the exporting regions. Furthermore, while batteries and hydrogen storage in steel tanks are considered, underground hydrogen storage is excluded due to the uncertain availability of salt caverns in many potential exporting regions[124,125]. We also assume that the energy supply chains dedicated to exports will be islanded from the rest of the local energy system, i.e., that curtailed electricity or waste heat could not be used locally.

For each vector, an annual reference export demand of 500 $TWh_{LHV}$ or 100 Mt of crude steel and HBI is assumed, mirroring large-scale energy and material infrastructures and export volumes, corresponding to approximately 40% of current European LNG imports[126] and 66% of European steel production[127]. Transport distances are calculated between the exporting regions and the twelve representative European import locations using the searoute Python tool[128] for shipborne vectors or crow-fly distances for pipeline or HVDC connections, and modified by a mode-specific detour factor. The chosen representative import locations are based on large ports and LNG terminals in the United Kingdom, the Netherlands, Poland, Greece, Italy, Spain, and Portugal, as well as pipeline entry points in Slovakia, Greece, Italy, and Spain. All energy supply chains are assumed to consume their energy vector as fuel for transport to Europe, except for HBI and crude steel, which use externally bought green methanol as shipping fuel. The capital costs of the ships and pipelines are also included, following the methodology of Hampp et al.[54].

For each combination of carrier, exporter, and importer, a linear capacity expansion optimisation is performed to determine cost-optimal investments and the operation of generation, conversion, storage, and transport capacities for all intermediary products to deliver 500 TWh a$^{-1}$ (or 100 Mt a$^{-1}$ for materials) of the final carrier to Europe. Dividing the total annual system costs by the targeted annual export volume yields the levelised cost of energy or material as seen by the European entry point. To match the multi-hourly resolution used for the European model, the TRACE model was configured to use a 3-hourly resolution for 2013, resulting in similar balancing requirements. Considering the reference export volume of 500 TWh a$^{-1}$ (or 100 Mt a$^{-1}$ for materials), the resulting levelised cost curves of imports for different import vectors and exporting regions are presented for the respective lowest-cost entry point to Europe in Supplementary Figs. 7–10. The curves show the varying cost composition of the country-carrier pairs. In this step, each import vector combination of carrier, exporter, and importer is optimised separately. Further constraints, like constraints on total export volumes per country, are imposed in the coupling to the European model.

A mathematical description of TRACE can be found in Section S3 in Hampp et al.[54]

### Coupling of import options to European model

The resulting levelised unit cost for each combination of carrier, exporter, and reference importer is then used as an exogenous input to the European model. For each candidate entry point in the 115 European model regions, we match the closest reference import location from TRACE and add the corresponding import cost curve as a supply option (Supplementary Figs. 7–10). Moreover, we limit energy exports from any one exporting region to Europe for the sum of all carriers to 500 TWh a$^{-1}$. This is to both prevent a single country from dominating the import mix and be consistent with the target export volume assumed in TRACE. Beyond that, the decision about the origin, destination, vector, volume, and timing of imports is largely endogenous to PyPSA-Eur.

However, imports may be further restricted by the expansion of domestic import infrastructure. For each vector, we identify locations where the respective carrier may enter the European energy system by considering where LNG terminals and cross-continental pipelines are located (Fig. 1b). For hydrogen imports by pipeline, imports must be near-constant, varying between 90–100% of peak imports, aligning with the high pipeline utilisation rates observed in the TRACE model. For methane imports by ship, existing LNG terminals reported in Global Energy Monitor's Europe Gas Tracker[92] can be used. For hydrogen by ship, new terminals can be built in regions where LNG terminals exist. To ensure regional diversity in potential gas and hydrogen imports and avoid vulnerable singular import locations, we allow the expansion beyond the reported capacities only up to a factor of 2.5, taking the median value of reported investment costs for LNG terminals[129]. A premium of 20% is added for hydrogen import terminals due to the lack of practical experience with them. For electricity, the capacity and operational patterns of the HVDC links can be endogenously optimised. Imports for carbonaceous fuels, ammonia, HBI, and steel are not spatially allocated to specific ports, given their low transport costs relative to value. Port capacities are assumed unconstrained since these commodities, particularly carbonaceous fuels, are comparable to the large fossil oil volumes currently handled at European ports.

Further conversion of imported fuels is also possible once they have arrived in Europe, e.g., hydrogen could be used to synthesise carbon-based fuels, ammonia could be cracked to hydrogen, methane could be reformed to hydrogen, and methane or methanol could be combusted for power generation. However, conversion losses can make it less attractive economically to use a high-value hydrogen derivative merely as a transport and storage vessel, only to reconvert it back to hydrogen or electricity.

The supply chain of electricity imports is endogenously optimised with the rest of the European system rather than using a constant levelised cost of electricity for each export region. This is because, owing to the greater challenge of storing electricity, the hourly variability of wind and solar electricity leads to higher price variability than hydrogen and its derivatives, and the intake needs to be more closely coordinated with the European power grid. The endogenous optimisation comprises wind and solar capacities, batteries and hydrogen storage in steel tanks, and the size and operation of HVDC link connections to Europe based on the renewable capacity factor time series as illustrated in Fig. 1b. Europe's connection options with exporting regions are confined to the 4% nearest regions, with additional ultra-long distance connection options to Ireland, Cornwall, and Brittany following the vision of the Xlinks project between Morocco and the United Kingdom[30]. Connections through Russia or Belarus are excluded. In addition to excluded entry points, some connections from Central Asia are affected by additional detours beyond the regularly applied detour factor of 125% of the as-the-crow-flies distance. Similar to intra-European HVDC transmission, a 3% loss per 1000 km and a 2% converter station loss are assumed.

Finally, we note that all mass-energy conversion is based on the lower heating value (LHV). To present energy and material imports in a common unit, the embodied energy in steel is approximated with the 2.1 kWh kg$^{-1}$ released in iron oxide reduction, i.e., energy released by combustion[130]. All currency values are given in €$_{2020}$.

## Data availability

A dataset of the model results is available on Zenodo under https://doi.org/10.5281/zenodo.14872184[131]. Data on techno-economic assumptions for years can be found at https://github.com/PyPSA/technology-data/releases/tag/import-benefits-v2 and has been archived on Zenodo under https://doi.org/10.5281/zenodo.14872325[132]. Source data are provided with this paper.

## Code availability

The code to reproduce the experiments is available at https://github.com/fneum/import-benefits/tree/v3.0.1 (v3.0.1), which uses a derivative of TRACE v1.1 and PyPSA-Eur v0.13.0. The code has been archived on Zenodo under https://doi.org/10.5281/zenodo.14872325[132]. We also refer to the documentation of PyPSA-Eur at https://pypsa-eur.readthedocs.io/en/v0.13.0-docs-fix for more details.

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

## Acknowledgements

J.H. gratefully acknowledges funding from the Kopernikus-Ariadne project (FKZ 03SFK5A and 03SFK5A0-2) by the German Federal Ministry of Education and Research (*Bundesministerium für Bildung und Forschung, BMBF*).

## Author contributions

F.N., J.H., and T.B. jointly conceived the study, designed the methodology, and carried out the investigation. F.N. and J.H. jointly developed the code. F.N. curated data, created visualisations, and drafted the manuscript. J.H. and T.B. reviewed and edited the manuscript.

## Funding

## Competing interests

The authors declare no competing interests.
