## [Transparent Peer Review file · Nature Communications]

Green energy and steel imports reduce Europe's net-zero infrastructure needs

Corresponding Author: Dr Fabian Neumann

Version 0:

Reviewer comments:

Reviewer #1

(Remarks to the Author)
Overview

The article assesses the important topic of green energy imports in the European context by coupling a spatially detailed energy system optimisation model with a supply chain model.

The main focus was to compare a self-sufficient net-zero European energy system against different scenarios where green energy imports are allowed. The scenarios have the following characteristics:

- Roughly 8 different import commodities are allowed, ranging from simple carriers like electricity and hydrogen, to more complex end-products such as Fischer-Tropsch fuels and green steel.
- 14 different exporting countries are considered, with the caveat that they must first satisfy their expected internal demands before exporting these products.
- 2030 prices for technologies are considered.
- The model uses a green field approach, and even allows heavy industries like steel to relocate within Europe to achieve optimal results.

The main result is that energy imports can marginally decrease the cost of a net-zero European system by up to about 5%. Importantly, the optimization model's preference for using steel and methanol imports appear more robust to price variations than hydrogen and Fischer-Tropsch fuel imports. However, there is a broad range of uncertainty in these results, particularly when it comes to price variations and import availability. In particular, the authors highlight the need to coordinate import strategies with infrastructure decisions (which intuitively makes sense).

These results will be important to both modelers and policy-makers in the field of energy planning, particularly in light of recent import targets set by the EU.

I would like to express some concerns about the study, both in terms of methods and framing. I believe that in its current form, the paper is not suitable for publication, and three major issues first need to be thoroughly resolved (numbered 1-3). In addition, I provide a list of minor issues that ought to be addressed before publication.

Major comments

1. Robustness and limited sensitivity analysis

The authors put emphasis in highlighting the robustness of their results in their introduction and conclusion (P2-L13, P5-L87). However, I do not believe that the paper's methods provide enough uncertainty analysis to justify such an assertion, especially considering that the core result is a low variation of total system costs of 5%.

Robustness usually implies quantifying the range in which a strategy performs adequately (either ex-post or within the model). The authors mainly address this by varying which imports are available, or by evenly varying the price of all imports (between +20/-30%). However, this range is not justified with a historical counterfactual. Considering that the increase of

annual natural gas prices by 200% is one of the motivating reasons for the policies driving this article, and that oil prices can spike by more than 100% between years, I struggle to see how such a low homogeneous increase addresses this sufficiently.

Other major assumptions of the paper (complete freedom to re-locate industry, and lack of competition between Europe and other world regions for fuel imports) also make me doubt the robustness of the analysis. Are you sure that these would not have significant impact on the results of your study? Similarly, in the S.I. you state that if some importing countries were not willing to export (P1-L27), costs could raise by 10%, yet this is not included in your sensitivity analysis.

I believe that one of two adjustments is needed: Either the paper needs to re-frame the findings to state the limitations more clearly in the main text, or the authors need to perform additional sensitivity analysis for the problematic dimensions outlined above.

2. Justification of assumptions

The methods section lists a wide range of assumptions made within the study, but in many cases a proper justification for them is not provided. Although I fully believe that the authors had proper reasons, it is not possible as a reader to discern why they are present or if the choices made are justified. I think the authors should provide justification for the following assumptions somewhere in the text:

- List of countries (P2-L85): only a few list of importing countries are included (14, for what I gather from Fig 1). But why these specifically? If this is based on analysis of policy texts, or just because they are the only countries evaluated by TRACE?
- Limitation of wind expansion (P10-L83): you state a limit of 1.5 MW/km² for onshore wind expansion, but your citation (Our World in Data) does not directly justify this figure. How was this value obtained, and what is its impact on your results?
- Fuels in hard to abate sectors (P10-L106): several fuel combinations are allowed for these technologies, and while they make broad sense to someone in the field of energy systems, they lack any sort of citation in the text. Ditto for minimum part load constraints for Fischer-Tropsch, methanolisation and methanation facilities.
- N-1 resilience (P11-L5): you state that a 70% limit on transmission line utilization is enough to cover N-1 criteria. But, again, no Is there a citation or further explanation is provided by the authors. How is this assumption justified for this choice?
- Demand projections for exporting countries (P11-L39): no context is given on the projected future energy demands for these nations. Many of the countries included are developing economies, meaning internal demand is expected to increase, which in turn would affect your export prices. This can have major impacts on the results of the study, since you assume local demands are met first before exportation. Can you give more context on how future local energy demand was projected?
- Underground storage (P11-L102): you state that underground hydrogen storage was excluded from the TRACE analysis due to limited availability in regions with high renewable potential. Yet your citation does not fully support that statement: the presentation you refer to shows that most countries included in your study (Eastern US, China, Argentina, Chile, MENA, Turkey, Kazakhstan, Canada) have salt cavern deposits, and it has no analysis on whether or not these caverns are located in regions with low renewable potential. Besides, you assume that the best regions will be used for internal demand, which also runs contrary to this statement. How can this assumption be justified?

3. Framing of the relocation of generation capacity

I would like to express some concerns in regards to the framing of the study, particularly in regards to the limited discussion of resistance by local communities to expansion of renewable projects in exporting countries, and within Europe to the displacement of heavy industries. While I understand that these are not within the scope of the methods of the study, I believe they are definitely within the scope of the discussion, particularly when local resistance within European nations is one of the drivers for the relocation of production facilities abroad (the authors have even constrained the model's ability to deploy onshore wind generation due to local resistance), and due to the overall concerns of extractivism in developing nations in the context of the energy transition.

At the very least, I would expect the limitations you mention when it comes to these topics to be somewhere in the main text, not relegated to the S.I. This goes beyond social concerns: these are prime sources of uncertainty in your study that are not addressed by your methods. Proper discussion is, in my opinion, necessary.

Additional detailed comments:

Some of the figures featured in the study are set up in ways that may confuse readers. Splitting them, or at least re-arranging them, would significantly improve readability.

- Figure 1: please consider inverting the choropleth so that its usage is consistent throughout the article. This figure misleads readers into thinking that Northern countries have the lowest hydrogen import costs, or that Argentina and Chile are the most expensive. See figure 7.
- Figure 4: please consider separating this into two different figures, or rearranging so that the map and price data are outside of the bar plot. The current configuration obstructs analysis more than aiding it. Similarly, I would suggest inverting the choropleth to match the setup used in figure 7.
- Figure 6: as it stands, readers might be misled to believe that the relation between import/export does not change between scenarios. Although I understand that only the base scenario was kept for readability, there is nothing to indicate that the

dashed line also changes significantly between these scenarios. The relation between domestic imports and exports is of core importance to the statements provided by the study. I suggest displaying this more clearly.

• Figure 7: the sizing of technologies is quite difficult to ascertain due to the limited space given for the six figures. At the moment it is difficult to discern the direction of hydrogen transmission. Similarly, the Bay of Biscay data in the center-left should be given its own space to increase clarity.

P1-L12: Suggest to replace “throttled” with “constrained” for clarity.

P2-L50: please clearly state that PyPSA-Eur only optimises for a single year here, and clearly state that the model has perfect foresight for technology operation.

P2-L99: consider removing the clarification that Argentina and Chile are the lowest-cost suppliers for Europe to another section. This appears to be result from your exercise, not introductory context. Unless you mean to say that only these countries were considered for hydrogen imports (which does not follow from your methods section).

Figure 3: import mix is missing the added sum of total supply (lower left).

P6-L7: “imported fuel fuels appear to be...”

Figure 6: “Cost alterations are uniformly applied to all imports options import options but direct electricity...”

Figure 7: to further improve readability, consider moving the technology legend to the bottom of the figure, this would provide more space for the maps without additional confusion.

P10-L53: please consider providing further justification of why this particular year was chosen to test the system.

P10-L68: please consider explaining the effects that this has on the study in the limitations (e.g., underestimating the system cost of negative emission technologies).

P10-L106: please consider providing proper citations to justify the assumptions for shipping, heavy-duty vehicles and aviation.

P10-L115: please consider providing a citation for the operational restrictions of Fischer-Tropsch, methanolisation and methanation.

P11-L15: please provide a citation for the costs of H2-DRI and EAF.

P11-L117: this entire paragraph appears to belong to the results section of the study and has a broken citation. Was it misplaced?

S.I.: please consider providing a full table with your cost assumptions and a citation for them.

(Remarks on code availability)

The code is available and has appropriate documentation, but I did not attempt to install and run it.

Reviewer #2

(Remarks to the Author)

(Remarks on code availability)

I've reviewed the code (mostly the snakemake workflow provided in the linked repository) and some of the data used. Things to note:

- Salt cavern potentials in the model appear to be at a national level. It's unclear how this is translated into the sub-regions used by the model (100+ nodes).
- Generally there are not a lot of notes on how or where the data is sourced from. This might not be an issue, but it is difficult to trace these processes without further notes from the authors.

Reviewer #3

(Remarks to the Author)

This is a nice article addressing a very relevant topic.

It is, however, difficult to understand the results without a better description of the data assumptions and modelling choices. In general, the conclusions should be substantiated by results, which it should be able to understand based on the descriptions of data and modelling. This could be improved. Some questions which the reader is left with are:

- How are the demands modelled? What can the different fuels be used for? Which are the competing supply options? And

how are the demands distributed across Europe?

- Which options compete with hydrogen for providing backup power and heating?
- Which fuel pathways are modelled and how? Are "blue" options taken into account?
- DAC seems to be cheap. Which cost assumptions are used for the different technologies?
- What do Fischer-Tropsch fuels entail? For which purposes can they be used? And how are the costs and competitions modelled?
- What is assumed regarding the potentials for solar and wind power in EU and globally? Which biomass types have been investigated and what are their assumed uses and potentials in EU - and similarly for biogenic CO₂?
- How is the variability of solar and wind power taken into account for the imported fuels? Do the cost estimates include potential required oversizing and batteries? The description of the modeling of green fuels in the TRACE model should be improved and similarly, and so should the linking of the TRACE model with the PyPSA model.
- It is concluded, that PtX flexibility has a benefit in terms of integrating renewables, however, no results are shown with respect to this - and the modeling of it is not clearly described
- An important assumption appears to be the flexibility of different synthesis processes. There is however no reference (just a question mark) for the assumption and as this is debated it could merit a sensitivity analysis what the impact of the assumption is on results
- Are the imported fuels assumed available e.g. in ports at a constant rate - or when the demand is there? Do the investments in infrastructure include bunkering and storage investments in ports? Which ports are considered?
- It is surprising that the MENA region including Morocco doesn't come out better. What are the assumed costs of shipping the fuels e.g. from Chile?
- Is the same WACC used for all countries?
- How is the relocation of industries modelled?
- Did you take the potential for repurposing infrastructure into account?
- Did you check whether the implemented pipeline and cavern storage sizes are realistic?
- The main sensitivity is on +/- 20% changes in imported fuel costs. This range seems narrow compared to the inherent uncertainties and is poorly argued.

The article is well written and with some very nice illustrations (e.g. Fig 2), although some figures are packed with information and rather confusing (e.g. Fig 7) and line 58-62 is difficult to understand.

Validation should be added in terms of comparing with the results of others - both in terms of global green fuel prices - and of future European energy supply and infrastructures.

(Remarks on code availability)

Reviewer #4

(Remarks to the Author)

Summary

The current analysis tries to assess the impact of importing energy vectors (e.g., hydrogen, methanol, ammonia, and steel) on the future of European electricity and hydrogen infrastructure. The research utilizes PyPSA-Eur, a large-scale open-source energy system model assuming perfect knowledge and competition. The Trace model, which is soft-linked to the previously mentioned European energy system model, is used to quantify the costs of imported energy vectors. The study reveals interesting findings and suggests that imports could reduce system costs by 1-14%, with hydrogen and its derivatives providing the greatest benefits. In addition, the current study challenges how electricity and hydrogen networks should be developed in conjunction with potential energy commodity imports.

Overall comments and questions

From an energy system modeling perspective, there are questions regarding the novelty of the model approach to model imports.

- 1) Should electricity imports be modeled exogenously with a constant price similar to the other commodities? Accounting for electricity endogenously yields a 4-hourly opportunity cost (i.e., endogenously estimated based on renewable availability in third countries) as well as the costs of establishing electricity transmission lines compared to importing hydrogen or other commodities. In other words, power imports are accounted for as the total of CAPEX and variable OPEX, whereas other commodities are imported as OPEX based. Is this a fair comparison? For example, since you have represented power imports via transmission lines, could hydrogen pipelines not be modeled?
- 2) Can you provide details on the import price estimation and assumptions by displaying them in tables? Supplementary material could be populated with additional data on the input parameters and assumptions on both PyPSA-Eur and Trace.
- 3) Additional information about the Trace model is necessary. Is this a simulation or optimization model? Please specify the levelized cost components that the model considers. Are the transportation costs to the European system boundaries accounted for?
- 4) It's not clear how you limit your exogenous imports. Do you have an annual limit per importing point? Do you assume that shipping imports can only occur once or twice per week? Overall, how do you limit imports in terms of time and space? There should be differences between ships and pipelines. Do you assume investment costs for building and importing infrastructure in ports?
- 5) Where are the European demands of hydrogen, derivatives, and steel geographically located? Do you account for transportation costs from the imported port node for steel or derivatives to the demand side?

- 6) Despite claiming great spatial aggregation, the Balkan countries are only depicted as one node per country (e.g., Fig. 1), even though the most importing choices are neighboring those countries. Is one node per country underestimating the expense of transporting those commodities to Central European countries?
- 7) Could you share insights into the potential utilization potential of imported options? The spatial disaggregation yields interesting results. Is there a formation of an importing corridor, such as methanol from Chile to the Netherlands? Where are the majority of hydrogen imports that account for the greatest system cost reduction?
- 8) The generation expansion results are based on uniform assumptions across European countries, assuming collaboration to attain a net-zero energy system aim. In recent years, we have noticed countries establishing or examining bilateral agreements (for example, Germany with Oman, the United Kingdom, and Canada). However, in the current modeling technique (if understood correctly), Germany's hydrogen demand might be met by pipeline imports from Italy that serve as a landing zone for imports from North Africa. Shouldn't a constrained scenario of network development (electricity and hydrogen) reflect the possibility of reaching those bilateral agreements? Overseas imports are projected to increase to satisfy the regional demand. Yet the total system cost will be bigger than all imported options and network expansion, demonstrating the benefit of having a European dedicated common strategy.
- 9) Although the authors claim that they do not model pathways, the literature shows that imports play a significant role in the years 2030-2040 when the installation rates of renewables are limited. Could there be lock-in effects from importing a specific technology-ready commodity over another alternative?
- 10) Please provide a summary mathematical description of Trace and PyPSA-EUR in SI. Since the importing topic and its modeling attempt are the main methodological novelty of the current study. Furthermore, how did you model the steel relocation (page 2, line 53)? So, is there a steel demand that can spatially change? Does the model utilize this option?
- 11) Hydrogen derivatives and steel are imported through marine infrastructure. Instead of depicting the LNG terminals that the model did not use, Figure 1 should display the location and capacity of the ports (pie chart illustrating the commodities) that will handle those shipping imports.

Specific comments

- Abstract: Please indicate that the current study incorporates 2050 insights.
- Line 1: Please offer a reference for why "importing renewable energy to Europe promises several advantages for achieving a swift energy transition".
- In line 7, avoid lump sum references (1-8) and instead appropriately cite the research or exclude unnecessary ones.
- Nature Communications does not permit footnotes—first page, right bottom.
- Page 2, line 20. Are you referring to Supplementary Figures 6 and 7? Those figures do not demonstrate the unique characteristics of the imported vectors.
- On page 2, line 50, many details are also mentioned in the method sections. You can choose to avoid some options, such as 110 areas or 4-hour resolutions.
- page 2, line 65, please include some figures or tables showing the spatial distribution of the demands (hydrogen, methanol, ammonia still, etc.). Could be at the SI. Your demand allocation drives the network development.
- page 2, even after finishing page 2, there is no mention that most of the assumptions (except technology cost) discussed and importing options are based on 2050 estimates. Kindly include some information for the time horizon of the study.
- Page 2, line 80. The title does not accurately reflect the content. Here, you reflect on some background information. The authors do not perform a qualitative or quantitative assessment of importing choices.
- page 2, line 85. "Our selection of exporting countries". Should it be importing countries?
- What is the domestic average cost of those energy vectors in Europe so as to compare to Figure 1b (violin plot)? Perhaps direct the reader to the figures in Appendix Sup Fig 5 where you show the supply curves. Or comment on it.
- In Figure 4, the bubble in Greece appears to have shifted down from the polygon center.
- Page 6, line 31. How is it feasible to integrate all the waste to heat? In some nations, district heating and other choices are unavailable. Even if you assume that waste to heat has a sector-coupled value, is that enough to influence the importing results? Did you attempt a sensitivity that assumes a smaller heat integration?
- Page 6, line 56. The 20% sensitivity to the default import cost assumes that the estimated initial import costs have the same level of uncertainty. However, this is not the case. For example, electricity imports, hydrogen imports, and steel exhibit different uncertainty costs. Could you kindly discuss the current estimates and their future evolution, as well as your decision to choose the 20% to perform the current sensitivity analysis?
- Figure 6: it is difficult to differentiate colors in the legend; try aggregating some possibilities. For example, fossil fuels is one category, and domestic renewable energy production is another, so as to demonstrate the impact of imports, which is the primary focus of the work.
- On page 8, line 64, please put your estimated values in perspective. What are the current annual expenditures of importing LNG compared to the future system savings from importing an alternative energy vector?
- Discussion and conclusion. It reads well but might benefit from additional information, such as key values and comparisons to other studies.
- Figure 7. Again, it's challenging to comprehend the graphs because so many variables have similar colours. Also, consider showing fewer plots in the same picture; someone must zoom in (at least 300%) to see where the production and consumption occur. In the fourth plot there is half-circled above Portugal and Spain. What does it represent? Why are two arrowheads connecting Romania, Hungary, and Austria in the same line?

(Remarks on code availability)

The link does include a readme file that contains sufficient information on how to install and use the model. I did not investigate whether I could install and run the model in this review stage.

(Remarks to the Author)

Key results

The study gives a very good insight into the effects of possible future energy imports in a carbon-neutral Europe on the resulting system costs and the needed infrastructure in Europe. The authors find relevant cost reduction potentials of energy imports (including steel) compared to a system with fully domestic production.

Validity

The authors are using a rather detailed representation of the energy system (with some necessary simplifications that do not harm the validity of the study). The resulting limited import shares in the main results lie in orders of magnitude of realistic import shares and quantities.

Significance

The study is highly significant, especially behind the backdrop of current policy discussions at the European level about future clean/green fuel imports and (planned) subsidy schemes for imports like the European Hydrogen Bank.

Data and methodology

The used data and model have been fully provided by the authors. The authors are using an established multi-energy system model that has been used in many prior publications. The model setup and the import extensions seem fully reasonable and the approach is thereby completely valid. Some assumptions are made for complexity reduction but they are discussed in the supplementary material. I can see a few possible improvements further discussed below. The input data is based on established sources and therefore also valid.

Analytical approach

The main results are presented in a very convincing manner. Scenarios and sensitivity tests are run to further examine the results. The chosen scenarios and sensitivities are sufficient for a robust derivation of conclusions. Some possible improvements are elaborated below.

Suggested improvements

The manuscript already adheres to a very high standard. Nevertheless, I would like to suggest thinking about the following improvements:

~ Import dependencies are mentioned in the text as a risk factor. In the results and discussion, this is not really taken up again. I could not find any detailed information on the quantities imported by country. It is very likely that the imports will only come from the lowest-cost region since supply seems to be abundant but maybe this is not the case and we could see an import mix. The supplementary material does discuss cost increases or decreases for an export location and also tests sensitivities but as far as I understood only as a uniform increase or decrease for all exporting regions. A more nuanced discussion of this aspect could increase the policy relevance of the paper. It could be interesting to remove certain (lowest-cost) countries to test the effects.

~ Domestic supply is assumed to have a non-flat cost-duration curve, while imports are considered at a fixed/average cost as far as I understood correctly. The supplementary material discusses the influence of running the export systems as islanded systems and the need to overplant or usage of storage. I am not fully convinced that the exporters would offer the energy vectors at a flat cost curve and think we would rather see seasonal price variations (due to the transport by ship or pipeline not hourly/daily but still). Could you please elaborate a bit on why you made these assumptions and - if possible - test the influence of that on the results? It does seem to favor imports significantly over domestic production but maybe the effect is negligible.

~ You are using a greenfield approach with cost data for 2030 to reflect the need to build infrastructure rather sooner than later. While I think this is a reasonable assumption, I am wondering about the effects of existing generation capacity in Europe that will most likely be in the market for a very long time to come. I am thinking of nuclear capacity here in countries like France, the UK, Sweden, and Finland but also in Eastern Europe with lifetime extensions of existing plants well into the 2030s but also new-build reactors that just entered the system or are about to enter it or will "soon" be there based on the current political discussion and situation in Europe. Together with the large capacities of fluctuating renewables and the rather inflexible nature of NPPs as well as the European regulation on clean/green hydrogen from electricity systems with very low emission rates, this could influence the competitiveness of domestic supply due to the inflexibility of generation - even though it is very expensive to build since NPPs are comparably cheap to operate and the construction costs are sunk anyways. I see that this is out of scope for your current greenfield model setup but I think it should still be touched upon in the discussion and/or limitations of the study. Already existing NPPs or those under construction or with an FID taken might not tip the scale due to the limited capacity. With the current push for many more new projects and new countries entering the nuclear market like Poland, the picture might change, though. Still, we can be very skeptical about the realization of such projects. As far as I see, the model setup currently does not consider any possible investments into new nuclear capacity (and it is unlikely to materialize at realistic cost assumptions). Maybe you don't want to open Pandora's box here and this is

for another paper - I would just like to at least see it discussed.

~ As I understood also the potential hydrogen pipeline infrastructure is based on a greenfield approach (while powerlines already exist). This is not in line with the current plans of repurposing large parts of the existing fossil gas infrastructure. Propagated industry plans but also other works of yours are assuming a rate of 60% repurposed pipelines in the future hydrogen system. While the real cost rates for repurposing are not yet known and potentially underestimated, this might still change parts of your outcomes. As far as I understood you did not include this aspect due to computational burdens. Could this not be depicted in a slim manner by assuming lower cost factors for existing relations? The aspect is only mentioned in the "Methods" section so I did not see any discussion on the implications of this assumption on the results.

~ Graphs: The choropleth layers in the map graphs are using color scales reaching from darker colors for lower specific costs to lighter colors for higher specific costs. While this can be intuitive in a way (more imports from lower-cost regions) it is a bit confusing to use an inverse scale here, especially since countries or regions not considered or not containing entry points are light gray and hard(er) to distinguish from high-cost areas. Maybe the scale could go from green/blue for low cost to red/orange for high costs to better distinguish this (or any other color suitable for colorblindness).

Clarity and context

The manuscript is drafted in a very clear and accessible language and follows a read thread. The results are presented in complex but informative graphs and the supplementary material covers a number of sensitivities and takes up the limitations of the model and setup at hand. The visual quality of the graphs stands out and the use of vector formats contributes to a very high legibility (in contrast to many other studies, unfortunately). (The manuscript contains a few typos.)

References

The study deals with the existing literature in detail and clearly distinguishes the study from already existing similar approaches and reasons the novelty at a sufficient level.

(Remarks on code availability)

I have checked the repository visually: the full code and data are available and well-documented for the experiment to be reproduced. I did not execute the code due to the complexity of the setup.

Version 1:

Reviewer comments:

Reviewer #1

(Remarks to the Author)

Overview

The new version of the article is significantly improved in terms of clarity and uncertainty assessment. In particular, I am pleased to see more nuance in the discussion of the trade-offs between European energy security and cost reductions, the improved clarity in the figures and the expanded explanation of the two models utilised.

However, I believe that some changes are still needed. I will go over these in a case-by-case basis.

Major comments

1. On whether or not allowing for relocation of energy intensive industries is a sound default case.

I am glad to see an exploration of not allowing this relocation in the supplementary material. The authors correctly identify the lack of data on this dimension in their response, meaning that comparing the relocation / no-relocation scenarios in terms of costs is difficult.

However, the article still needs more transparency on the degree of uncertainty that this assumption implies. Other studies not accounting for this implies a gap in the field, not a general truth. It should not be casually dismissed.

I have two suggestions:

* Directly stating this assumption in the abstract of the article. Without it, the 1-10% cost reduction mentioned in it would be misleading.

* Figures S6 and S25 are useful, but not enough to put other results into context. Even if comparisons to your default case are difficult, consider generating something similar to Figure 2 for this case. I am sure other researchers and policy-makers would appreciate a nuanced discussion.

2. Techno-economic costs and their implication.

The first version of this article utilised cost projections for 2030, stating that infrastructure must be built well in advance. I am disappointed to see these costs moved to 2040, as it adds even more uncertainty to your results. I would argue that this does

not really make the article easier to contextualise, as you compare to targets for 2030 in some sections (P8L212), state that your import costs reflect those of studies focused on 2050 (P15L450) and use potentials from 2050 for biomass (P18L595).

Suggestions:

- * The date should be consistent with the general question you are trying to answer.
- * Justify and contextualise the implications of this change and the imbalances it creates in the limitations section, following your initial arguments.

3. Making scenarios clearer.

The article features a wide array of scenarios dedicated to both parametric and structural changes in both of your models. To make them easier to trace, I suggest detailing them in a table in the supplementary material. Stating if a figure relates to your default case explicitly rather than implicitly (e.g., including 'default' or something similar in figures like Supplementary Figure 11) would also help readers.

Additional detailed comments:

P16L536: in "Existing hydro-electric power plants are included, as well as nuclear power plants 536 built before 1990 or currently under construction according...", perhaps you meant "built after 1990"?

(Remarks on code availability)

Reviewer #2

(Remarks to the Author)

(Remarks on code availability)

The improved documentation of the model's assumptions is appreciated. However, the authors do not directly state which version of PyPSA-Eur they used in the GitHub repository. Similarly, the link to the model's documentation points to the newest version of the model, meaning that it'll deviate from the version used for this study over time. Since it is not versioned, it does not really ensure reproducibility.

There are ways to infer which version of the tool was used, but they are GitHub submodule links that may break over time.

I have some suggestions:

- Directly state which version of the key tools was used for the study. This can be through an environment file or a simple README section.
- Consider versioning your model's documentation.

Reviewer #3

(Remarks to the Author)

Dear authors

Thanks for the substantial revision.

For most of my questions and comments, I can see from the response that you have either amended the text, included new explanations or new model runs in the main article or in the supplementary information. There are however comments for which it is unclear to me if you have addressed the comments apart from in the response.

As I asked the questions and provided the comments on behalf of potential future readers and not for my own curiosity, I would appreciate that they are addressed in either the main article or if need be in the supplementary material.

(Remarks on code availability)

Reviewer #4

(Remarks to the Author)

(Remarks on code availability)

I would like to thank the authors for their efforts in addressing the first round of comments and providing an updated manuscript, and providing additional sensitivities.

Although the manuscript looks polished and updated, some questions still need to be clarified.

#major

1) If this aspect is addressed in your manuscript or supplementary information, please clarify it again in the main manuscript, similar to your response, by explaining the reasoning behind not spatially fixing the hydrogen demand.

Currently, approximately 8 MT of grey or black hydrogen is produced domestically in Europe, with production typically located near industrial clusters. Will these existing projects become stranded assets, or are they expected to relocate as well?

-Your answer

In fact, only in scenarios without relocation do we have substantial amounts of spatially-fixed hydrogen consumption for steel and ammonia production. Apart from these industries, there is just a 5 TWh/a demand for hydrogen for high-value chemicals (other than the liquid feedstocks), which is counterbalanced by spatially-fixed hydrogen production of around 10 TWh/a from chlor-alkali electrolysis in the chlorine production. High-temperature industry heat is assumed to be supplied by methane (fossil, synthetic, or biogenic). In the transport sector, there is no direct hydrogen consumption in light-duty or heavy-duty road transport since it is all electrified. In district heating and the power sector, backup hydrogen capacities (CHP or turbines) are endogenously sited just as as the production capacities of other hydrogen derivatives (Fischer-Tropsch, methane, methanol). As there is little to no spatially-fixed hydrogen demand assumed when the steel and ammonia production can relocate, there is little impetus for transporting hydrogen directly, apart from transporting hydrogen to e-fuel production sites in regions where there is high value for its excess heat. The result is a network that is much smaller than, for instance, envisioned in the European Hydrogen Backbone [50].

2) In your modeling exercise, you do not account for the costs associated with the relocation of hydrogen demand or the relocation of other industrial activities, nor do you consider the final transportation costs of the produced hydrogen to the end consumer. While you acknowledge that hydrogen demand may not remain fixed in the future, the model does not appear to incorporate the additional benefits and costs associated with shifting activities or pathways. As a result, Figure 7 in the main text depicts a highly concentrated hydrogen network and electrolysis production, primarily in 3–4 European countries, which absorbs most hydrogen-related activities. This outcome seems skewed and differs significantly from Figure 7, which was presented during the first round of revisions. (Please explain the reasons behind those differences from the initial submitted manuscript). Consequently, while the solution represents a valid modeling result (solved to optimality), it becomes difficult to evaluate its novelty or its relevance for large-scale energy system solutions that could inform future policy.

3) Furthermore, while the model accounts for multiple imports and an optimal allocation of hydrogen and derivative demands, the benefits are reported to amount to €37.1 billion per year (or a 4.44% improvement in the objective function), mainly driven by reduced renewable capacity expansion, smaller networks and storage requirements, and decreased electrolysis and related infrastructure. However, these results focus on a minimum-cost perspective (without cost components of comment 2) and do not consider critical economic impacts, such as local value creation around industrial hubs. For example, the steel industry alone generates €166 billion in annual turnover, representing 1.3% of EU GDP and providing 328,000 direct jobs (as highlighted in the European Commission's report: https://ec.europa.eu/commission/presscorner/detail/fr/memo_16_805), alongside numerous indirect and dependent jobs. Additionally, the current analysis disregards first-mover effects, the development of R&D, and other socio-economic benefits. These aspects should be highlighted to present a more comprehensive evaluation of large-scale hydrogen deployment and its broader implications for regional development and policy orientation.

4) The authors claim to have described their modeling approach and mathematical formulation in previous work. However, the references provided do not include any explicit formulation or methodology for relocating demands (e.g., steel), leaving this aspect of the model unclear.

5) In your discussion you claim:

Policymakers in Europe might prefer such easy-to-implement systems featuring lower domestic infrastructure requirements, reuse of existing infrastructure, lower technology risk, and reduced land usage for broader public support than the most cost-effective solution. Moreover, policies favoring local energy supply chains and importing intermediary products like sponge iron could preserve European jobs while outsourcing only the most energy-intensive processes. However, in shifting potential land use and infrastructure conflicts abroad, where population densities are often lower, potential exporting countries must weigh the prospect of economic development against internal social and environmental concerns, particularly in countries with a history of colonial exploitation.

These elements are not clearly supported by the literature or by explicit outcomes of your research. For instance, what is meant by the term easy-to-implement systems? Did you demonstrate how much land usage your modeling approach could save? Additionally, did you showcase a solution that is indeed cost-effective?

Overall, I recommend that the authors reconsider the final paragraph of the paper. It would be more appropriate for the discussion-conclusion to synthesize the key results of the manuscript rather than presenting hypothetical claims.

#minor

Figure 6, in the legend, is missing the hydrogen transmission costs; are they negligible?

Reviewer #5

(Remarks to the Author)

Thank you for the very thorough revisions and detailed answers to the reviewers' comments. I have no further comments since all my suggested improvements have been dealt with.

(Remarks on code availability)

I have checked the repository visually: the full code and data are available and well-documented for the experiment to be reproduced. I did not execute the code due to the complexity of the setup.

Version 2:

Reviewer comments:

Reviewer #1

(Remarks to the Author)

The authors have adequately dealt with all recommendations. No further comments or issues!

(Remarks on code availability)

Reviewer #2

(Remarks to the Author)

(Remarks on code availability)

I partially reviewed the code and documentation and found it transparent enough. The modifications to ensure version history are appreciated.

Reviewer #3

(Remarks to the Author)

Thanks for the thorough answers and reviews

I am now satisfied that the article is ready for publication

In the end, the title could however be adjusted to better reflect the actual content of the article and the commodities which are imported including steel, which is the main novelty of the article

(Remarks on code availability)

Reviewer #4

(Remarks to the Author)

(Remarks on code availability)

Author Response to Reviews of

Energy Imports and Infrastructure in a Carbon-Neutral European Energy System

Fabian Neumann, Johannes Hampp, Tom Brown

Revision #1: November 14, 2024

RC: Reviewer Comment, AR: Author Response, ■ *Manuscript text*

We thank the reviewers for their constructive comments and acknowledge the time and effort they have spent in assessing our work. We have revised the paper based on the feedback from the reviewers and hope that we could adequately address their primary concerns.

In response to the reviewers' comments, we have rerun all scenarios and supplemented our study with additional sensitivity analyses. We have also expanded the manuscript to provide more detailed explanations of TRACE and PyPSA-Eur. Moreover, we took the revision as an opportunity to update our analysis to the latest version of PyPSA-Eur (from v0.9 to v0.13), incorporating model developments from the past year. These include, in particular, updates to technology assumptions (e.g. higher electrolysis costs of 950€/kW_{el}), updates of energy balances (JRC-IDEES 2021), the addition of new technologies like floating offshore wind, losses in power distribution systems, OpenStreetMap as new source for high-voltage grid data, updates to planned transmission projects, new data on the locations and capacities of steel plants, ammonia plants and LNG terminals, as well as improvements to how heat demand and district heating systems are modelled. We also now include existing and planned nuclear power plants, as well as hot briquetted iron (HBI) as an additional import commodity. More details on recent model changes can be found here: https://pypsa-eur.readthedocs.io/en/latest/release_notes.html

To follow the revisions made, we have highlighted differences compared with the previous submitted version of the paper in blue and red text in an attached file. Given the length of the review responses, we chose not to show quotes from the revised manuscript, but refer to the marked up version to follow text changes. Figure numbers in the responses refer to the numbering in the revised manuscript.

Reviewer #1

RC: Overview

The article assesses the important topic of green energy imports in the European context by coupling a spatially detailed energy system optimisation model with a supply chain model.

The main focus was to compare a self-sufficient net-zero European energy system against different scenarios where green energy imports are allowed. The scenarios have the following characteristics:

- Roughly 8 different import commodities are allowed, ranging from simple carriers like electricity and hydrogen, to more complex end-products such as Fischer-Tropsch fuels and green steel.
- 14 different exporting countries are considered, with the caveat that they must first satisfy their expected internal demands before exporting these products.
- 2030 prices for technologies are considered.
- The model uses a green field approach, and even allows heavy industries like steel to relocate within Europe to achieve optimal results.

The main result is that energy imports can marginally decrease the cost of a net-zero European system by up to about 5%. Importantly, the optimization model's preference for using steel and methanol imports appear more robust to price variations than hydrogen and Fischer-Tropsch fuel imports. However, there is a broad range of uncertainty in these results, particularly when it comes to price variations and import availability. In particular, the authors highlight the need to coordinate import strategies with infrastructure decisions (which intuitively makes sense).

These results will be important to both modelers and policy-makers in the field of energy planning, particularly in light of recent import targets set by the EU.

I would like to express some concerns about the study, both in terms of methods and framing. I believe that in its current form, the paper is not suitable for publication, and three major issues first need to be thoroughly resolved (numbered 1-3). In addition, I provide a list of minor issues that ought to be addressed before publication.

AR: Thank you for your assessment that this paper will catch interest among policy-makers and energy system modellers. We hope we could dispel your concerns with our revisions and responses below.

RC: Major comments

1. Robustness and limited sensitivity analysis

The authors put emphasis in highlighting the robustness of their results in their introduction and conclusion (P2-L13, P5-L87). However, I do not believe that the paper's methods provide enough uncertainty analysis to justify such an assertion, especially considering that the core result is a

low variation of total system costs of 5%.

Robustness usually implies quantifying the range in which a strategy performs adequately (either ex-post or within the model). The authors mainly address this by varying which imports are available, or by evenly varying the price of all imports (between +20/-30%). However, this range is not justified with a historical counterfactual. Considering that the increase of annual natural gas prices by 200% is one of the motivating reasons for the policies driving this article, and that oil prices can spike by more than 100% between years, I struggle to see how such a low homogeneous increase addresses this sufficiently.

AR: We agree that historical energy crises have set notable precedents for drastic price spikes in global fossil fuel trade. Most of these price spikes have been induced by either real, artificial, political or perceived scarcity (e.g. 1973 OPEC embargo, 1979 Iranian revolution, 2021-2022 energy crisis) rather than changes in production or extraction costs of the fuels. Since such scarcity prices are difficult to predict and often do not persist, for our long-term modelling we focus on uncertainty in the costs rather than prices. In particular, we seek to examine the system impact of cost variations relative to domestic supply chains (e.g. higher/lower WACC, DAC costs, electrolyser costs abroad). Furthermore, previous energy price spikes have occurred for fossil fuels, where production is concentrated in a small number of states that have exploited their market power. Because good renewable resources are more widely distributed, we do not expect such extreme market power or resulting price spikes for green products.

In response to your comment, we have now expanded the range of uncertainty considered to $\pm 50\%$. We believe that this range sufficiently captures the interactions with European energy infrastructure, considering that beyond 30% higher import costs the model no longer imports energy and below 50% lower import costs, the costs for the domestic energy system are only marginally reduced (Figure 6). In Table 1, which was moved to the main body, we also showcase what can cause such cost variations.

We also concur that uncertainty in different cost components affects import vectors to varying extents. We therefore now perform three (previously two) categories of import cost variations to reflect inhomogenous uncertainties in costs: one where cost variation is applied to all import vectors (Figure 5a), one where it is applied to all import vectors but electricity, reflecting cost uncertainty in electrolysers (Figure 5b), and one where it is applied only to carbonaceous fuel import vectors, reflecting cost uncertainty in carbon provision and hydrogen conversion (Figure 5c). For computational reasons, we needed to group the technology uncertainties in this way.

RC: Other major assumptions of the paper (complete freedom to re-locate industry, and lack of competition between Europe and other world regions for fuel imports) also make me doubt the robustness of the analysis. Are you sure that these would not have significant impact on the results of your study? Similarly, in the S.I. you state that if some importing countries were not willing to export (P1-L27), costs could raise by 10%, yet this is not included in your sensitivity analysis.

AR: **On the relocation of the steel and ammonia industry:** We agree that free relocation of steel

and ammonia production is a strong assumption. Therefore, we include additional scenarios where current production sites for steel and ammonia are kept. In the revised manuscript, we have expanded the analysis of relocation patterns; for instance, we added a series of maps showing the observed relocation movements in Supplementary Fig. 6. However, given limited data on what the migration costs might be, we cannot associate a cost with the relocation we allow. We now mention this in the methods section.

Nevertheless, we think that for our study this assumption is a good default case as it isolates the benefits of imports from abroad from those of relocating steel and ammonia production to where hydrogen is less costly. By allowing relocation, the best sites for ammonia and steel production within Europe can compete with imports from abroad. Otherwise, the benefits of imports from outside of Europe might be overestimated. The consideration of industry relocation is also in line with other recent literature [1, 2, 3, 4].

On global competition for green fuels: We concur that competition with other importers would affect pricing of green fuels and other commodities. Here, our assumption that exporters offer their commodities based on their average cost resembles long-term purchase agreements between exporters and importers that ensure cost recovery on an annual basis. In the alternative market-based pricing approach you mention, market prices would generally be higher than those from the lowest-cost exporter, as market pricing would allow cost recovery even for higher-cost exporters while offering higher profits for lower-cost exporters. While not explicitly modelling mark-ups of global market clearing prices, our import cost variations can be used to approximately gauge the impact of certain mark-up levels.

Reflecting the global market dynamics endogenously would require including all potential exporters as well as potentially competing offtakers next to Europe, which we believe goes beyond the scope of the paper. Our cost-based pricing approach could also be seen as approximating a scenario where both direct trade agreements and market-based trading exist together, similar to today's LNG market. Export/import agreements would cover most of the expected demand (as modeled here), while the market would address deviations from longer-term demand/supply expectations (which we neglect). We now discuss this aspect in the revised manuscript.

On cheapest exporters dropping out of market: We think there is a small misunderstanding here. The 10% cost increase refers to the unit cost increase of the imported fuel and not to the total system cost. We have clarified this in the revised manuscript. Additionally, we now show the import cost curves for different fuels in Supplementary Figs. 7 to 10, which illustrate the cost rise when the cheapest region(s) drop out of the market. While not explicitly modelling this scenario, the resulting shift of the import cost curve to the left is – with some limitations – similar to moving the curve up by a certain percentage (which we do show in our main sensitivity analyses). With our revised selection of exporting regions, the gap between low-cost suppliers is now also much smaller.

RC: I believe that one of two adjustments is needed: Either the paper needs to re-frame the findings to state the limitations more clearly in the main text, or the authors need to perform additional

sensitivity analysis for the problematic dimensions outlined above.

AR: In the revised manuscript, we have adjusted both. We use assertions implying robustness more carefully and moved a revised limitations section from the supplementary material to the discussion in the main text, highlighting that potential competition with other world regions for fuel imports is not reflected. Second, we have expanded the range of sensitivity analysis regarding import costs and study the impact of industry relocation of steel and ammonia on our results in more detail (as outlined above).

RC: 2. Justification of assumptions

The methods section lists a wide range of assumptions made within the study, but in many cases a proper justification for them is not provided. Although I fully believe that the authors had proper reasons, it is not possible as a reader to discern why they are present or if the choices made are justified. I think the authors should provide justification for the following assumptions somewhere in the text:

AR: We have improved the documentation of sources for our assumptions in the revised manuscript. More specific responses follow below.

RC: List of countries (P2-L85): only a few list of importing countries are included (14, for what I gather from Fig 1). But why these specifically? If this is based on analysis of policy texts, or just because they are the only countries evaluated by TRACE?

AR: In principle, TRACE could calculate costs for any country/region. We agree that our previous selection of countries was perhaps a bit too limited. In response to your comment, we have now expanded the coverage to 53 regions and split some large countries into multiple subregions for a more differentiated view (USA, Argentina, Brazil and China). We believe this selection to be expansive and include the most promising regions for green fuel exports.

Broadly, we include countries with favourable renewable potentials according to the Global Wind Atlas (<https://globalwindatlas.info>) and Global Solar Atlas (<https://globalsolaratlas.info>). We exclude countries which do not have sufficient potential to cover their own demand plus 500 TWh of fuel as well as some landlocked countries outside the reach of realistic pipeline transport to Europe due to an assumed lack of access to shipping ports. Furthermore, we exclude some countries due to political instability (e.g. Sudan, Somalia, Yemen), using a Fragile States Index value of 100 as a threshold (<https://fragilestatesindex.org>), or due to imposed sanctions by the UN/EU beyond asset freezing and admission restrictions (e.g. Iran, Russia, Venezuela, Iraq), following the EU Sanctions Map (<https://www.sanctionsmap.eu>).

The full list of countries included is: Canada, USA, Mexico, Colombia, Peru, Brazil, Bolivia, Chile, Uruguay, Argentina, Ukraine, Turkey, Senegal, Mauritania, Western Sahara, Morocco, Algeria, Tunisia, Libya, Egypt, Saudi Arabia, Oman, Nigeria, Eritrea, Ethiopia, Kenya, Tanzania, Mozambique, Madagascar, South Africa, Namibia, Angola, India, Pakistan, Turkmenistan, Uzbekistan, Kazakhstan, China, Mongolia, India, Thailand, Australia.

RC: Limitation of wind expansion (P10-L83): you state a limit of 1.5 MW/km² for onshore wind expansion, but your citation (Our World in Data) does not directly justify this figure. How was this value obtained, and what is its impact on your results?

AR: The citation referred to the first part of the sentence on the historical growth of wind energy capacity. It shows that for many European countries, the growth has been slower in the past few years than in the past (e.g. Germany, United Kingdom, Denmark, Norway).

The value of 1.5 MW/km² is conservative but falls into the range of values observed in the literature for land-use requirements of onshore wind that include the land for spacing between wind turbines, according to a review by Turkovska et al. [5], which we now reference. We choose a conservative value so that some of the eligible land is reserved for wind replenishment and to avoid too highly concentrated wind capacity expansion in individual regions for which there would likely be no public acceptance. This assumption in combination with our land-eligibility criteria leads, for example, to a 244 GW potential for Germany, which is still higher than the current government's 160 GW target by 2045 (<https://openenergytracker.org/en/docs/germany/electricity/#wind-energy>).

In a previous paper (Neumann et al. [6], Section S13.3, Figure S21), we investigated the impact of constraining onshore wind expansion from a base value of 3 MW/km² (100%). This analysis was done for a domestic scenario with no imports, which places more pressure on onshore wind potentials. The cost premium of reducing onshore wind potentials to 50% (1.5 MW/km²) was limited to less than 2% of the total system cost. Therefore, we deem the impact of this assumption on our results to be minor, while it aligns more closely to officially stated targets.

RC: Fuels in hard to abate sectors (P10-L106): several fuel combinations are allowed for these technologies, and while they make broad sense to someone in the field of energy systems, they lack any sort of citation in the text. Ditto for minimum part load constraints for Fischer-Tropsch, methanolisation and methanation facilities.

AR: We have revised this paragraph to include references to justify our assumptions about transport sector fuels and the operational flexibility of fuel synthesis units. We provide further explanations in your corresponding specific questions in the "Additional detailed comments" section.

RC: $N - 1$ resilience (P11-L5): you state that a 70% limit on transmission line utilization is enough to cover $N-1$ criteria. But, again, no Is there a citation or further explanation is provided by the authors. How is this assumption justified for this choice?

AR: The heuristic of loading transmission lines no more than 70% of their nominal rating is commonly applied by transmission system operators in Europe and circumvents the computationally very demanding formulation of $N - 1$ constraints. This assumption has been scrutinized in Gazafroudi et al. [7] while investigating topology-based approximations of $N - 1$ contingency constraints in Germany, and was found to be a satisfactory approximation for the $N - 1$ criterion without computational overhead. We now refer to this paper in the manuscript.

RC: Demand projections for exporting countries (P11-L39): no context is given on the projected future

energy demands for these nations. Many of the countries included are developing economies, meaning internal demand is expected to increase, which in turn would affect your export prices. This can have major impacts on the results of the study, since you assume local demands are met first before exportation. Can you give more context on how future local energy demand was projected?

AR: We use the GEGIS tool [8] to create projections about future local electricity demand for each export region. The tool utilises machine learning on historic time-series, weather data and macroeconomic factors to create artificial electricity demand time-series based on population and GDP growth scenarios following the SSP2 scenario of the *Shared Socioeconomic Pathways* [9]. From these time-series we take the annual demand and increase it by a factor of two to account for further electrification of other sectors – an aspect not reflected in GEGIS.

These projections are meant to serve as rough estimates, as we were unable to find other suitable demand estimates with global coverage. While rough, we believe this to be of sufficient quality for our purpose. As shown for a few selected exporting regions in Supplementary Fig. 11, often only the first few hundred TWh of electricity have exceptionally low levelised cost, followed mostly by an extensive flat part in the LCOE curve. We think it is crucial to reserve the limited *lowest-cost* resources for domestic supply, but given that *low-cost* potentials are usually extensive on the LCOE curve the exact value for the domestic demand seems not as relevant and some uncertainty seems acceptable given the limited data availability.

RC: Underground storage (P11-L102): you state that underground hydrogen storage was excluded from the TRACE analysis due to limited availability in regions with high renewable potential. Yet your citation does not fully support that statement: the presentation you refer to shows that most countries included in your study (Eastern US, China, Argentina, Chile, MENA, Turkey, Kazakhstan, Canada) have salt cavern deposits, and it has no analysis on whether or not these caverns are located in regions with low renewable potential. Besides, you assume that the best regions will be used for internal demand, which also runs contrary to this statement. How can this assumption be justified?

AR: We acknowledge that our initial submission was not very clear about this. While the geographical locations of salt deposits shown in the presentation by Hévin [10] may be indicative of underground hydrogen storage potential, it is unclear which of these deposits have the suitable geological conditions for it. We now cite an additional report from the IEA Hydrogen TCP [11] which describes a variety of hurdles for scoping underground hydrogen storage potentials. While more or less comprehensive assessments exist for some regions like Europe [12, 13], South Asia [14], Australia [15], China [16] or the United States [17], we are not aware of an adequate global dataset.

As a conservative estimate, we therefore default to the assumption that underground hydrogen storage is not available in the exporting regions. We have rephrased our statement on “limited availability” to “uncertain availability”. Moreover, the availability of geological hydrogen storage is one sensitivity we explore in Table 1 for Fischer-Tropsch fuel imports from Southern Argentina. If geological storage is available, import unit costs are reduced by 4%, by reducing the need for flexible

power-to-X operation. Given this limited effect and our consideration of import cost variations in a range of $\pm 50\%$, we believe that neglecting underground hydrogen storage in potential exporting regions does not substantially impact our key results.

RC: 3. Framing of the relocation of generation capacity

I would like to express some concerns in regards to the framing of the study, particularly in regards to the limited discussion of resistance by local communities to expansion of renewable projects in exporting countries, and within Europe to the displacement of heavy industries. While I understand that these are not within the scope of the methods of the study, I believe they are definitely within the scope of the discussion, particularly when local resistance within European nations is one of the drivers for the relocation of production facilities abroad (the authors have even constrained the model's ability to deploy onshore wind generation due to local resistance), and due to the overall concerns of extractivism in developing nations in the context of the energy transition.

At the very least, I would expect the limitations you mention when it comes to these topics to be somewhere in the main text, not relegated to the S.I. This goes beyond social concerns: these are prime sources of uncertainty in your study that are not addressed by your methods. Proper discussion is, in my opinion, necessary.

AR: We fully agree that non-cost decision factors need to be evaluated with care, for both importing and exporting regions. We tried to highlight that in our concluding remarks in the previous version, but have now expanded our discussion of it – for instance, by moving the discussion of limitations to the concluding section.

However, we would not say that local resistance within Europe is the main driver for the relocation of steel and ammonia production we observe. It is rather the difference in renewable resource quality and relocation of energy-intensive production would be observed even with more optimistic land availability for onshore wind expansion in Europe. Moreover, we would like to highlight that not all exporting regions we consider are developing, considering that Australia, Chile, Argentina, Canada and the USA are among the selected regions.

RC: Additional detailed comments:

Some of the figures featured in the study are set up in ways that may confuse readers. Splitting them, or at least re-arranging them, would significantly improve readability.

AR: We outline the changes made to the figures in the following responses.

RC: Figure 1: please consider inverting the choropleth so that its usage is consistent throughout the article. This figure misleads readers into thinking that Northern countries have the lowest hydrogen import costs, or that Argentina and Chile are the most expensive. See figure 7.

AR: We have changed the colormap to 'viridis', which is now consistently used for costs on maps.

RC: Figure 4: please consider separating this into two different figures, or rearranging so that the map and price data are outside of the bar plot. The current configuration obstructs analysis more than aiding it. Similarly, I would suggest inverting the choropleth to match the setup used in figure 7.

AR: We have simplified Supplementary Fig. 4. First, we have removed the time series inset axis and relegated these time series plots to a standalone Supplementary Fig. 22. Second, we have rearranged the figure such that the map is to the side of the bar plots. Third, we use the same colormap, ‘*viridis*’, as in Figure 1a.

RC: Figure 6: as it stands, readers might be misled to believe that the relation between import/export does not change between scenarios. Although I understand that only the base scenario was kept for readability, there is nothing to indicate that the dashed line also changes significantly between these scenarios. The relation between domestic imports and exports is of core importance to the statements provided by the study. I suggest displaying this more clearly.

AR: We have split the figure into two subplots where the left panel shows the variation of system cost and the import/export cost relation for different scenarios, and the right panel shows the system cost composition for the base scenario. Versions of this figure for other import cost scenarios are presented in Supplementary Figs. 14 to 17.

RC: Figure 7: the sizing of technologies is quite difficult to ascertain due to the limited space given for the six figures. At the moment it is difficult to discern the direction of hydrogen transmission. Similarly, the Bay of Biscay data in the center-left should be given its own space to increase clarity.

AR: We have made the following changes to improve readability:

- We have zoomed in on the map.
- We have adjusted the size of the pie charts, arrows and pipeline widths.
- We have tweaked some of the most similar colors to be more distinct.
- We have improved the labelling of the half-circle in the Gulf of Biscay, which represents the imports of hydrogen derivatives that are not spatially resolved in the model.

RC: P1-L12: Suggest to replace “throttled” with “constrained” for clarity.

AR: Done.

RC: P2-L50: please clearly state that PyPSA-Eur only optimises for a single year here, and clearly state that the model has perfect foresight for technology operation.

AR: We now state in the introduction that no pathways are modelled in our overnight scenarios and a single weather year is used. We provide more background on the choice of weather year later in the Methods section, where we also state that the model has perfect operational foresight.

RC: P2-L99: consider removing the clarification that Argentina and Chile are the lowest-cost suppliers

for Europe to another section. This appears to be result from your exercise, not introductory context. Unless you mean to say that only these countries were considered for hydrogen imports (which does not follow from your methods section).

AR: We have removed this sentence from said section.

RC: Figure 3: import mix is missing the added sum of total supply (lower left).

AR: We have added the total sum of imports to the figure.

RC: P6-L7: “imported fuel fuels appear to be...”

AR: Done.

RC: Figure 6: “Cost alterations are uniformly applied to all imports options import options but direct electricity...”

AR: Done.

RC: Figure 7: to further improve readability, consider moving the technology legend to the bottom of the figure, this would provide more space for the maps without additional confusion.

AR: We have experimented with alternative legend positions, but placing the legend to the side fits presenting three rows of scenarios on one page best. Instead, we have revised several graphic elements as outlined above.

RC: P10-L53: please consider providing further justification of why this particular year was chosen to test the system.

AR: Figure 1 of Gøtske et al. [18] shows a comparison of capacity expansion results of PyPSA-Eur for 60 different weather years, and in that study the weather year 2013 was fairly average in terms of wind and solar availability and ambient temperatures. We have added this reference to put our choice of weather year into context. We acknowledge that, if it had been computationally viable, using multiple weather years would have been preferable.

RC: P10-L68: please consider explaining the effects that this has on the study in the limitations (e.g., underestimating the system cost of negative emission technologies).

AR: The comment relates to the lack of spatial resolution for CO₂. The consequence of this model simplification is that transport of CO₂ within Europe is unconstrained, i.e. the potential need to build CO₂ pipelines to usage or sequestration sites is neglected. This means that the location of biogenic or industrial point sources of CO₂ is not a siting factor that this model version considers for PtX processes. Consequently, with limited waste heat usage potential, the model will place fuel synthesis plants where H₂ is most cheaply available (e.g. European periphery), implicitly assuming that the CO₂ would be transported there.

We acknowledge that this is a model limitation and now mention its implications more clearly in

the discussion section, pointing to work by Hofmann et al. [19], which looks at the complementarity of hydrogen and carbon dioxide transport in Europe in more spatial detail. In their scenario with the highest CO₂ network cost, namely when no hydrogen can be transported, the CO₂ network cost is no higher than 3 bn€/a (0.4% of total system cost).

RC: P10-L106: please consider providing proper citations to justify the assumptions for shipping, heavy-duty vehicles and aviation.

AR: For domestic and international shipping, we assume methanol for propulsion as it has a higher technology-readiness level (TRL9: Commercial operation in relevant environment) compared to ammonia (TRL6: Full prototype at scale) or hydrogen (TRL4-5: Early prototype, Large prototype) according to the IEA ETP Clean Energy Technology Guide [20] and IRENA [21]. Among green shipping initiatives in a report by the Global Maritime Forum, most consider methanol as energy source [22] and the shipping company Maersk has already deployed a methanol-powered container vessel in 2023 [23]. Recent research also points to the economic viability of methanol as a marine fuel compared to ammonia and other alternative fuels [24].

For heavy-duty vehicles, we base our modelling decision to assume full electrification on Link et al. [25]: “Our TCO [total cost of ownership] indicates that BETs [battery electric trucks] may constitute the most cost-effective pathway in reaching TCO parity with less policy support needed, in contrast to FCETs [fuel cell electric trucks], which might require more policy support throughout the 2030s.” Other studies, like Nykvist and Olsson [26] and Phadke et al. [27], also paint a positive outlook on the role of battery electric trucks for long-haul heavy-duty road transport.

For aviation, we assume synthetic kerosene for propulsion because of the low TRL of other propulsion systems. The IEA ETP Clean Energy Technology Guide lists a TRL of 6-7 for hydrogen fuel cell propulsion and a TRL of 4-5 for battery electric planes [20].

We now refer to a selection of these references in the revised manuscript.

RC: P10-L115: please consider providing a citation for the operational restrictions of Fischer-Tropsch, methanolisation and methanation.

AR: This information was first obtained for a study by Brown and Hampp [28] from a confidential source with firsthand operating experience of a wind-based methanol synthesis plant in Chile. They suggested that realistic settings were minimum part load of 20-30% and higher values for Fischer-Tropsch synthesis. Recent literature also suggests that a minimum part load of 10-15% could be achieved for methanol synthesis [29, 30], whereas the operating ranges for Fischer-Tropsch synthesis seem to be more narrow [31, 30]. For methanation, Mucci et al. [32] suggest a minimum load range of 40%, while considering a minimum load range of 20% for methanolisation.

We acknowledge, that there is some uncertainty in these values, but we feel the literature supports our assumption that methanol synthesis is more flexible than other hydrogen-to-X processes. A fitting further reference that explores the impact of flexible methanol production is Svitnič and Sundmacher [33].

RC: P11-L15: please provide a citation for the costs of H2-DRI and EAF.

AR: For these two technologies, we rely on assumptions from the Steel Sector Transition Strategy Model (ST-STSM) of the Mission Possible Partnership (<https://github.com/missionpossiblepartnership/mpp-steel-model>), which was used for the “Making Net-Zero Steel Possible” report [34]. For values, directions to particular files of the source repository and documentation of unit conversions, see the entries ‘direct iron reduction’ and ‘electric arc furnace’ in our Github repository for technology data (<https://github.com/PyPSA/technology-data/releases/tag/import-benefits-v2>).

We have added the ST-STSM and the report as a reference in the manuscript.

RC: P11-L117: this entire paragraph appears to belong to the results section of the study and has a broken citation. Was it misplaced?

AR: We have moved this paragraph to the section where Figure 1 is discussed, which outlines the calculations of global import vectors. The broken citation is now resolved.

RC: S.I.: please consider providing a full table with your cost assumptions and a citation for them.

AR: We now provide the data on techno-economic assumptions in machine-readable format on Github at <https://github.com/PyPSA/technology-data/releases/tag/import-benefits-v2>. The link is versioned and specific to the manuscript version. The same data is used in PyPSA-Eur and TRACE.

RC: The code is available and has appropriate documentation, but I did not attempt to install and run it.

AR: Thank you.

Reviewer #2

RC: I co-reviewed this manuscript with one of the reviewers who provided the listed reports. This is part of the Nature Communications initiative to facilitate training in peer review and to provide appropriate recognition for Early Career Researchers who co-review manuscripts.

I’ve reviewed the code (mostly the snakemake workflow provided in the linked repository) and some of the data used. Things to note:

1. Salt cavern potentials in the model appear to be at a national level. It’s unclear how this is translated into the sub-regions used by the model (100+ nodes).

AR: The data we use for salt caverns was programmatically extracted from Caglayan et al. [13]. Several graphics in this report are regionally resolved – see `data/bundle/h2_salt_caverns_GWh_per_sqkm.geojson` – which allows proportional mapping of the reported country totals to the model’s sub regions.

These figures have been added as Supplementary Fig. 37.

RC: 2. Generally there are not a lot of notes on how or where the data is sourced from. This might not be an issue, but it is difficult to trace these processes without further notes from the authors.

AR: We agree that the data documentation could be improved. In response to your comment, we have added a dedicated section in the PyPSA-Eur documentation listing the source, link and license of each dataset used (https://pypsa-eur.readthedocs.io/en/latest/data_sources.html). We now refer to the documentation in our data and code availability statement.

Reviewer #3

RC: This is a nice article addressing a very relevant topic. It is, however, difficult to understand the results without a better description of the data assumptions and modelling choices. In general, the conclusions should be substantiated by results, which it should be able to understand based on the descriptions of data and modelling. This could be improved.

AR: We agree that the methods section could benefit from some more detail. In the revised manuscript, we have improved the description of the data and modelling assumptions, and believe our discussion points are now either substantiated by the results or clearly marked as speculative. We outline the specific measures taken in the responses to your comments below.

RC: Some questions which the reader is left with are:

How are the demands modelled? What can the different fuels be used for? Which are the

competing supply options? And how are the demands distributed across Europe?

AR: For an overview about what different fuels can be used for and what competing fuel pathways are considered, we have created a carrier-specific flow chart of supply, withdrawal and balancing options, which is now shown in Supplementary Fig. 2.

We make some exogenous decisions about which carriers supply certain demands, such as using electricity in road transport, methane for high-temperature industrial heat, solid biomass for most low-temperature heat, methanol for shipping, diesel for agricultural machinery, and kerosene for aviation. Some demands are associated with a time-varying profiles (i.e. residential/services electricity, electric vehicles, heating demand) based on travel patterns or ambient weather conditions, while the other exogenous demands are assumed to be time-constant (e.g. kerosene, naphtha, methanol, ammonia, industry electricity). Furthermore, some demands are spatially fixed, e.g. residential/industry/services/agriculture electricity, heat, electric vehicles, solid biomass for industry, and naphtha/methanol feedstocks. This also includes hydrogen demand for steel and ammonia production, unless these are allowed to relocate.

Because the siting and the operation of most conversion units are endogenously optimised by the model, there are many spatially-variable demands (e.g. electricity demand for electrolyzers or the hydrogen demand for methanolisation). In fact, when steel and ammonia production is allowed to relocate, there is only minimal spatially-fixed hydrogen consumption/production (≈ 5 TWh net production due to chlor-alkali electrolysis from chlorine production). We now show the spatially-fixed demands for different fuels in Supplementary Fig. 4 as well as the exogenous total final energy and non-energy demands in Supplementary Fig. 5.

RC: Which options compete with hydrogen for providing backup power and heating?

AR: The model includes a variety of options for providing backup power and heating, which compete with hydrogen turbines and fuel cell CHPs (cf. Supplementary Fig. 2). For backup power, the model considers gas and methanol turbines. For backup heat, the model considers gas boilers and resistive heaters. For combined backup heat and power, the model considers biomass and gas CHPs. Furthermore, flexible demands like electric vehicles, heat pumps and fuel synthesis units, as well as batteries and thermal storage in district heating, can reduce the need for backup capacities. In fact, we observe that as some flexible domestic power-to-X is displaced by imported fuels domestic backup power capacities increase from 129 GW (no imports) to 276 GW (all imports allowed). We have amended this description in the manuscript.

RC: Which fuel pathways are modelled and how? Are "blue" options taken into account?

AR: Yes, we consider "blue" options such as steam reforming with carbon capture for hydrogen. The capture rate is assumed to be 90% for all possible carbon capture applications, which include steam methane reforming, gas/biomass CHPs, in biogas upgrading and process emissions. We include the investment cost, efficiencies and added energy consumption of the carbon capture units. The possible fuel pathways can now be inferred from the carrier-specific supply and consumption options listed in Supplementary Fig. 2.

RC: DAC seems to be cheap. Which cost assumptions are used for the different technologies?

AR: We now provide the data on techno-economic assumptions in machine-readable format on Github at <https://github.com/PyPSA/technology-data/releases/tag/import-benefits-v2>. The link is versioned and specific to the manuscript version. The same data is used in PyPSA-Eur and TRACE.

For DAC, we use techno-economic assumptions listed in the technology catalogues published by the Danish Energy Agency (DEA). For the results shown, we used use a version from November 2021 (https://ens.dk/sites/ens.dk/files/Analyser/data_sheets_for_carbon_capture_transport_storage_-_version_1.xlsx) which assumes 5 M€/t/h for CAPEX in 2040 based on Fasihi et al. [35]. The new version of the same dataset released in April 2024 goes down to 1.9 M€/t/h for CAPEX in 2040 (https://ens.dk/sites/ens.dk/files/Analyser/technology_data_for_carbon_capture_transport_storage.xlsx) Here, the DEA lists Ozkan et al. [36] as the data source. However, we deem the first version more reliable, as the second seems to list prices as advertised by producers after subsidies have been applied. The cost for electricity and heat demand are endogenously determined in additiona to the CAPEX and maintenance costs.

RC: What do Fischer-Tropsch fuels entail? For which purposes can they be used? And how are the costs and competitions modelled?

AR: Fischer-Tropsch fuels are synthetic liquid hydrocarbons produced by catalytically converting carbon monoxide and hydrogen (synthesis gas) into various liquid fuels and chemicals. For computational reasons, these are not further resolved in PyPSA-Eur. In the model, Fischer-Tropsch fuels can be used to satisfy demand for kerosene for aviation, diesel in the agricultural sector and feedstocks in the high-value chemicals industry. Investments and operation of Fischer-Tropsch synthesis plants are also optimized in the model according to the techno-economic assumptions referenced above. These synthesis plants compete with plants for the methanol-to-kerosene routes, respectively, while the total demand for kerosene is exogenously set. The feedstock demand for high-value chemicals production is also exogenously set.

RC: What is assumed regarding the potentials for solar and wind power in EU and globally? Which biomass types have been investigated and what are their assumed uses and potentials in EU - and similarly for biogenic CO₂?

AR: For wind and solar potentials in PyPSA-Eur, we calculate capacity limits per region and technology based on land eligibility criteria considering land type, natural protection areas, topography/bathymetry and distances to settlements, and assumed maximum deployment densities per technology. The renewable generation potential for the available area is then evaluated based on reanalysis weather data (ERA5 [37]) and satellite observations for solar irradiation (SARAH-3, [38]) in combination with solar panel and wind turbine models. We refer to Neumann et al. [6] (Section S6) for more details. The process is similar in the TRACE model, but partially using different data sources that have global coverage. Because TRACE looks at individual countries, the wind and solar potentials are additionally split into a number of resource classes (30 each for onshore wind and solar, 10 for offshore wind), whereas in PyPSA-Eur each of the 115 regions

receives on renewable time series and capacity potential per technology. We refer to Hampp et al. [39] for more details.

For biomass, we assume a total domestic potential of 1,372 TWh/a, which splits into 358 TWh/a for biogas and 1,014 TWh/a for solid biomass. This potential is based on the NUTS2-level medium biomass potentials from the JRC-ENSPRESO database [40]. Here, we only consider biomass residues and no energy crops or biomass imports. The solid biomass potential includes agricultural waste, fuelwood residues, secondary forestry residues (woodchips), sawdust, and residues from landscape care. The biogas potential encompasses manure (solid and liquid) and sludge. Biogas can be upgraded to methane with or without capturing the CO₂. It can then be used as a substitute for synthetic or fossil gas for high-temperature industrial heat, for gas boilers, for CHPs, for gas power plants or for reforming to hydrogen. Solid biomass can be used for heat in biomass boilers and biomass CHPs with or without carbon capture, low-temperature industrial heat, and for producing electrobiofuels with subsequent use as aviation fuel or chemical feedstock. Further options like biomass-to-liquid, biomass-to-methanol, biomass-to-SNG have been disabled after an initial screening to reduce computational complexity. Any captured biogenic CO₂ can be used as an input to Fischer-Tropsch, methanol or methane synthesis, or sequestered for negative emissions. We refer to Millinger et al. [41] for an in-depth analysis – performed with PyPSA-Eur – of biomass usage pathways in Europe and more detailed representation of biomass technologies.

RC: How is the variability of solar and wind power taken into account for the imported fuels? Do the cost estimates include potential required oversizing and batteries? The description of the modeling of green fuels in the TRACE model should be improved and similarly, and so should the linking of the TRACE model with the PyPSA model.

AR: For the imported fuels,¹ we calculate a levelized cost for an export volume of 500 TWh separately for each country and carrier combination. This cost includes the generation, storage, conversion and transport costs of the fuel supply chain to Europe. Within the confines of the renewable potentials, the model can optimise the size, curtailment and intermediate storage in batteries or hydrogen. So, yes, if economic, the model might oversize components or build storage to smooth feed-in profiles. Since TRACE looks at whole countries or large regions within them without further spatial resolution, the renewable potentials and profiles are split into different resources classes to reduce smoothing effects (30 each for onshore wind and solar, 10 for offshore wind). Furthermore, while capable of hourly modelling, the temporal resolution of the model is set to a 3-hourly resolution for the year 2013 for consistency with the multi-hourly resolution used in PyPSA-Eur (i.e. similar balancing requirements).

We have amended the manuscript with a more detailed description of the modelling in TRACE and the linking between TRACE and PyPSA-Eur.

RC: It is concluded, that PtX flexibility has a benefit in terms of integrating renewables, however, no results are shown with respect to this - and the modeling of it is not clearly described

¹Apart from electricity imports, for which generation, storage and transmission capacities and operation are endogenously optimised in the PyPSA-Eur model.

AR: We have run this case as an another sensitivity analysis, where all domestic power-to-X processes must run inflexibly at full capacity (incl. electrolysis, methanolisation, Fischer-Tropsch, methanation, ammonia synthesis). With inflexible PtX, costs rise by 8.7% without imports and by 1.5% with imports allowed. Inflexible PtX makes domestic more expensive and system shifts towards more imports, increasing the attainable system cost reduction through imports from 4.3% to 10.7%. It should be noted that fully inflexible PtX is an extreme restriction and the sensitivity analysis represents rather a thought experiment to quantify the value of PtX flexibility. We now discuss the role of PtX flexibility in more detail in the section “Interactions of import strategy & domestic infrastructure”.

RC: An important assumption appears to be the flexibility of different synthesis processes. There is however no reference (just a question mark) for the assumption and as this is debated it could merit a sensitivity analysis what the impact of the assumption is on results

AR: We apologize that the reference to Brown et al. [28] was not correctly resolved in the initial submission, which presents sensitivities around the operational flexibility of methanol synthesis plants. We have added further references pointing to the potential flexibility of different fuel synthesis plants. Recent literature suggests that a minimum part load of 10-15% could be achieved perspectivevely for methanol synthesis [29, 30], whereas the operating ranges considered for Fischer-Tropsch synthesis seem to be more narrow [31, 30]. For methanation, Mucci et al. [32] suggest a minimum load range of 40%, while considering a minimum load range of 20% for methanolisation.

We acknowledge, that there is some uncertainty in the exact values, but we feel the literature supports our assumption that methanol synthesis is more flexible than other hydrogen-to-X processes. However, given the uncertainty, we have narrowed the difference in minimum part loads based on reviewing the literature above, using 20% for methanol and 50% for other hydrogen-to-X processes.

The lower operational flexibility makes Fischer-Tropsch fuels slightly more expensive than methanol in the exporting countries, as it is quite beneficial in those islanded systems. Without it, more alternative balancing approaches need to be deployed. We show in Table 1, which was moved to the main body, that reducing the minimum part load of Fischer-Tropsch synthesis plants from 50% to 20% reduces levelised cost for imports from Southern Argentina by 3.6 €/MWh (2.8%) to 124.9 €/MWh, which is then closer to the 125.3 €/MWh we see for methanol.

RC: Are the imported fuels assumed available e.g. in ports at a constant rate - or when the demand is there? Do the investments in infrastructure include bunkering and storage investments in ports? Which ports are considered?

AR: For import by hydrogen pipelines, we now assume a minimum part load of 90% to align with the high pipeline utilisation rates observed in the TRACE model, resulting in near constant hydrogen imports by pipeline. For electricity, the investment and utilisation rate of HVDC links is endogenously chosen by the model. For imports of hydrogen or methane by ship, the utilisation rate of terminals is not constrained by shipment schedules. However, the model can choose to

use existing LNG terminals for methane or invest in new terminals for methane or hydrogen imports up to twice today's capacity for LNG imports. Thus, low utilisation rates come at the expense of additional infrastructure cost for the same import volumes. For the approximate costs and operational capacities of the terminals, we use data from the Global Energy Monitor's "Europe Gas Tracker" [42]. All other carriers considered for imports are not spatially resolved for computational reasons as their transport costs are very low. For these carriers, we make no specification about which ports are used. We assume import capacities to be unconstrained because especially carbonaceous fuels are similar to fossil oil which is handled at European ports today in large quantities. We account for the storage cost of all fuels, except for steel and HBI, which we assume to be low.

RC: It is surprising that the MENA region including Morocco doesn't come out better. What are the assumed costs of shipping the fuels e.g. from Chile?

AR: We gladly provide a little more background on the assumed shipping costs. For methanol, the shipping cost from Chile adds up to 1.8 €/MWh compared to a levelized cost of 124.9 €/MWh. The transport cost is calculated for a methanol transport ship with a loading capacity of 75,000 t (415,208 MWh) and an assumed energy demand of 0.2428 MWh/km, as derived from Hurskainen [43]. The economic parameters for the ship, based on Runge et al. [44], are: investment costs of 35 mn€, fixed operation and maintenance costs of 1.75 mn€/a, lifetime of 15 years and discount rate of 7%. The sea distance from Chile to the closest port in Europe (Sines LNG terminal, Portugal) is 12,778 km based on the *searoute* Python package (<https://github.com/genthalili/searoute-py>). For the outward and return journey, we assume that the fuel cost is the same as the levelised cost of the fuel it carries. Shipping parameters for other fuels can be found in Hampp et al. [39] (Table S5). Further, techno-economic parameters are included in the technology database used for this study (<https://github.com/PyPSA/technology-data/releases/tag/import-benefits-v2>).

Regarding Morocco: In the backend, we changed the source of the country shapes to GADM (<https://gadm.org/>), which represents the territory of Western Sahara separately from Morocco. A combination of excellent solar and wind resources leads to low cost in that region comparable to those for green methanol from Chile (Supplementary Fig. 11).

RC: Is the same WACC used for all countries?

AR: Yes, a uniform WACC of 7% was used for all countries. We acknowledge that projects in different countries could incur different WACC [45, 46], and show in Table 1, which was moved to the main body, to what extent changed WACC affect the import costs. However, we choose a uniform WACC because current differences may not persist in the long-term and are highly uncertain [47]. For our long-term view, assuming uniform financing costs may therefore be more appropriate [48]. Moreover, international development banks (e.g. European hydrogen bank, https://energy.ec.europa.eu/topics/energy-systems-integration/hydrogen/european-hydrogen-bank_en) might finance renewable energy projects abroad and, thereby, reduce the risk and cost of capital through sovereign backing. In this case, the WACC is more accurately determined by a company or project-specific level but subject to changes in available national and international financing instruments.

Therefore, we have chosen to use uniform WACC assumptions instead to enhance comparability of results and to focus on the natural resource endowments of each country.

RC: How is the relocation of industries modelled?

AR: Without relocation of steel and ammonia production allowed, the production volumes of primary steel (DRI+EAF) and ammonia (Haber-Bosch) are spatially-fixed, resulting in exogenous hydrogen demand per region. For the spatial distribution, we use data from Global Energy Monitor's Global Steel Plant Tracker [49] and manually collected data on the few ammonia plants in Europe (https://github.com/PyPSA/pypsa-eur/blob/master/data/ammonia_plants.csv). With allowed relocation, the model endogenously chooses the production volumes of primary steel/HBI per region, subject to the availability of cheap hydrogen. Secondary/recycled steel production in electric arc furnaces remains in proportion at current locations in either case.

In both cases, the model can choose the operation mode and capacity of Haber-Bosch, direct iron reduction (DRI) plants and electric arc furnaces (EAF) individually, considering investment costs and efficiencies taken from the Steel Sector Transition Strategy Model (ST-STSM) of the Mission Possible Partnership (<https://github.com/missionpossiblepartnership/mpp-steel-model>). Steel and HBI can be stored and transported without constraints within Europe.

We have expanded the description of the relocation of industries in the manuscript. We have also added a series of maps showing the relocation patterns observed in Supplementary Fig. 6.

RC: Did you take the potential for repurposing infrastructure into account?

AR: In the previous version, we did not include the retrofitting of pipelines like in Neumann et al. [6]. However, we have now reactivated this option, so that gas pipelines can be retrofitted to hydrogen pipelines at a cost that is approximately half that of a new hydrogen pipeline.

For other infrastructure, we did not include any repurposing potentials. For instance, it is not assumed that existing LNG terminal capacities can be retrofitted for hydrogen imports. Neither are existing gas power plants assumed to be converted to allow hydrogen combustion.

RC: Did you check whether the implemented pipeline and cavern storage sizes are realistic?

AR: We acknowledge that neglecting the discrete dimensioning of pipelines is a limitation, but a computational necessity for a capacity expansion model with both spatial and temporal resolution. Assuming minimum discrete unit sizes of 20-inch pipelines with a capacity of 1.2 GW according to a European Hydrogen Backbone publication [50], rounding up all fractional capacities to multiples of this unit size would increase the network volume by approximately 60%, ensuring technical feasibility. Despite potentially underestimated costs and neglecting the discreteness of repurposing gas pipelines, the overall optimised hydrogen network is much smaller than, for instance, outlined in the European Hydrogen Backbone plans [50] or the German hydrogen core network (<https://fnb-gas.de/en/hydrogen-core-network/>). This is due to our consideration of relocation of steel and ammonia production, full road transport electrification, CO₂ transport

to CCU locations, and other backup heat/power capacities than hydrogen. We therefore do not believe that our overall modelling results are significantly impacted by neglecting discrete pipeline sizing.

The geological hydrogen storage potentials in Europe, we take from an external source, Caglayan et al. [13]. In our domestic scenario, the model builds a total underground hydrogen storage capacity of 15 TWh, which only uses a fraction of the technical potential in Caglayan et al. [13] and also falls within more conservative recent estimates by Talukdar et al. [12]. The spatial distribution can be seen in Figure 7. A test site for cavern storage of hydrogen in Rüdersdorf, near Berlin, currently operates with a capacity of 6 t (200 MWh) (<https://www.ewe.com/de/media-center/pressemitteilungen/2023/10/wasserstoff-speicher-rdersdorf-ewe-lagert-erstmal-wasserstoff-ein-ewe-ag>). There are also operational hydrogen storage projects in the United States with volumes of 500,000-1,000,000 m³ [51]. TWh-scale storage would be implemented with a series of separate caverns in a salt dome [52]. Thus, we believe the cavern storage sizes to be scalable.

RC: The main sensitivity is on +/- 20% changes in imported fuel costs. This range seems narrow compared to the inherent uncertainties and is poorly argued.

AR: We were convinced and have expanded the range of uncertainty considered to $\pm 50\%$. We believe that this range is sufficient, considering that beyond 30% higher import costs the model no longer imports energy and below 50% lower import costs, the costs for the domestic energy system are only marginally reduced (Figure 4). In Table 1, which was moved to the main body, we also showcase what can cause such cost variations. It is also important to note that the cost variation is relative to the domestic production costs and does not reflect the overall uncertainty of levelised costs and the contributing technologies.

RC: The article is well written and with some very nice illustrations (e.g. Fig 2), although some figures are packed with information and rather confusing (e.g. Fig 7) and line 58-62 is difficult to understand. Validation should be added in terms of comparing with the results of others - both in terms of global green fuel prices - and of future European energy supply and infrastructures.

AR: Thank you for your comment. We have zoomed in on the maps of Figure 7 and adjusted the marker sizes to improve readability. We were unsure which page the comment on line 58-62 refers to and skipped this comment, but hope that it was clarified with out extensive revisions.

We have also amended the discussion/conclusion section with a paragraph comparing our results to a selection of other studies looking at the relation between energy imports and the European energy system: Kountouris et al. [53], Wetzal et al. [54], Frischmuth et al. [55], Schmitz et al. [56]. We also relate our import costs to the meta-study by Genge et al. [57].

Reviewer #4

RC: Summary

The current analysis tries to assess the impact of importing energy vectors (e.g., hydrogen, methanol, ammonia, and steel) on the future of European electricity and hydrogen infrastructure. The research utilizes PyPSA-Eur, a large-scale open-source energy system model assuming perfect knowledge and competition. The Trace model, which is soft-linked to the previously mentioned European energy system model, is used to quantify the costs of imported energy vectors. The study reveals interesting findings and suggests that imports could reduce system costs by 1-14%, with hydrogen and its derivatives providing the greatest benefits. In addition, the current study challenges how electricity and hydrogen networks should be developed in conjunction with potential energy commodity imports.

Overall comments and questions

From an energy system modeling perspective, there are questions regarding the novelty of the model approach to model imports.

1) Should electricity imports be modeled exogenously with a constant price similar to the other commodities? Accounting for electricity endogenously yields a 4-hourly opportunity cost (i.e., endogenously estimated based on renewable availability in third countries) as well as the costs of establishing electricity transmission lines compared to importing hydrogen or other commodities. In other words, power imports are accounted for as the total of CAPEX and variable OPEX, whereas other commodities are imported as OPEX based. Is this a fair comparison? For example, since you have represented power imports via transmission lines, could hydrogen pipelines not be modeled?

AR: We believe electricity imports require a different modeling approach compared to other fuels because electricity is difficult and costly to store, leading to highly variable prices that are more sensitive to the hourly renewable energy availability. Chemical fuels or materials that are easy to store do not exhibit this price variability. To be able to reasonably assume a constant price for electricity imports, the TRACE model would have to build a substantial amount of storage in the exporting region, which would drive up costs and disadvantage electricity imports. Relying solely on levelised cost of electricity generation plus transport costs, on the other hand, could unrealistically lead the model to importing electricity in challenging periods, when electricity may also be unavailable in the exporting region. Therefore, we model the import supply chain for electricity endogenously.

The cost of HVDC links for electricity imports is – and must be – considered, just like transport costs (e.g. pipeline costs) are considered as component of the levelised costs for the other carriers. Either way, both CAPEX and OPEX for generation, storage, conversion and transport to Europe

are accounted for. Just the level of endogenous decisions in the European model regarding import supply chains is different (e.g. the sizing and utilisation rate of HVDC links). The reason is that, unlike hydrogen pipelines, the optimised energy transfer in the HVDC links strongly varies with time. Our setup also reflects expected market conditions, in that electricity is traded on an hourly basis, whereas storable, shippable commodities like ammonia or methanol would be contracted with long-term purchase agreements that cover both CAPEX and OPEX.

RC: 2) Can you provide details on the import price estimation and assumptions by displaying them in tables? Supplementary material could be populated with additional data on the input parameters and assumptions on both PyPSA-Eur and Trace.

AR: We now provide the data on techno-economic assumptions in machine-readable format on Github at <https://github.com/PyPSA/technology-data/releases/tag/import-benefits-v2> due to the large number of parameters. The link is versioned and specific to the manuscript version. The same data is used in PyPSA-Eur and TRACE. We refer to it in the “Data and code availability” section.

To show the cost components per import vector and potential exporting country, in Supplementary Figs. 7 to 10 we have now added stacked bar plots illustrating the import cost supply curves. Supplementary Fig. 3 further shows a map similar to Figure 1a for direct hydrogen imports.

Furthermore, we have increased the detail of the model description and coupling of TRACE and PyPSA-Eur and refer to previous studies by Hampp et al. [39] for TRACE and Neumann et al. [6] for PyPSA-Eur for more methodological background.

RC: 3) Additional information about the Trace model is necessary. Is this a simulation or optimization model? Please specify the levelized cost components that the model considers. Are the transportation costs to the European system boundaries accounted for?

AR: As stated above, we have now amended the TRACE model description in the revised manuscript.

To answer the questions in short: TRACE is an optimization model for investment and operation of green fuel production sites. The levelized costs include the costs of electricity generation from wind and solar, direct air capture and iron ore (where applicable), conversion from electricity to hydrogen and further derivatives including liquefaction and regasification (where applicable), batteries and storage of fuels, and transport by ship or pipeline to different entry points in Europe. Values for different cost components are shown, for instance, in Figure 1a and Supplementary Figs. 7 to 10.

RC: 4) It’s not clear how you limit your exogenous imports. Do you have an annual limit per importing point? Do you assume that shipping imports can only occur once or twice per week? Overall, how do you limit imports in terms of time and space? There should be differences between ships and pipelines. Do you assume investment costs for building and importing infrastructure in ports?

AR: The decision about the origin, destination, vector, volume, and timing is endogenous to the PyPSA-Eur model. For exporting regions we limit total fuel exports to 500 TWh. For entry-points to

Europe, the modelling approach depends on the carrier. For ammonia, methanol, Fischer-Tropsch, HBI and steel we assume no capacity limits. For hydrogen by pipeline, imports have to be near-constant, varying between 90-100% of peak imports. For methane by ship, existing LNG terminals can be used and expanded up to double today's capacity considering an annuitized investment cost of 7018 €/MW_{th}/a, based on cost data logged in Global Energy Monitor's "Europe Gas Tracker" for past projects [42]. For hydrogen by ship, terminals can be built in regions where LNG terminals exist at a cost that is 20% higher than what we assume for LNG terminals, but starting with no existing capacity. For electricity, the capacity and operational patterns of the HVDC links can be individually chosen by the model.

Beyond potential capacity constraints of import terminals, we do not assume any constraints in the shipping schedules. We deem this an acceptable inaccuracy, given that the commodities transported by ship can be easily and very cheaply stored and we have observed little to no effect on the results during previous model runs. This is in contrast to static imports of electricity via transmission lines, where the impact of such an assumption would be significant and unconstrained imports would make these options appear to the model like zero-cost electricity storage options. For this reason we have included electricity imports with their exporter's renewables availability time-series at the model's temporal resolution.

RC: 5) Where are the European demands of hydrogen, derivatives, and steel geographically located? Do you account for transportation costs from the imported port node for steel or derivatives to the demand side?

AR: We have now added Supplementary Fig. 4 showing the spatial distribution of spatially-fixed exogenous demands, as well as Supplementary Fig. 6 showing the relocation patterns of steel and ammonia production observed in the model when allowed. We also include a graphic of the total final energy and non-energy demands covered by the model in Supplementary Fig. 5.

Only some demands are spatially fixed, e.g. electricity for residential, industry, services, agriculture; heat; electric vehicles; solid biomass for industry; naphtha/methanol feedstocks. This also includes hydrogen demand for steel and ammonia production, unless these industries are allowed to relocate. Because the siting and the operation of most conversion units are endogenously optimised by the model, there are also many spatially variable demands (e.g. electricity demand for electrolyzers or the hydrogen demand for methanolisation). In fact, when relocation of steel and ammonia production is enabled, virtually all hydrogen consumption is spatially variable.

Regarding transport costs: While we consider the transport costs of steel products and hydrogen derivatives over long distances to the European system boundaries, we do not consider the costs of further transport within Europe. While not spatially resolving dense fuels also comes with computational benefits (transport networks drive model complexity), we believe that it is an acceptable simplification. Relative to the fuel value, transport costs of dense fuels are low. Moreover, already today they are transported across Europe in large quantities from production sites to where they are eventually consumed.

RC: 6) Despite claiming great spatial aggregation, the Balkan countries are only depicted as one node per country (e.g., Fig. 1), even though the most importing choices are neighboring those countries. Is one node per country underestimating the expense of transporting those commodities to Central European countries?

AR: The standard distribution of clusters per country is proportional to the country's energy demand. We have now modified the distribution to ensure that most countries, especially those in South-eastern Europe, have more than one cluster (e.g. Hungary, Romania, Greece, Bulgaria, Serbia, Czechia). We have also slightly increased the spatial resolution from 110 to 115 regions. The size of clusters is now more evenly distributed, except for the remote parts of Scandinavia, which we deem acceptable for the study at hand.

RC: 7) Could you share insights into the potential utilization potential of imported options? The spatial disaggregation yields interesting results. Is there a formation of an importing corridor, such as methanol from Chile to the Netherlands? Where are the majority of hydrogen imports that account for the greatest system cost reduction?

AR: Compared to the initial submission, we have now imposed a limit of 500 TWh per exporting country – for the sum of all carriers – to both prevent a single country from dominating the import mix and match the export volume assumed for the levelised cost calculation in the TRACE model for consistency. This change allows us to include a new Sankey chart, which was added to Figure 3 and shows the origins and destinations of different fuels. The Sankey chart presents an optimised import mix of carbonaceous fuels from Argentina and Chile, as well as steel, hydrogen, ammonia and some electricity from the MENA region. However, we should caution against overinterpreting these country-specific results, as the import cost supply curves, which we now present for different vectors in Supplementary Figs. 7 to 10, are relatively flat. The Sankey charts of further selected scenarios are presented in Supplementary Figs. 19 and 20.

From Figure 1, it seems like the cheapest hydrogen imports come from the MENA region (Algeria) with entry-points in Southern Spain. Figure 7 later shows that the model uses this option, but directly converts it to methanol and Fischer-Tropsch fuel rather than transporting it further inland, assuming that the carbon would be transported there.

RC: 8) The generation expansion results are based on uniform assumptions across European countries, assuming collaboration to attain a net-zero energy system aim. In recent years, we have noticed countries establishing or examining bilateral agreements (for example, Germany with Oman, the United Kingdom, and Canada). However, in the current modeling technique (if understood correctly), Germany's hydrogen demand might be met by pipeline imports from Italy that serve as a landing zone for imports from North Africa. Shouldn't a constrained scenario of network development (electricity and hydrogen) reflect the possibility of reaching those bilateral agreements? Overseas imports are projected to increase to satisfy the regional demand. Yet the total system cost will be bigger than all imported options and network expansion, demonstrating the benefit of having a European dedicated common strategy.

AR: While we appreciate the suggestion, we feel such a scenario would go beyond the scope of our analysis. Although not explicitly modelled, bilateral agreements are somewhat reflected by the option of the model to import carriers in countries where they are consumed. It is also true that the model allows that “Germany’s hydrogen demand might be met by pipeline imports from Italy that serve as a landing zone for imports from North Africa”, but this just reflects current development plans for transnational infrastructure. In November 2023, Germany and Italy signed an action plan which includes plans for building a pipeline between Germany and Italy targeted at imports from North Africa (<https://www.cleanenergywire.org/news/germany-italy-push-new-pipeline-across-alps>, <https://www.south2corridor.net/south2>). Numerous further cross-border hydrogen pipeline projects were included in European Commission’s 6th list of Projects of Common Interest (https://ec.europa.eu/commission/presscorner/detail/en/ip_23_6047, https://ec.europa.eu/energy/infrastructure/transparency_platform/map-viewer/main.html). Therefore, we think it makes sense to include this option.

For a scenario with European energy self-sufficiency, the effects of constrained pan-continental hydrogen or electricity network expansion were also explored previously in Neumann et al. [6]. As we now see a more limited role of the hydrogen network and very limited electricity imports in this study, we do not think the difference between the two suggested scenarios (constrained versus unconstrained cross-border hydrogen and electricity transmission) would be substantial. Therefore, we decided not to expand the scenarios further in that regard.

RC: 9) Although the authors claim that they do not model pathways, the literature shows that imports play a significant role in the years 2030-2040 when the installation rates of renewables are limited. Could there be lock-in effects from importing a specific technology-ready commodity over another alternative?

AR: We concur that there may be certain path dependencies in choosing import vectors and volumes that we neglect. Once built, it may be more attractive to continue using the same import vector rather than changing it. The carrier-specific demands may also develop differently as the economy transforms. The larger role for imports during the transition phase towards net-zero is, for instance, shown in Kountouris et al. [53]. This result may also be influenced by different rates of technological learning and cost reductions that determine the competitiveness of domestic production versus imports.

However, we believe technology readiness, in general, to play a subordinate role for potential infrastructure lock-ins. According to a review by Sterner et al. [58] (Table 10), most import vectors we consider are already at a high technology readiness level: pipeline hydrogen (8.5), liquid hydrogen (7), liquid methane (8), ammonia (8), methanol (8, 6.5 with DAC), Fischer-Tropsch fuels (7.5, 6.5 with DAC). The TRL for hydrogen-based direct iron reduction are considered to be as high as 8 [59] and electric arc furnaces are a mature technology. While we acknowledge the lower TRL of DAC (7 according to [20]), its need could be circumvented by using biogenic CO₂ in the mid-term (e.g. <https://www.uruguayxxi.gub.uy/en/news/article/compania-chilena-concretara-millonaria-inversion-en-uruguay-para-producir-ecomcombustibles-a-partir-de-hidrogeno-v>

while experience with current DAC-based synthesis plants grows (e.g. <https://hifglobal.com/haru-oni>).

We have added the lack of pathway planning to the discussion of the limitations as running all scenarios with pathway optimisation would exceed reasonable computational resources. However, this section is now more prominently featured in the main body of the manuscript.

RC: 10) Please provide a summary mathematical description of Trace and PyPSA-EUR in SI. Since the importing topic and its modeling attempt are the main methodological novelty of the current study. Furthermore, how did you model the steel relocation (page 2, line 53)? So, is there a steel demand that can spatially change? Does the model utilize this option?

AR: The mathematical description of TRACE and PyPSA-Eur have been previously published in Hampp et al. [39] (Section S3) and Neumann et al. [6] (Section S12), respectively. We now refer to these sections, but do not think it is prudent to reprint the equations as the model structure has not changed and only some new supply technologies for the import have been added. We have expanded our description of how the relocation of the steel and ammonia industry is modelled.

In the scenario without relocation, the ammonia and primary steel production volume, and thereby the hydrogen demand, is spatially fixed based on current production volumes. In the scenario with relocation, the model can choose each region's operation mode and capacities of Haber-Bosch, DRI and EAF plants individually, subject to costs and efficiencies. The model makes use of this option abundantly, as we now show in Supplementary Fig. 6.

RC: 11) Hydrogen derivatives and steel are imported through marine infrastructure. Instead of depicting the LNG terminals that the model did not use, Figure 1 should display the location and capacity of the ports (pie chart illustrating the commodities) that will handle those shipping imports.

AR: While we agree that it would yield a nice map, this suggestion cannot be implemented as we do not optimize the import infrastructure requirements for steel products or fuels other than electricity (HVDC), hydrogen (pipelines, terminals) and methane (terminals). For the remaining carriers, we make no specification about which ports are used. We assume import capacities and transport within Europe to be unconstrained because infrastructure for handling these goods is either widely available and/or transport costs relative to their value are low.

Especially green methanol and Fischer-Tropsch fuels are similar to fossil oil which is handled at European ports today in large quantities. In 2022, *eurostat* energy balances report imports of oil and petroleum products for the EU27 of 9500 TWh (https://ec.europa.eu/eurostat/cache/infographs/energy_balances/enbal.html), which is much larger than the turnover of carbonaceous fuels we expect in our net-zero scenarios (1670 TWh). Thus, we do not expect any bottlenecks in the port infrastructure and distribution for these fuels.

Also steel is widely traded today both within the EU (93.1 Mt) and with other non-EU countries (39.2 Mt) (<https://worldsteel.org/data/world-steel-in-figures-2024/>). The Port of Antwerp alone had a throughput of more than 9 Mt of steel products in 2021 (<https://eurometal.net/>

port-of-antwerp-expects-strong-steel-traffic/). As we assume the share of secondary steel production to increase from 40% to 70% through increasing the amount and quality of scrap metal collected [60], we expect that existing port infrastructure and distribution logistics in Europe could handle the remaining volume of 52 Mt of primary steel, whether it originates from European ports or ports abroad.

For ammonia, we were unable to find reliable estimates for terminal costs but recent developments in Europe also suggest no shortage in ammonia terminal capacity to supply the 85 TWh/a demand we assume for our net-zero scenario. The two German ammonia terminals in Rostock and the one in Brunsbüttel – that opened in October 2024 – together have a capacity of 3.6 Mt/a (18.7 TWh/a) (<https://www.chemanager-online.com/en/news/yara-modifies-german-ammonia-terminals>, <https://www.yara.com/corporate-releases/yara-drives-hydrogen-economy-with-new-ammonia-import-terminal/>). Another example is the ammonia terminal in Rotterdam, which is planned to be expanded to a capacity of 3 Mt/a (15.6 TWh/a) (<https://www.portofrotterdam.com/en/news-and-press-releases/oci-expands-import-terminal-for-green-ammonia>). Further smaller ammonia terminals are distributed across Europe (<https://futurefuels.imo.org/wp-content/uploads/2024/03/WorldAmmoniaMap2024.pdf>) that handle the European share of today's 20.6 Mt/a (107.2 TWh/a) global ammonia trade [61].

RC: Specific comments

Abstract: Please indicate that the current study incorporates 2050 insights.

AR: We now mention that we look at “net-zero emission scenarios”. We think this is more accurate than referring to a particular year.

RC: Line 1: Please offer a reference for why “importing renewable energy to Europe promises several advantages for achieving a swift energy transition”.

AR: This is a fairly generic opening statement. As it does not make a factual claim, we do not think it requires a scientific citation. It rather reflects the ongoing political debate, which we highlight in more detail in the fourth paragraph. We have slightly rephrased the sentence using “may/might offer” instead of “promises” to clarify that this is a hypothesis.

RC: In line 7, avoid lump sum references (1-8) and instead appropriately cite the research or exclude unnecessary ones.

AR: We concur that bulk references should generally be avoided. However, in this case, we would argue that it is appropriate. First, the references are all further categorized in the subsequent paragraph starting “While many previous academic studies...”. Second, all references are equally relevant to support the statement of the sentence.

RC: Nature Communications does not permit footnotes—first page, right bottom.

AR: We have removed all footnotes.

RC: Page 2, line 20. Are you referring to Supplementary Figures 6 and 7? Those figures do not

demonstrate the unique characteristics of the imported vectors.

AR: We originally referenced the energy balances in the SI because they show the consumption volumes and production routes of each carrier. Since the energy balances pre-empt the results, we have now removed these references.

RC: On page 2, line 50, many details are also mentioned in the method sections. You can choose to avoid some options, such as 110 areas or 4-hour resolutions.

AR: We acknowledge that there is some duplication between the mentioned paragraph and the methods section. Nevertheless, we believe that the paragraph is necessary to provide a high-level overview of the model and the scenario due to the ordering of the results and method sections. In response to your comment, we have shortened the paragraph a bit and expanded detail in the methods section.

RC: page 2, line 65, please include some figures or tables showing the spatial distribution of the demands (hydrogen, methanol, ammonia still, etc.). Could be at the SI. Your demand allocation drives the network development.

AR: We agree that network infrastructure can be driven by exogenous demand allocation. As outlined in our response to comment 5), we have now added Supplementary Fig. 4 showing the spatial distribution of spatially-fixed exogenous demands, as well as Supplementary Fig. 6 showing the relocation patterns of steel and ammonia production observed in the model when allowed.

In fact, only in scenarios without relocation do we have substantial amounts of spatially-fixed hydrogen consumption for steel and ammonia production. Apart from these industries, there is just a 5 TWh/a demand for hydrogen for high-value chemicals (other than the liquid feedstocks), which is counterbalanced by spatially-fixed hydrogen *production* of around 10 TWh/a from chlor-alkali electrolysis in the chlorine production. High-temperature industry heat is assumed to be supplied by methane (fossil, synthetic, or biogenic). In the transport sector, there is no direct hydrogen consumption in light-duty or heavy-duty road transport since it is all electrified. In district heating and the power sector, backup hydrogen capacities (CHP or turbines) are endogenously sited just as as the production capacities of other hydrogen derivatives (Fischer-Tropsch, methane, methanol).

As there is little to no spatially-fixed hydrogen demand assumed when the steel and ammonia production can relocate, there is little impetus for transporting hydrogen directly, apart from transporting hydrogen to e-fuel production sites in regions where there is high value for its excess heat. The result is a network that is much smaller than, for instance, envisioned in the European Hydrogen Backbone [50].

RC: page 2, even after finishing page 2, there is no mention that most of the assumptions (except technology cost) discussed and importing options are based on 2050 estimates. Kindly include some information for the time horizon of the study.

AR: We mention – before the end of page 2 – that we use techno-economic assumptions for 2040

(previously 2030), which besides costs includes assumptions about efficiencies and other technical parameters:

We utilize techno-economic assumptions for 2040, reflecting that infrastructure required for achieving carbon neutrality must be built well in advance of reaching this goal.

We also mention that we look at “net-zero emission scenarios”:

In this study, we explore the full range between the two poles of complete self-sufficiency and wide-ranging renewable energy imports into Europe in scenarios with high shares of wind and solar electricity and net-zero carbon emissions.

We think this is more accurate than referring to a particular year.

RC: Page 2, line 80. The title does not accurately reflect the content. Here, you reflect on some background information. The authors do not perform a qualitative or quantitative assessment of importing choices.

AR: We chose the title as Figure 1, which is mainly referenced in this section, shows our cost assessment of energy and material import vectors to Europe. We wish to keep the title, but now discuss more results in this section.

RC: page 2, line 85. "Our selection of exporting countries". Should it be importing countries?

AR: No, “exporting countries” is correct for referring to countries that are considered as candidates for exporting energy to Europe.

RC: What is the domestic average cost of those energy vectors in Europe so as to compare to Figure 1b (violin plot)? Perhaps direct the reader to the figures in Appendix Sup Fig 5 where you show the supply curves. Or comment on it.

AR: The domestic average production costs in comparison to the import costs can be seen in Figure 4. In the caption of Figure 4, we now refer to the supply curves in Supplementary Fig. 21. We opted not to include this information in Figure 1, as the domestic production costs is a scenario-dependent result and not an input to PyPSA-Eur. For instance, the domestic production cost of methanol is lower if it can use lower-cost hydrogen from abroad and also depends on how the model chooses to operate the methanolisation units.

RC: In Figure 4, the bubble in Greece appears to have shifted down from the polygon center.

AR: Right, we have now aligned the centre-point position with other graphics. The shift was due to using an alternative method for calculating the representative points of the model regions.

RC: Page 6, line 31. How is it feasible to integrate all the waste to heat? In some nations, district heating and other choices are unavailable. Even if you assume that waste to heat has a sector-coupled

value, is that enough to influence the importing results? Did you attempt a sensitivity that assumes a smaller heat integration?

AR: Based on your comment and other feedback we received when presenting this work, we have now reduced the utilizable share of waste heat from power-to-X processes in district heating to 25% in the main scenarios. This estimate accounts for the fact that within the 115 larger model regions only a fraction of fuel synthesis plants might be connected to district heating systems. Additionally, the waste heat can only be used where the population density allows for district heating and there is heat demand, otherwise the waste heat is vented. The value of the waste heat is endogenously determined by the model in each region based on heat demand and supply options, corresponding to the shadow price of the heat balance constraints.

We conducted additional sensitivity analyses with 0% waste heat availability and a case where all waste heat is useable for district heat. In the scenario without imports, the total system cost difference between no and full waste heat utilization is 20 bn€/a (2.4%). The first row of Supplementary Fig. 25 also demonstrates that the utilization potential of waste heat can have an influence on the siting of power-to-X plants, as they locate closer to regions with district heating and high value for additional heat supply. Since the margin of import costs versus domestic production can be quite small (Figure 4), we conclude that a revenue from heat integration can also shift the balance towards more domestic production. The cost benefit of imports reduced from 4.4% to 3.3% when waste heat is was fully available. This aspect is now discussed in more detail in the sectino “Interactions of import strategy & domestic infrastructure” in the revised manuscript.

RC: Page 6, line 56. The 20% sensitivity to the default import cost assumes that the estimated initial import costs have the same level of uncertainty. However, this is not the case. For example, electricity imports, hydrogen imports, and steel exhibit different uncertainty costs. Could you kindly discuss the current estimates and their future evolution, as well as your decision to choose the 20% to perform the current sensitivity analysis?

AR: We agree that uncertainty in different cost components affects import vectors to varying extents. For instance, higher costs of electrolysers will affect hydrogen and all its derivatives, whereas higher costs for direct air capture will only affect carbonaceous fuels. We therefore now perform three (previously two) categories of import cost variations: one where cost variation is applied to all import vectors (Figure 5a), one where it is applied to all import vectors but electricity, reflecting cost uncertainty in hydrogen conversion (Figure 5b), and one where is applied only to carbonaceous fuel import vectors, reflecting cost uncertainty in carbon provision (Figure 5c). For computational reasons, we needed to group the technology uncertainties in this way.

We have also expanded the range of uncertainty considered to $\pm 50\%$. We believe that this range is sufficient, considering that beyond 30% higher import costs the model no longer imports energy and below 50% lower import costs, the costs for the domestic energy system are only marginally reduced (Figure 6). In Table 1, which was moved to the main body, we also showcase what can cause such cost variations.

- RC:** Figure 6: it is difficult to differentiate colors in the legend; try aggregating some possibilities. For example, fossil fuels is one category, and domestic renewable energy production is another, so as to demonstrate the impact of imports, which is the primary focus of the work.
- AR:** In the new version of the manuscript, we have revised the design of Figure 4. We have reduced the number of technology categories, aggregating some cost components further (e.g. ‘fossil’, ‘gas-to-power/heat’, and ‘other’ for smaller components). The order of colors in the plot now also matches the order in the single-column legend, and we split the figure into two panels.
- RC:** On page 8, line 64, please put your estimated values in perspective. What are the current annual expenditures of importing LNG compared to the future system savings from importing an alternative energy vector?
- AR:** According to *eurostat*, total EU LNG imports in 2023 are reported to have valued at around 65 bn€ for 1,390 TWh (≈ 46.7 €/MWh) (<https://ec.europa.eu/eurostat/web/products-eurostat-news/w/ddn-20240923-1>). Additionally, the EU LNG terminal service fares range between 0.5-2.90 €/MWh, adding up to 4 bn€ [62]. The benefit of 37.1 bn€/a from green energy imports we see in our central scenario, is around 54% of the total LNG import costs.
- RC:** Discussion and conclusion. It reads well but might benefit from additional information, such as key values and comparisons to other studies.
- AR:** We reiterate a few key values in the discussion/conclusion and have amended the section with a paragraph comparing our results to a selection of other studies looking at the relation between energy imports and the European energy system: Kountouris et al. [53], Wetzel et al. [54], Genge et al. [57], Frischmuth et al. [55], Schmitz et al. [56].
- RC:** Figure 7. Again, it’s challenging to comprehend the graphs because so many variables have similar colours. Also, consider showing fewer plots in the same picture; someone must zoom in (at least 300%) to see where the production and consumption occur. In the fourth plot there is half-circled above Portugal and Spain. What does it represent? Why are two arrowheads connecting Romania, Hungary, and Austria in the same line?
- AR:** We acknowledge that it takes some time to find all details included in Figure 7. Nevertheless, we also think it is useful to have a comparison of the electricity and hydrogen system of different scenarios on a single page. We have made the following changes to improve readability:
- We have zoomed in on the map.
 - We have adjusted the size of the pie charts, arrows and pipeline widths.
 - We have tweaked some of the most similar colors to be more distinct.
 - We have improved the labelling of the half-circle in the Gulf of Biscay, which represents the imports of hydrogen derivatives that are not spatially resolved in the model.
 - For the arrows, previously bidirectional flows were not netted out, which we have now corrected.

RC: The link does include a readme file that contains sufficient information on how to install and use the model. I did not investigate whether I could install and run the model in this review stage.

AR: Thank you. No further response required.

Reviewer #5

RC: Key results

The study gives a very good insight into the effects of possible future energy imports in a carbon-neutral Europe on the resulting system costs and the needed infrastructure in Europe. The authors find relevant cost reduction potentials of energy imports (including steel) compared to a system with fully domestic production.

AR: Thanks. No further response required.

RC: Validity

The authors are using a rather detailed representation of the energy system (with some necessary simplifications that do not harm the validity of the study). The resulting limited import shares in the main results lie in orders of magnitude of realistic import shares and quantities.

AR: Thanks. No further response required.

RC: Significance

The study is highly significant, especially behind the backdrop of current policy discussions at the European level about future clean/green fuel imports and (planned) subsidy schemes for imports like the European Hydrogen Bank.

AR: Thanks. We have picked up the point about the European Hydrogen Bank in the introduction as another example for interest in green fuel imports in Europe.

RC: Data and methodology

The used data and model have been fully provided by the authors. The authors are using an established multi-energy system model that has been used in many prior publications. The model setup and the import extensions seem fully reasonable and the approach is thereby completely valid. Some assumptions are made for complexity reduction but they are discussed in the supplementary material. I can see a few possible improvements further discussed below. The input data is based on established sources and therefore also valid.

AR: Thanks. We have moved the discussion of limitations to the main body of the revised manuscript.

RC: Analytical approach

The main results are presented in a very convincing manner. Scenarios and sensitivity tests are run to further examine the results. The chosen scenarios and sensitivities are sufficient for a robust derivation of conclusions. Some possible improvements are elaborated below.

AR: Thanks. No further response required.

RC: Suggested improvements

The manuscript already adheres to a very high standard. Nevertheless, I would like to suggest thinking about the following improvements:

Import dependencies are mentioned in the text as a risk factor. In the results and discussion, this is not really taken up again. I could not find any detailed information on the quantities imported by country. It is very likely that the imports will only come from the lowest-cost region since supply seems to be abundant but maybe this is not the case and we could see an import mix. The supplementary material does discuss cost increases or decreases for an export location and also tests sensitivities but as far as I understood only as a uniform increase or decrease for all exporting regions. A more nuanced discussion of this aspect could increase the policy relevance of the paper. It could be interesting to remove certain (lowest-cost) countries to test the effects.

AR: Thank you for the suggestion. We now discuss a new Sankey chart, which was added to Figure 3, showing origins and destinations of different fuels. The Sankey charts of further scenarios are presented in Supplementary Figs. 19 and 20.

Compared to the initial submission, we have now also imposed a limit of 500 TWh per exporting country – for the sum of all carriers – to both prevent a single country from dominating the import mix and match the export volume assumed for the levelised cost calculation in the TRACE model for consistency. Thereby, we can now see an import mix of carbonaceous fuels from Argentina and Chile, as well as steel, hydrogen, ammonia and some electricity from the MENA region. However, we should caution against overinterpreting these country-specific results, as the import cost supply curves, which we now present for different vectors in Supplementary Figs. 7 to 10, are relatively flat. Other countries could often step in at marginally higher costs.

While not explicitly modelling the scenario where the lowest-cost country drops out of the market, the resulting shift of the import cost curve to the left is – with some limitations – similar to moving the cost curve up by a certain percentage. Thus, the impact of some of the low-cost regions dropping out could be approximately inferred by the import cost variations we already study (e.g. +10% for all (carbonaceous) carriers).

RC: Domestic supply is assumed to have a non-flat cost-duration curve, while imports are considered at a fixed/average cost as far as I understood correctly. The supplementary material discusses the influence of running the export systems as islanded systems and the need to overplant or usage

of storage. I am not fully convinced that the exporters would offer the energy vectors at a flat cost curve and think we would rather see seasonal price variations (due to the transport by ship or pipeline not hourly/daily but still). Could you please elaborate a bit on why you made these assumptions and - if possible - test the influence of that on the results? It does seem to favor imports significantly over domestic production but maybe the effect is negligible.

AR: Our assumption that exporters offer their commodities at a time-constant cost reflects the existence of short- or long-term off-take agreements between exporters and importers, ensuring cost recovery on an annual basis.

The alternative scenario you mention, involving seasonally or time-varying commodity costs, would resemble a market-driven structure where exporters offer products based on current production costs to recover their costs in the short term. Modelling such a market would require including all potential exporters and as well as offtakers and their seasonal demand – even those outside the European system we are focusing on. In this scenario, market prices for import vectors would have to be estimated and would generally be higher than those from the lowest-cost exporter, as market pricing would ensure that even higher-cost exporters can recover their expenses.

We anticipate a future scenario where both direct export/import agreements and market-based trades coexist, similar to today's LNG market. Export/import agreements would cover most of the expected demand (as modeled here), while the market would address unplanned or excess demand. For a recent analysis of how international hydrogen trade could develop, see Antweiler and Schlund [63].

Therefore, we assume that import vectors traded on the market would be at a cost disadvantage (i.e., higher cost) compared to what we have modeled. Seasonal prices would either favour more domestic production or encourage additional import and storage infrastructure to allow fuel purchases when prices are low as they are largely easy and cheap to store.

RC: You are using a greenfield approach with cost data for 2030 to reflect the need to build infrastructure rather sooner than later. While I think this is a reasonable assumption, I am wondering about the effects of existing generation capacity in Europe that will most likely be in the market for a very long time to come. I am thinking of nuclear capacity here in countries like France, the UK, Sweden, and Finland but also in Eastern Europe with lifetime extensions of existing plants well into the 2030s but also new-build reactors that just entered the system or are about to enter it or will "soon" be there based on the current political discussion and situation in Europe. Together with the large capacities of fluctuating renewables and the rather inflexible nature of NPPs as well as the European regulation on clean/green hydrogen from electricity systems with very low emission rates, this could influence the competitiveness of domestic supply due to the inflexibility of generation - even though it is very expensive to build since NPPs are comparably cheap to operate and the construction costs are sunk anyways. I see that this is out of scope for your current greenfield model setup but I think it should still be touched upon in the discussion and/or limitations of the study. Already existing NPPs or those under construction or with an FID taken might not tip the scale due to the limited capacity. With the current push for many more new

projects and new countries entering the nuclear market like Poland, the picture might change, though. Still, we can be very skeptical about the realization of such projects. As far as I see, the model setup currently does not consider any possible investments into new nuclear capacity (and it is unlikely to materialize at realistic cost assumptions). Maybe you don't want to open Pandora's box here and this is for another paper - I would just like to at least see it discussed.

AR: In response to your comment and similar discussions with others when presenting the work we have made the following changes:

First, because our assumption of 2030 costs for net-zero scenarios repeatedly caused confusion, we now use cost projections for 2040 as a compromise.

Second, we now include 52 GW of nuclear power plants which are either operational today and were phased-in after 1990 (20 GW) or are currently in (pre-)construction (32 GW), according to Global Energy Monitor's Nuclear Power Plant Tracker (<https://globalenergymonitor.org/projects/global-nuclear-power-tracker/>). Thereof, 18 GW are in France, 13 GW are in the UK and below 4 GW each in Poland, Romania, Netherlands, Slovakia, Bulgaria, Hungary and Finland. This is around half of the currently operating nuclear capacity of 106 GW, of which 85 GW are built before 1990, of which 50 GW are in France. By taking 1990 as a cut-off date, we assume a maximum lifetime of 60 years for existing nuclear plants, which is for instance assumed in NREL's Annual Technology Baseline database (<https://atb.nrel.gov/electricity/2024/nuclear>).

Because modelling proper unit-commitment constraints is not computationally viable in this setting, we assume that the existing nuclear fleet is operated at full capacity, applying country-specific availability factors of 2021-2023 period as a time-constant factor (e.g. 61.6% for France, <https://pris.iaea.org/PRIS/WorldStatistics/ThreeYrsEnergyAvailabilityFactor.aspx>). At a cost of 8594 €/kW (Lazard's levelized cost of energy analysis - version 16.0 (2023): <https://www.lazard.com/media/typdgxmm/lazards-lcoeplus-april-2023.pdf>), the nuclear capacity can also be freely expanded in regions that currently have, once had or have announced nuclear capacity according to Global Energy Monitor's Nuclear Power Plant Tracker. However, as you say, these investment costs are not low enough to make this option economically attractive.

Since we do not think including nuclear plants in operation and under construction significantly impacted our results, we explain our approach in the methods section but do not discuss nuclear power in more detail elsewhere to not distract from the key insights of the paper.

RC: As I understood also the potential hydrogen pipeline infrastructure is based on a greenfield approach (while powerlines already exist). This is not in line with the current plans of repurposing large parts of the existing fossil gas infrastructure. Propagated industry plans but also other works of yours are assuming a rate of 60% repurposed pipelines in the future hydrogen system. While the real cost rates for repurposing are not yet known and potentially underestimated, this might still change parts of your outcomes. As far as I understood you did not include this aspect due to computational burdens. Could this not be depicted in a slim manner by assuming lower cost factors for existing relations? The aspect is only mentioned in the "Methods" section so I did not

see any discussion on the implications of this assumption on the results.

AR: Correct, in a previous study, we found a retrofitting rate of 69% of all hydrogen pipelines built [6]. In response to your comment, we have now reactivated this option, so that existing gas pipelines can be retrofitted to hydrogen pipelines at a cost that is approximately half that of a new hydrogen pipeline. Given that in the study referenced above, the system benefit of the hydrogen network far outweighed its cost, we believe the impact of pipeline costs on the results would be limited.

Generally, we also see a smaller role for the hydrogen network in the results as we allow the relocation of hydrogen demands for steel and ammonia industries, limit the use of waste heat of hydrogen conversion units to derivative fuels, and assume that the CO₂ can be transported across Europe at low cost. Thus, we do not touch on the ratio of new and repurposed pipelines in detail.

RC: Graphs: The choropleth layers in the map graphs are using color scales reaching from darker colors for lower specific costs to lighter colors for higher specific costs. While this can be intuitive in a way (more imports from lower-cost regions) it is a bit confusing to use an inverse scale here, especially since countries or regions not considered or not containing entry points are light gray and hard(er) to distinguish from high-cost areas. Maybe the scale could go from green/blue for low cost to red/orange for high costs to better distinguish this (or any other color suitable for colorblindness).

AR: We have changed the colormap to 'viridis', which is now consistently used for costs on maps. We have also tried to indicate more clearly which potential exporting regions we consider.

RC: Clarity and context

The manuscript is drafted in a very clear and accessible language and follows a read thread. The results are presented in complex but informative graphs and the supplementary material covers a number of sensitivities and takes up the limitations of the model and setup at hand. The visual quality of the graphs stands out and the use of vector formats contributes to a very high legibility (in contrast to many other studies, unfortunately). (The manuscript contains a few typos.)

AR: Thank you. We have further simplified some figures and added additional sensitivity analyses. We have also corrected any typos we could find.

RC: References

The study deals with the existing literature in detail and clearly distinguishes the study from already existing similar approaches and reasons the novelty at a sufficient level.

AR: Thank you. We have also amended our literature review with some references that were released or found after the initial submission.

RC: I have checked the repository visually: the full code and data are available and well-documented

for the experiment to be reproduced. I did not execute the code due to the complexity of the setup.

AR: Thank you.

References

- [1] Gielen, D., Saygin, D., Taibi, E. & Birat, J.-P. Renewables-based decarbonization and relocation of iron and steel making: A case study. *Journal of Industrial Ecology* **24**, 1113–1125 (2020).
- [2] Samadi, S., Fischer, A. & Lechtenböhmer, S. The renewables pull effect: How regional differences in renewable energy costs could influence where industrial production is located in the future. *Energy Research & Social Science* **104**, 103257 (2023).
- [3] Verpoort, P. C., Gast, L., Hofmann, A. & Ueckerdt, F. Impact of global heterogeneity of renewable energy supply on heavy industrial production and green value chains. *Nature Energy* **9**, 491–503 (2024).
- [4] Egerer, J., Farhang-Damghani, N., Grimm, V. & Runge, P. The industry transformation from fossil fuels to hydrogen will reorganize value chains: Big picture and case studies for Germany. *Applied Energy* **358**, 122485 (2024).
- [5] Turkovska, O. *et al.* Land-use Requirements of Solar and Wind Power. <https://eartharxiv.org/repository/view/6319/> (2023).
- [6] Neumann, F., Zeyen, E., Victoria, M. & Brown, T. The potential role of a hydrogen network in Europe. *Joule* **7**, 1793–1817 (2023).
- [7] Shokri Gazafroudi, A., Neumann, F. & Brown, T. Topology-based approximations for N - 1 contingency constraints in power transmission networks. *International Journal of Electrical Power & Energy Systems* **137**, 107702 (2022).
- [8] Mattsson, N., Verendel, V., Hedenus, F. & Reichenberg, L. An autopilot for energy models – Automatic generation of renewable supply curves, hourly capacity factors and hourly synthetic electricity demand for arbitrary world regions. *Energy Strategy Reviews* **33**, 100606 (2021).
- [9] Riahi, K. *et al.* The Shared Socioeconomic Pathways and their energy, land use, and greenhouse gas emissions implications: An overview. *Global Environmental Change* **42**, 153–168 (2017).
- [10] Hévin, G. Underground storage of hydrogen in salt caverns. <https://energnet.eu/wp-content/uploads/2021/02/3-Hevin-Underground-Storage-H2-in-Salt.pdf> (2019).
- [11] Hydrogen TCP-Task 42. Underground Hydrogen Storage: Technology Monitor Report. <https://www.ieahydrogen.org/task/task-42-underground-hydrogen-storage/> (2023).
- [12] Talukdar, M., Blum, P., Heinemann, N. & Miocic, J. Techno-economic analysis of underground hydrogen storage in Europe. *iScience* **27**, 108771 (2024).
- [13] Caglayan, D. G. *et al.* Technical potential of salt caverns for hydrogen storage in Europe. *International Journal of Hydrogen Energy* **45**, 6793–6805 (2020).
- [14] Guha Roy, D., Goyal, J. & Talukdar, M. Capacity assessment and economic analysis of geologic storage of hydrogen in hydrocarbon basins: A South Asian perspective. *International Journal of Hydrogen Energy* **92**, 917–929 (2024).

- [15] Amirthan, T. & Perera, M. Underground hydrogen storage in Australia: A review on the feasibility of geological sites. *International Journal of Hydrogen Energy* **48**, 4300–4328 (2023).
- [16] Zhu, S. *et al.* Site selection evaluation for salt cavern hydrogen storage in China. *Renewable Energy* **224**, 120143 (2024).
- [17] Lackey, G. *et al.* Characterizing Hydrogen Storage Potential in U.S. Underground Gas Storage Facilities. *Geophysical Research Letters* **50**, e2022GL101420 (2023).
- [18] Gøtske, E. K., Andresen, G. B., Neumann, F. & Victoria, M. Designing a sector-coupled European energy system robust to 60 years of historical weather data. <https://arxiv.org/abs/2404.12178> (2024).
- [19] Hofmann, F., Tries, C., Neumann, F., Zeyen, E. & Brown, T. H₂ and CO₂ Network Strategies for the European Energy System. <https://arxiv.org/abs/2402.19042> (2024).
- [20] IEA. ETP Clean Energy Technology Guide. <https://www.iea.org/data-and-statistics/data-tools/etp-clean-energy-technology-guide> (2024).
- [21] IRENA. A pathway to decarbonise the shipping sector by 2050. https://www.irena.org/-/media/Files/IRENA/Agency/Publication/2021/Oct/IRENA_Decarbonising_Shipping_2021.pdf (2021).
- [22] Global Maritime Forum. Annual Progress Report on Green Shipping Corridors. <https://globalmaritimeforum.org/report/2023-annual-progress-report-on-green-shipping-corridors/> (2023).
- [23] Maersk. Maersk to deploy first large methanol-enabled vessel on Asia - Europe trade lane. <https://www.maersk.com/news/articles/2023/12/07/maersk-to-deploy-first-large-methanol-enabled-vessel-on-asia-europe-trade-lane> (2023).
- [24] Korberg, A., Brynolf, S., Grahn, M. & Skov, I. Techno-economic assessment of advanced fuels and propulsion systems in future fossil-free ships. *Renewable and Sustainable Energy Reviews* **142**, 110861 (2021).
- [25] Link, S., Stephan, A., Speth, D. & Plötz, P. Rapidly declining costs of truck batteries and fuel cells enable large-scale road freight electrification. *Nature Energy* **9**, 1032–1039 (2024).
- [26] Nykvist, B. & Olsson, O. The feasibility of heavy battery electric trucks. *Joule* **5**, 901–913 (2021).
- [27] Phadke, A., Khandekar, A., Abhyankar, N., Wooley, D. & Rajagopal, D. Why Regional and Long-Haul Trucks are Primed for Electrification Now. <https://www.osti.gov/servlets/purl/1834571/> (2021).
- [28] Brown, T. & Hampp, J. Ultra-long-duration energy storage anywhere: Methanol with carbon cycling. *Joule* **7**, 2414–2420 (2023).
- [29] Mbatha, S. *et al.* Power-to-methanol process: A review of electrolysis, methanol catalysts, kinetics, reactor designs and modelling, process integration, optimisation, and techno-economics. *Sustainable Energy & Fuels* **5**, 3490–3569 (2021).
- [30] Dieterich, V., Buttler, A., Hanel, A., Spliethoff, H. & Fendt, S. Power-to-liquid via synthesis of methanol, DME or Fischer-Tropsch-fuels: A review. *Energy & Environmental Science* **13**, 3207–3252 (2020).
- [31] Wentrup, J., Pesch, G. R. & Thöming, J. Dynamic operation of Fischer-Tropsch reactors for power-to-liquid concepts: A review. *Renewable and Sustainable Energy Reviews* **162**, 112454 (2022).
- [32] Mucci, S., Mitsos, A. & Bongartz, D. Power-to-X processes based on PEM water electrolyzers: A review of process integration and flexible operation. *Computers & Chemical Engineering* **175**, 108260 (2023).

- [33] Svitnič, T. & Sundmacher, K. Renewable methanol production: Optimization-based design, scheduling and waste-heat utilization with the FluxMax approach. *Applied Energy* **326**, 120017 (2022).
- [34] Mission Possible Partnership. Making Net-Zero Steel Possible. <https://3stepsolutions.s3-accelerate.amazonaws.com/assets/custom/010856/downloads/Making-Net-Zero-Steel-possible.pdf> (2022).
- [35] Fasihi, M., Efimova, O. & Breyer, C. Techno-economic assessment of CO₂ direct air capture plants. *Journal of Cleaner Production* **224**, 957–980 (2019).
- [36] Ozkan, M., Nayak, S. P., Ruiz, A. D. & Jiang, W. Current status and pillars of direct air capture technologies. *iScience* **25**, 103990 (2022).
- [37] Hersbach, H. *et al.* The ERA5 global reanalysis. *Quarterly Journal of the Royal Meteorological Society* **146**, 1999–2049 (2020).
- [38] Pfeifroth, U. *et al.* Surface Radiation Data Set - Heliosat (SARAH) - Edition 3. https://wui.cmsaf.eu/safira/action/viewDoiDetails?acronym=SARAH_V003 (2023).
- [39] Hampp, J., Düren, M. & Brown, T. Import options for chemical energy carriers from renewable sources to Germany. *PLOS ONE* **18**, e0262340 (2023).
- [40] Ruiz, P. *et al.* ENSPRESO - an open, EU-28 wide, transparent and coherent database of wind, solar and biomass energy potentials. *Energy Strategy Reviews* **26**, 100379 (2019).
- [41] Millinger, M. *et al.* Diversity of biomass usage pathways to achieve emissions targets in the European energy system. <https://www.researchsquare.com/article/rs-3097648/v1> (2023).
- [42] Global Energy Monitor. Europe Gas Tracker. <https://globalenergymonitor.org/projects/europe-gas-tracker/> (2024).
- [43] Hurskainen, M. Liquid organic hydrogen carriers (LOHC). <https://cris.vtt.fi/en/publications/liquid-organic-hydrogen-carriers-lohc-concept-evaluation-and-tech> (2019).
- [44] Runge, P. *et al.* Economic comparison of different electric fuels for energy scenarios in 2035. *Applied Energy* **233–234**, 1078–1093 (2019).
- [45] Calcaterra, M. *et al.* Reducing the cost of capital to finance the energy transition in developing countries. *Nature Energy* **9**, pages 1241–1251 (2024).
- [46] Egli, F., Steffen, B. & Schmidt, T. S. Bias in energy system models with uniform cost of capital assumption. *Nature Communications* **10**, 4588 (2019).
- [47] Bogdanov, D., Child, M. & Breyer, C. Reply to ‘Bias in energy system models with uniform cost of capital assumption’. *Nature Communications* **10**, 4587 (2019).
- [48] Lonergan, K. E. *et al.* Improving the representation of cost of capital in energy system models. *Joule* **7**, 469–483 (2023).
- [49] Global Energy Monitor. Global Steel Plant Tracker. <https://globalenergymonitor.org/projects/global-steel-plant-tracker/> (2024).
- [50] Gas for Climate. European Hydrogen Backbone - Analysing future demand, supply, and transport of hydrogen. https://gasforclimate2050.eu/wp-content/uploads/2021/06/EHB_Analysing-the-future-demand-supply-and-transport-of-hydrogen_June-2021.pdf (2021).
- [51] Miocic, J. *et al.* Underground hydrogen storage: A review. *Geological Society, London, Special Publications* **528**, 73–86 (2023).

- [52] Ozarslan, A. Large-scale hydrogen energy storage in salt caverns. *International Journal of Hydrogen Energy* **37**, 14265–14277 (2012).
- [53] Kountouris, I. *et al.* A unified European hydrogen infrastructure planning to support the rapid scale-up of hydrogen production. *Nature Communications* **15**, 5517 (2024).
- [54] Wetzel, M., Gils, H. C. & Bertsch, V. Green energy carriers and energy sovereignty in a climate neutral European energy system. *Renewable Energy* **210**, 591–603 (2023).
- [55] Frischmuth, F. & Härtel, P. Hydrogen sourcing strategies and cross-sectoral flexibility trade-offs in net-neutral energy scenarios for Europe. *Energy* **238**, 121598 (2022).
- [56] Schmitz, R. *et al.* Implications of hydrogen import prices for the German energy system in a model-comparison experiment. *International Journal of Hydrogen Energy* **63**, 566–579 (2024).
- [57] Genge, L., Scheller, F. & Müsgens, F. Supply costs of green chemical energy carriers at the European border: A meta-analysis. *International Journal of Hydrogen Energy* **48**, 38766–38781 (2023).
- [58] Sterner, M. *et al.* 19 Import options for green hydrogen and derivatives - An overview of efficiencies and technology readiness levels. *International Journal of Hydrogen Energy* **90**, 1112–1127 (2024).
- [59] Shahabuddin, M., Brooks, G. & Rhamdhani, M. A. Decarbonisation and hydrogen integration of steel industries: Recent development, challenges and technoeconomic analysis. *Journal of Cleaner Production* **395**, 136391 (2023).
- [60] Material Economics. The circular economy – a powerful force for climate mitigation. <https://www.sitra.fi/app/uploads/2018/06/the-circular-economy-a-powerful-force-for-climate-mitigation.pdf> (2018).
- [61] Egerer, J., Grimm, V., Niazmand, K. & Runge, P. The economics of global green ammonia trade – “Shipping Australian wind and sunshine to Germany”. *Applied Energy* **334**, 120662 (2023).
- [62] ACER. Analysis of the European LNG market developments - 2024 Market Monitoring Report. https://www.acer.europa.eu/sites/default/files/documents/Publications/ACER_2024_MMR_European_LNG_market_developments.pdf (2024).
- [63] Antweiler, W. & Schlund, D. The emerging international trade in hydrogen: Environmental policies, innovation, and trade dynamics. *Journal of Environmental Economics and Management* **127**, 103035 (2024).

Author Response to Reviews of

Energy Imports and Infrastructure in a Carbon-Neutral European Energy System

Fabian Neumann, Johannes Hampp, Tom Brown

Revision #2: February 27, 2025

RC-2.n: Reviewer Comment, AR-2.n: Author Response, ■ *Manuscript text*

We thank the reviewers for their constructive comments and acknowledge the time and effort they have spent in assessing our work. We have revised the paper based on the feedback from the reviewers and hope that we could adequately address their remaining concerns.

In response to the reviewers' comments, we (1.) performed 99 additional sensitivity runs without steel or ammonia relocation and with techno-economic assumptions projected for 2030 and 2050, (2.) expanded the supplementary material with a scenario overview table and a mathematical description of the model, (3.) improved the instructions for reproducing our results, and (4.) provided more context and warnings about interpreting the scenarios with domestic relocation of steel and ammonia production.

To follow the revisions made, we have highlighted differences compared with the previous submitted version of the paper in blue and red text in an attached file. Figure numbers in the responses refer to the numbering in the revised manuscript.

Reviewer #1

RC-2.1: Overview

The new version of the article is significantly improved in terms of clarity and uncertainty assessment. In particular, I am pleased to see more nuance in the discussion of the trade-offs between European energy security and cost reductions, the improved clarity in the figures and the expanded explanation of the two models utilised.

However, I believe that some changes are still needed. I will go over these in a case-by-case basis.

AR-2.1: We are glad that you enjoyed the improvements of the revised version. To address your remaining concerns, we have run 99 additional scenarios without relocation of steel and ammonia production

(1.) and with alternative techno-economic assumptions (2.), which we integrated into the main figures and text. We also provide a more detailed overview of the scenarios in the supplementary material in tabular format (3.).

RC-2.2: Major comments

1. On whether or not allowing for relocation of energy intensive industries is a sound default case.

I am glad to see an exploration of not allowing this relocation in the supplementary material. The authors correctly identify the lack of data on this dimension in their response, meaning that comparing the relocation / no-relocation scenarios in terms of costs is difficult.

However, I the article still needs more transparency on the degree of uncertainty that this assumption implies. Other studies not accounting for this implies a gap in the field, not a general truth. It should not be casually dismissed.

I have two suggestions:

- Directly stating this assumption in the abstract of the article. Without it, the 1-10% cost reduction mentioned in it would be misleading.
- Figures S6 and S25 are useful, but not enough to put other results into context. Even if comparisons to your default case are difficult, consider generating something similar to Figure 2 for this case. I am sure other researchers and policy-makers would appreciate a nuanced discussion.

AR-2.2: We concur that the influence of assuming intra-European steel and ammonia industry relocation on our results should be evaluated more thoroughly. Therefore, we have repeated the scenarios of Figure 2 without relocation and amended Figure 2 with these results. An additional element in the upper panel is now comparing the relative cost reductions for equivalent import vectors allowed but no relocation. Note that we compare relative rather than absolute cost savings for these additional runs because the base cost level (no-imports case) is different. The added results demonstrate that assumptions about domestic steel/ammonia industry relocation only marginally affect the cost reduction potentials from imports, strengthening the previous findings. Thank you for suggesting it.

We considered adjusting the abstract according to your suggestion. However, since domestic industry relocation proved peripheral to import-based cost reductions (which we now demonstrate), we believe the current version is not misleading and, therefore, decided to keep it as is to focus on the key findings.

In re-analyzing scenarios without steel and ammonia industry relocation, we identified an error that over-counted existing capacities when we do not allow relocation, leading to an excess of CAPEX spending. With this error corrected, the cost reductions achieved by allowing steel and

ammonia production to relocate within Europe reduced from 22 bn€/a (2.5%) to 2.5 bn€/a (0.3%) in the no-imports case. We have updated the text accordingly.

To be more transparent about the uncertainty in modeling relocation of steel and ammonia production, we have added a paragraph to the methods section outlining its limitations (starting with 'A limitation of the relocation modelling...'). We also added a supplementary note with the mathematical formulation of the model, including an explanation of how demand relocation is modeled.

RC-2.3: 2. Techno-economic costs and their implication.

The first version of this article utilised cost projections for 2030, stating that infrastructure must be built well in advance. I am disappointed to see these costs moved to 2040, as it adds even more uncertainty to your results. I would argue that this does not really make the article easier to contextualise, as you compare to targets for 2030 in some sections (P8L212), state that your import costs reflect those of studies focused on 2050 (P15L450) and use potentials from 2050 for biomass (P18L595).

Suggestions:

- The date should be consistent with the general question you are trying to answer.
- Justify and contextualise the implications of this change and the imbalances it creates in the limitations section, following your initial arguments.

AR-2.3: We are sorry to see your disappointment with our change to choose technology assumptions for 2040 instead of 2030. Several researchers suggested using more progressive assumptions for our long-term scenarios as feedback to our preprint. Even though they could understand our reasoning for not using assumptions for 2050 (long-term projections are very uncertain, and much infrastructure must be built years before the carbon-neutrality goal is reached), they argued that technology assumptions for 2030 might understate upcoming technological developments and overstate the costs of decarbonization. We felt that using assumptions for 2040 would be a reasonable compromise.

We have taken this discussion as an opportunity to expand the net-zero emission scenarios shown in Figures 2 and 5 by additional scenarios with technology assumptions for 2030 and 2050. The added analysis demonstrates that the technology projection year does not affect the key conclusions. Considering the three years for techno-economic assumptions (2030, 2040, 2050), the potential cost savings from imports range between 3.6% and 5.4% around our central import cost estimate and decrease with technological development. We observed no change in dynamics when limiting the choice of import vectors in Figure 2 for alternative years for techno-economic assumptions. We have amended the text to include these added insights and added uncertain technological developments to the limitations.

As for the specific sections you mention: (1.) In P8L212, we make a connection with the RePowerEU targets for 2030 regarding the share of imports rather than the overall volume. As there are no specific targets for later years, the desired import shares could be indicative of future targets. (2.) In P15L450, we compare our import costs to those of Genge et al. [1], which use 2030 and 2050 costs, neither of which matches the 2040 costs. Leveraging the added sensitivity analysis, we now also compare import costs for 2030 and 2050. (3.) In P18L595, we use biomass potentials for 2050 as the infrastructure build-out argument for using technology assumptions for earlier years is weakened for biomass. Unlike technology deployment, biomass potential depends more on long-term land use changes and ecological constraints than infrastructure development timelines. Overall, we feel that these comparisons /assumptions do not create untenable inconsistencies in the model.

RC-2.4: 3. Making scenarios clearer.

The article features a wide array of scenarios dedicated to both parametric and structural changes in both of your models. To make them easier to trace, I suggest detailing them in a table in the supplementary material. Stating if a figure relates to your default case explicitly rather than implicitly (e.g., including 'default' or something similar in figures like Supplementary Figure 11) would also help readers.

AR-2.4: That is a great suggestion. We have now added Supplementary Table 1, which gives an overview of the 352 scenarios run for this study and which figures they contribute to. We have also revised some figure captions to clarify what scenarios are shown.

RC-2.5: Additional detailed comments:

P16L536: in "Existing hydro-electric power plants are included, as well as nuclear power plants 536 built before 1990 or currently under construction according...", perhaps you meant "built after 1990"?

AR-2.5: An obvious mistake; sorry about this! We have corrected the statement to read "built after 1990".

Reviewer #2

RC-2.6: I co-reviewed this manuscript with one of the reviewers who provided the listed reports. This is part of the Nature Communications initiative to facilitate training in peer review and to provide appropriate recognition for Early Career Researchers who co-review manuscripts.

The improved documentation of the model's assumptions is appreciated. However, the authors do not directly state which version of PyPSA-Eur they used in the GitHub repository. Similarly, the link to the model's documentation points to the newest version of the model, meaning that it'll deviate from the version used for this study over time. Since it is not versioned, it does not really

ensure reproducibility. There are ways to infer which version of the tool was used, but they are GitHub submodule links that may break over time. I have some suggestions:

- Directly state which version of the key tools was used for the study. This can be through an environment file or a simple README section.
- Consider versioning your model's documentation.

AR-2.6: For the study, we use derivatives of TRACE v1.1 and PyPSA-Eur v0.13.0 with custom changes regarding the modelling of imports and related infrastructure that are not part of the official releases. We now mention this in the revised manuscript and README file.

We also now provide the documentation for past versions of PyPSA-Eur (specifically <https://pypsa-ur.readthedocs.io/en/v0.13.0-docs-fix>),¹ and point to the latest common version of the documentation in the manuscript and README file.

In response to your comment, we have further taken additional measures to facilitate reproducing our analysis:

- **Instructions:** We have expanded the README file with detailed step-by-step instructions to reproduce the results, from installing the environments to running the *snakemake* commands. Instructions for high-performance computing are left out since they are highly dependent on the user's infrastructure. However, we reference the *snakemake* documentation for cluster integration.
- **Environment:** We provide two frozen *conda* environment files (one for TRACE, one for PyPSA-Eur) that specify the fixed versions of all used packages, and added instructions how to install them.
- **Code and data:** In addition to the Github repository (<https://github.com/fneum/import-benefits>), we have deposited the code (including the various *git* submodules) for long-term archiving on Zenodo (<https://doi.org/10.5281/zenodo.14872324>), separate from the results data repository (<https://doi.org/10.5281/zenodo.14872183>).

Reviewer #3

RC-2.7: Dear authors, thanks for the substantial revision. For most of my questions and comments, I can see from the response that you have either amended the text, included new explanations or new model runs in the main article or in the supplementary information. There are however comments for which it is unclear to me if you have addressed the comments apart from in the response. As I asked the questions and provided the comments on behalf of potential future readers and not for

¹Recent changes in how *Read the Docs* builds the documentation required a small amendment to v0.13.0.

my own curiosity, I would appreciate that they are addressed in either the main article or if need be in the supplementary material.

AR-2.7: First of all, we are glad we could adequately address your questions. In response to your feedback, we went through your previous comments and checked that they are addressed in the revised manuscript. We understand that this may have been hard to track, given the amount of text changes in the previous revision. We list below how the responses were incorporated in the manuscript's previous revision, and outline where we have amended our explanations in the current revision:

1. **Demand modelling and distribution:** In the previous revision, we addressed this by adding an overview of competing supply and consumption options in Supplementary Fig. 2, an overview of spatial distribution of demands in Supplementary Fig. 4, and an overview total final energy and non-energy demands in Supplementary Fig. 5. The spatial distribution of demands is described in PyPSA-Eur: Overview of European energy system model, in the paragraph starting 'Spatially, the model resolves...'. In the current revision, we added the mathematical model formulation including how exogenous and endogenous demands are modelled in Supplementary Note 1.1 and expanded our description of temporal demand variations in PyPSA-Eur: Overview of European energy system model, in the paragraph starting 'Temporally, the model is...'.
2. **Backup power and heating supply:** In the previous revision, we addressed this by including backup heat/power options in Supplementary Fig. 2 and discussing how import scenarios affect backup requirements in Interactions of import strategy & domestic infrastructure, in the paragraph starting 'Changes in the magnitude...'. In the current revision, we added a short section Backup heat and power options to the methods.
3. **Fuel pathways and blue pathways:** In the previous revision, we included an overview of fuel conversion routes, including 'blue' pathways, in Supplementary Fig. 2. We also mention the consideration of 'blue' and 'grey' options in Technical constraints of synthetic fuel production. In the current revision, we have added Supplementary Note 1.1, which includes a description of how conversion processes are modelled.
4. **DAC costs:** In the previous revision, we added a versioned link to the techno-economic assumptions we used,² which includes the assumptions used for direct air capture. The rest of our response gives context, but, in our opinion, is too specific for the manuscript or supplementary material.
5. **Uses of Fischer-Tropsch fuels:** In the previous revision, we included an overview of fuel conversion routes in Supplementary Fig. 2, including Fischer-Tropsch fuels and competing technologies. We also outline where they are used in Supplementary Fig. 5. In the current revision, we have added Supplementary Note 1.1, which includes a description of how costs and competition between conversion processes is modelled.

²<https://github.com/pypsa/technology-data/releases/tag/import-benefits-v2>

6. **Wind and solar potentials (EU/globally):** In the previous revision, we have added a methods section on Wind and solar potentials for PyPSA-Eur, and similarly a description of wind and solar potentials in exporting countries (TRACE: Import supply chain modelling), in the paragraph starting ‘To determine the levelised cost...’.
7. **Biomass types and usage options:** In the previous revision, we have added an abridged version of our response in a methods section on Biomass potentials. In the current revision, we emphasize that energy crops or biomass imports are not considered and added a note on how biogenic CO₂ can be provided. We also corrected a typo for the total sum of biogas and solid biomass potentials.
8. **Description of TRACE and coupling to PyPSA-Eur:** In the previous revision, we have improved the description of the TRACE model in TRACE: Import supply chain modelling as well as the coupling to PyPSA-Eur in Coupling of import options to European model.
9. **Benefit of PtX flexibility:** In the previous revision, we mention the benefit of PtX flexibility in Interactions of import strategy & domestic infrastructure, in the paragraph starting with ‘Changes in the magnitude...’. In the current revision, we have added the model equations in Supplementary Note 1.1, which also outlines how must-run constraints of power-to-X processes are implemented.
10. **References for PtX flexibility:** In the previous revision, we have added references for our modelling decisions on PtX flexibility in Technical constraints of synthetic fuel production and showed in Table 1 the impact of relaxing must-run constraints. In the current revision, we added a small note in Technical constraints of synthetic fuel production that strict operational requirements may be a competitive disadvantage where Fischer-Tropsch and methanol synthesis compete.
11. **Temporal flexibility of imports and ports:** In the previous revision, we have expanded our description of the coupling of TRACE and PyPSA-Eur in Coupling of import options to European model, including an explanation how temporal aspects of imports are dealt with and what existing port infrastructure we consider, in the paragraph starting with ‘However, imports may be further restricted...’.
12. **Shipping costs from Chile:** In the previous revision, we have expanded our description of the TRACE model in TRACE: Import supply chain modelling, including an explanation how the transport from exporter to importer by ship, pipeline or power transmission line is modelled in the paragraph starting with ‘For each vector, an annual reference...’. The paragraph includes an abridged version of our review response. In the current revision, we have added a brief note that also the capital costs for ships and pipelines are included. For more details we prefer to refer to Hampp et al. [2], since we believe the question was quite specific.
13. **Uniform WACC:** In the previous revision, we expanded the description of the TRACE model, including the uniform WACC assumption in TRACE: Import supply chain modelling,

in the first paragraph.

14. **Relocation modelling:** In the previous revision, we added an extensive section Industry relocation modelling for crude steel and ammonia production. In the current revision, this section has been expanded by a discussion of its limitations, in a paragraph starting with ‘A limitation of the relocation modelling...’.
15. **Infrastructure repurposing:** In the previous revision, we have rerun all scenarios with pipeline retrofitting enabled and mention this in Gas and electricity network modelling. In Coupling of import options to European model, we write that we consider only new hydrogen import terminals, no repurposed LNG terminals.
16. **Realistic pipeline and cavern storage sizes:** We believe this question was quite specific, and while we gladly expanded on it in our responses, we prefer not to include this aspect in the manuscript. Note, that the peer review files will also be published.
17. **Import cost sensitivity:** In the previous revision, we have expanded the uncertainty range of import costs relative to the European technology costs to $\pm 50\%$ for Figure 5. In the current revision, we added additional scenarios where we also varied the overall years for techno-economic assumptions (2030, 2040, 2050; see Figures 2 and 5).
18. **Visualisation and comparison to other studies:** In the previous revision, we cleaned up all figures and added a paragraph comparing our results to a selection of other studies in the Discussion, in the paragraph starting with ‘In relation to other studies...’.

Reviewer #4

RC-2.8: I co-reviewed this manuscript with one of the reviewers who provided the listed reports. This is part of the Nature Communications initiative to facilitate training in peer review and to provide appropriate recognition for Early Career Researchers who co-review manuscripts.

I would like to thank the authors for their efforts in addressing the first round of comments and providing an updated manuscript, and providing additional sensitivities. Although the manuscript looks polished and updated, some questions still need to be clarified.

AR-2.8: We appreciate your assessment of our revised manuscript, and your acknowledgement of the work we have done in the previous revision. We outline below how we have addressed your remaining concerns in the current revision.

RC-2.9: 1) If this aspect is addressed in your manuscript or supplementary information, please clarify it again in the main manuscript, similar to your response, by explaining the reasoning behind not spatially fixing the hydrogen demand. Currently, approximately 8 MT of grey or black hydrogen is produced domestically in Europe, with production typically located near industrial clusters. Will these existing projects become stranded assets, or are they expected to relocate as well?

AR-2.9: Thank you for the suggestion. We have incorporated our previous response you referenced into the main text. In the methods section PyPSA-Eur: Overview of European energy system model, we now write:

Most other hydrogen demands are spatially variable. Only a small demand of 5 TWh a⁻¹ in the chemicals industry (excluding liquid feedstocks) remains, which is offset by spatially fixed hydrogen production of around 10 TWh a⁻¹ from chlor-alkali electrolysis for chlorine production. High-temperature industrial heat is supplied by methane, shipping and aviation use carbonaceous fuels, and land transport is fully electrified. In district heating and the power sector, backup hydrogen capacities are endogenously sized and sited just as the production capacities of hydrogen derivatives (Fischer-Tropsch, methane, methanol), which account for more than 80% of the hydrogen consumption. Since the model optimizes the siting and operation of these fuel synthesis plants and electrolyzers, many demands are spatially variable (e.g. electricity demand for electrolyzers or hydrogen demand for methanolisation).

In the results section Interactions of import strategy & domestic infrastructure, we now write:

However, the main reason why hydrogen consumption is mainly concentrated in regions with low-cost production is that over 80% of the hydrogen is used to produce electrofuels for aviation, shipping, and chemical feedstocks, compared to about 10% for crude steel and ammonia production. These liquid fuels can be transported at lower cost to airports, ports, and industrial sites across Europe than hydrogen. Consequently, there is low impetus for transporting hydrogen directly, resulting in a hydrogen network that is much smaller than envisioned in the European Hydrogen Backbone.

Existing grey/black hydrogen production capacities are not considered. Instead, green, blue, and grey hydrogen production capacities are endogenously sized and sited in the model using a greenfield approach. We believe this approach is justified for several reasons: (1.) the existing production of 7.9 Mt/a is small compared to the 59 Mt/a of electrolytic hydrogen production in net-zero carbon emission scenarios; (2.) as usage of hydrogen is shifted away from refining fossil fuels (4.5 Mt of 7.9 Mt current production)³ to large quantities of methanol and Fischer-Tropsch fuels (49.4 Mt), reliance on existing refinery locations becomes less critical; (3.) many existing capacities will reach their end of life within the next two to three decades, offering an opportunity for technological and locational renewal without causing significant stranded assets; (4.) continued use of grey/black hydrogen would largely be incompatible with climate-neutrality targets under low carbon sequestration volumes; (5.) the retrofitting of existing facilities with CCS is generally considered to be more expensive than building new facilities with CCS from the start.

In the revised manuscript, we now mention that existing hydrogen production capacities other than chlor-alkali electrolysis are not considered:

³See https://hydrogeneurope.eu/wp-content/uploads/2023/10/Clean_Hydrogen_Monitor_11-2023_DIGITAL.pdf, page 7.

Existing hydrogen production capacities from fossil gas reforming are not considered, as they are expected to reach end of life over the model horizon.

We do not believe that this assumption impacts our findings.

RC-2.10: 2) In your modeling exercise, you do not account for the costs associated with the relocation of hydrogen demand or the relocation of other industrial activities, nor do you consider the final transportation costs of the produced hydrogen to the end consumer. While you acknowledge that hydrogen demand may not remain fixed in the future, the model does not appear to incorporate the additional benefits and costs associated with shifting activities or pathways. As a result, Figure 7 in the main text depicts a highly concentrated hydrogen network and electrolysis production, primarily in 3-4 European countries, which absorbs most hydrogen-related activities. This outcome seems skewed and differs significantly from Figure 7, which was presented during the first round of revisions. (Please explain the reasons behind those differences from the initial submitted manuscript). Consequently, while the solution represents a valid modeling result (solved to optimality), it becomes difficult to evaluate its novelty or its relevance for large-scale energy system solutions that could inform future policy.

AR-2.10: It is correct that the costs of relocating steel and ammonia production and other socio-economic effects are not considered, but we believe that we are transparent about this model limitation and its implications for interpreting the results. We have now added a paragraph in the methods section:

A limitation of the relocation modelling of steel and ammonia production is that it only considers the cost of energy in the siting of these industries. Other factors, such as impacts on regional economies and local jobs, integration with other production processes, or availability of other existing infrastructure are not considered, largely due to a lack of data. The resulting relocation patterns should therefore be interpreted with caution as they might underestimate total relocation costs and frictions. We allow domestic relocation nevertheless in most scenarios, as it would be inconsistent to allow steel and ammonia imports from abroad while preventing relocation within Europe.

We also now demonstrate the limited impact relocation of steel and ammonia production within Europe has on the potential cost reductions from imports from abroad, even as costs of relocating are neglected (Figure 2).

It is also important to note that relocation of crude steel and ammonia production is not the main driver of the regional concentration of hydrogen hubs (green steel and ammonia production only accounts for around 10% of the hydrogen consumed in the model). It stems mainly from the cost-effective siting of future power-to-liquid plants (more than 80% of hydrogen consumption), largely replacing current (imported) fossil oil usage in the shipping, aviation, and chemicals industries. Transporting the resulting liquid fuels is more cost-effective than transporting hydrogen directly, which explains the small hydrogen network. We believe that it is plausible that a majority

of the new power-to-X infrastructure will not be bound heavily by current industrial clusters and concentrate in few regions, although we acknowledge that optimisation models tend to extremise such dynamics. We have added an explanation for the regional concentration of hydrogen consumption in Interactions of import strategy & domestic infrastructure:

However, the main reason why hydrogen consumption is mainly concentrated in regions with low-cost production is that over 80% of the hydrogen is used to produce electrofuels for aviation, shipping, and chemical feedstocks, compared to about 10% for crude steel and ammonia production. These liquid fuels can be transported at a lower cost to airports, ports, and industrial sites across Europe than hydrogen. Consequently, there is low impetus for transporting hydrogen directly, resulting in a hydrogen network that is much smaller than envisioned in the European Hydrogen Backbone.

The differences in power-to-X siting compared to the initial submission are caused by reducing the waste heat utilisation potential from 100% to 25%, which you encouraged in the previous review iteration and we agreed with. As demonstrated in Supplementary Fig. 26 (top row), the potential revenue from using all power-to-X waste heat is high enough in Central European district heating networks to offset hydrogen transport costs from more remote regions. This leads to more geographically diverse production sites for hydrogen derivatives and more hydrogen pipelines. We discuss this effect in Interactions of import strategy & domestic infrastructure:

To realise these benefits, Fischer-Tropsch and Haber-Bosch plants tend to be geographically distributed where space heating demand is high (e.g. Paris or Hamburg) (Supplementary Fig. 26), which increases hydrogen network build-out to 98 TWkm (+72%) compared to the reference scenario with 25% waste heat utilisation.

We find that our results add substantial value to the political discourse around green energy and energy-intensive material imports. We provide 352 scenarios that demonstrate a large option space, which allows policymakers to develop an informed import strategy that keeps energy costs low while also offering maneuvering room to consider non-cost factors (e.g. energy security). We give detailed explanations for many of the dynamics we observe (e.g. regarding dependencies between import vectors/volumes and domestic infrastructure, implications of domestic power-to-X flexibility for backup requirements, effect of waste heat usage on power-to-X siting, effects of over-/undershooting cost-optimal import levels, preferences for certain import fuels, and many more).

RC-2.11: 3) Furthermore, while the model accounts for multiple imports and an optimal allocation of hydrogen and derivative demands, the benefits are reported to amount to €37.1 billion per year (or a 4.44% improvement in the objective function), mainly driven by reduced renewable capacity expansion, smaller networks and storage requirements, and decreased electrolysis and related infrastructure. However, these results focus on a minimum-cost perspective (without cost components of comment 2) and do not consider critical economic impacts, such as local value creation around industrial hubs. For example, the steel industry alone generates €166 billion in annual

turnover, representing 1.3% of EU GDP and providing 328,000 direct jobs (as highlighted in the European Commission’s report:⁴ alongside numerous indirect and dependent jobs. Additionally, the current analysis disregards first-mover effects, the development of R&D, and other socio-economic benefits. These aspects should be highlighted to present a more comprehensive evaluation of large-scale hydrogen deployment and its broader implications for regional development and policy orientation.

AR-2.11: We concur that impacts on the regional economy and local jobs are valid concerns. However, since green steel imports as well as relocation of crude steel production within Europe are increasingly discussed, we believe that it is appropriate to include these potential real-world dynamics as options in the model. With the current revision, we now provide additional sensitivity analyses without relocation of crude steel and ammonia production (in particular, in Figure 2) to alleviate the criticism about simplified relocation modelling. The complementary scenarios demonstrate that assumptions about domestic crude steel/ammonia industry relocation do not affect the cost reduction potentials from imports substantially, which is the main focus of our study.

While we believe that incorporating implications of regional economic developments falls outside the scope of this study, we now highlight it in the limitations section (in addition to our mention of it when we explain how relocation is modelled in the methods section):

We also do not assess potential impacts on the regional economy and local employment effects within Europe as some energy-intensive manufacturing relocate in the model.

As for the numbers mentioned for the steel industry, we would like to clarify that we consider imports of intermediate products (such as HBI or crude steel), while the revenues and jobs above seem to include finished steel production and related industries. According to the Prodcom database, which provides statistics on the production value of goods produced by companies in the EU, a large share of the value is added in the finishing steps of steel production⁵ – which is also where most jobs are.⁶ If only semi-finished products are imported, only part of the jobs and value creation of finished steel products is offshored. This would allow Europe to retain the most value-adding parts of the value chain and most jobs, while outsourcing the most energy-intensive processes. This strategy has been coined as a potential ‘sweet spot’ for the steel industry by Verpoort et al. [3]. We cover this aspect with a note in the concluding paragraphs:

Moreover, policies favoring local energy supply chains and importing intermediary products like sponge iron could be favoured to preserve European jobs while outsourcing only the most energy-intensive processes.

⁴https://ec.europa.eu/commission/presscorner/detail/fr/memo_16_805

⁵See https://ec.europa.eu/eurostat/databrowser/view/ds-056120__custom_15338751/default/table, or specifically <https://ec.europa.eu/eurostat/databrowser/bookmark/30cfcaf4-1dea-4529-83f6-350cca8c7da7>, where codes 24101* refer to basic iron products, 24102* refer to crude steel, and 2410{3,4}* refer to hot/cold rolled steel.

⁶See https://www.agora-industry.org/fileadmin/Projekte/2021/2021-06_IND_INT_GlobalSteel/A-EW_298_GlobalSteel_Insights_WEB.pdf, Figure 11.

We have also clarified in the manuscript that we only model the relocation of crude steel and hot briquetted iron, not finished steel products.

RC-2.12: 4) The authors claim to have described their modeling approach and mathematical formulation in previous work. However, the references provided do not include any explicit formulation or methodology for relocating demands (e.g., steel), leaving this aspect of the model unclear.

AR-2.12: We have now reprinted the mathematical formulation of the optimisation problem from Neumann et al. [4] in Supplementary Note 1.1 with some updates and modifications explaining how endogenous/exogenous demands and relocation are modelled; in particular, in the paragraph starting “By modelling the conversion...”.

RC-2.13: 5) In your discussion you claim:

Policymakers in Europe might prefer such easy-to-implement systems featuring lower domestic infrastructure requirements, reuse of existing infrastructure, lower technology risk, and reduced land usage for broader public support than the most cost-effective solution. Moreover, policies favoring local energy supply chains and importing intermediary products like sponge iron could preserve European jobs while outsourcing only the most energy-intensive processes. However, in shifting potential land use and infrastructure conflicts abroad, where population densities are often lower, potential exporting countries must weigh the prospect of economic development against internal social and environmental concerns, particularly in countries with a history of colonial exploitation.

These elements are not clearly supported by the literature or by explicit outcomes of your research. For instance, what is meant by the term easy-to-implement systems? Did you demonstrate how much land usage your modeling approach could save? Additionally, did you showcase a solution that is indeed cost-effective?

Overall, I recommend that the authors reconsider the final paragraph of the paper. It would be more appropriate for the discussion-conclusion to synthesize the key results of the manuscript rather than presenting hypothetical claims.

AR-2.13: Thank you for expressing your concern. We would like to emphasize that we do not seek to make any claims in this paragraph (or give that appearance), but rather to discuss the implications of our results in a broader context. We have already summarized the key results in the first paragraph and believe that broadening the perspective in the concluding paragraphs beyond the paper’s direct scope is common practice and helpful for understanding how the paper’s insights might be applied. Therefore, we wish to retain this paragraph, but have carefully revised the language to avoid any impression of making claims that are not supported by our results.

The small differences in cost observed between some scenarios are particularly relevant because factors other than pure costs, which are not reflected in our infrastructure optimisation model, might then drive ~~the designs of~~ import strategies. ~~The~~ To some, the relatively limited cost benefit of imports and offshoring of industrial production may speak against imports. Concerns about energy ~~sovereignty would~~ security could motivate more domestic supply and diversified imports. For instance, shipborne imports of hydrogen derivatives ~~would~~ could be preferred to reduce pipeline lock-in and to mitigate the risks of sudden supply disruptions and ~~abuse~~ exercise of market power. From a practical perspective, it may also be more appealing to focus on carriers that are already globally traded commodities and to prefer infrastructure offering quick deployment.

Policymakers in Europe might prefer ~~such easy to implement~~ alternative systems featuring, for instance, lower domestic infrastructure requirements, reuse of existing infrastructure, lower technology risk, and reduced land usage for broader public support than the most cost-effective solution. Moreover, policies favoring local energy supply chains and importing intermediary products like sponge iron could be favoured to preserve European jobs while outsourcing only the most energy-intensive processes. However, in shifting potential land use and infrastructure conflicts to abroad, where population densities are often lower, potential exporting countries ~~must~~ would need to weigh the prospect of economic development against internal social and environmental concerns, particularly in countries with a history of colonial exploitation. Ultimately, Europe's energy strategy ~~must~~ would likely seek to balance cost savings from green energy and material imports with broader concerns like geopolitics, economic development, public opinion, and the willingness of potential exporting countries in order to ensure a swift, secure, and sustainable energy future. Our research shows that there is maneuvering space around Europe's energy import strategies to accommodate such non-cost concerns.

RC-2.14: Figure 6, in the legend, is missing the hydrogen transmission costs; are they negligible?

AR-2.14: Yes, hydrogen network costs have been grouped to “other” in the legend as its costs remained below the selected threshold for individual technology groups to be shown. The hydrogen pipeline costs across all scenarios, including new and retrofitted pipelines, did not exceed 1.35 bn€/a.

Reviewer #5

RC-2.15: Thank you for the very thorough revisions and detailed answers to the reviewers' comments. I have no further comments since all my suggested improvements have been dealt with.

AR-2.15: We are glad we could address all your concerns. No further action required.

RC-2.16: I have checked the repository visually: the full code and data are available and well-documented for the experiment to be reproduced. I did not execute the code due to the complexity of the setup.

AR-2.16: In response to reviewer #2, we have further improved code archiving and instructions to reproduce all scenarios. No further action required.

References

- [1] Genge, L., Scheller, F. & Müsgens, F. Supply costs of green chemical energy carriers at the European border: A meta-analysis. *International Journal of Hydrogen Energy* **48**, 38766–38781 (2023).
- [2] Hampp, J., Düren, M. & Brown, T. Import options for chemical energy carriers from renewable sources to Germany. *PLOS ONE* **18**, e0262340 (2023).
- [3] Verpoort, P. C., Gast, L., Hofmann, A. & Ueckerdt, F. Impact of global heterogeneity of renewable energy supply on heavy industrial production and green value chains. *Nature Energy* **9**, 491–503 (2024).
- [4] Neumann, F., Zeyen, E., Victoria, M. & Brown, T. The potential role of a hydrogen network in Europe. *Joule* **7**, 1793–1817 (2023).

Author Response to Reviews of

Green energy and steel imports reduce Europe's net-zero infrastructure needs

Fabian Neumann, Johannes Hampp, Tom Brown

Revision #3: May 5, 2025

RC-3.n: Reviewer Comment, AR-3.n: Author Response, ■ *Manuscript text*

Reviewer #1

RC-3.1: The authors have adequately dealt with all recommendations. No further comments or issues!

AR-3.1: No further action required.

Reviewer #2

RC-3.2: I co-reviewed this manuscript with one of the reviewers who provided the listed reports. This is part of the Nature Communications initiative to facilitate training in peer review and to provide appropriate recognition for Early Career Researchers who co-review manuscripts.

I partially reviewed the code and documentation and found it transparent enough. The modifications to ensure version history are appreciated.

AR-3.2: No further action required.

Reviewer #3

RC-3.3: Thanks for the thorough answers and reviews I am now satisfied that the article is ready for publication. In the end, the title could however be adjusted to better reflect the actual content of the article and the commodities which are imported including steel, which is the main novelty of the article.

AR-3.3: We have changed the title from *Energy Imports and Infrastructure in a Carbon-Neutral European Energy System* to *Green energy and steel imports reduce Europe's net-zero infrastructure needs*.

Reviewer #4

RC-3.4: I co-reviewed this manuscript with one of the reviewers who provided the listed reports. This is part of the Nature Communications initiative to facilitate training in peer review and to provide appropriate recognition for Early Career Researchers who co-review manuscripts.

AR-3.4: No further action required.